# Structure Learning from Time-Series Data with Lag-Agnostic Structural Prior

**Taiyu Ban,**\* **Changxin Rong,**\* **Xiangyu Wang,**† **Lyuzhou Chen**
**Yanze Gao, Xin Wang, Huanhuan Chen**†
School of Computer Science and Technology
University of Science and Technology of China, Hefei, China

## Abstract

Learning instantaneous and time-lagged causal relationships from time-series data is essential for uncovering fine-grained, temporally-aware interactions. Although this problem has been formulated as a continuous optimization task amenable to modern machine learning methods, the integration of coarse-grained lag-agnostic causal priors, an important and commonly available form of prior knowledge, remains largely unaddressed. To address this gap, we propose a novel framework for structure learning from time series to integrate lag-agnostic priors, enabling the discovery of lag-specific causal links without requiring precise information on the exact lag of causality. We introduce formulations to precisely characterize the lag-agnostic priors, and demonstrate their consequential and process-equivalence to priors, maintaining consistency with the intended semantics of the priors throughout optimization. We further analyze the challenge for optimization due to the increased non-convexity by lag-agnostic prior constraints, and introduce a data-driven initialization to mitigate this issue. Experiments on both synthetic and real-world datasets show that our method effectively incorporates lag-agnostic prior knowledge to enhance the recovery of fine-grained, lag-aware structures.

## 1 Introduction

Time-series data, which captures the dynamic evolution of variables over time, are fundamental to research across both science and artificial intelligence (Sarropoulos et al., 2021; Kim et al., 2024), where uncovering the underlying temporal causal mechanisms is a critical task (Li et al., 2023; Rajapakse and Zhou, 2007). This temporal causal relationship is characterized by a time lag, i.e., the causality between variables manifests after some delay. Such *lag-specific* interactions[1] are known as inter-slice causality, or intra-slice causality when the interaction is instant (i.e., at zero lag).

Accordingly, uncovering time-series causal mechanisms can be formalized as a structure learning task aimed at recovering the lag-specific structure from temporal data. In this formulation, inter-slice causality is represented by a set of structures corresponding to different time lags, while intra-slice causality is captured by a directed acyclic graph (DAG). By expressing the DAG constraint through a smooth characterization (Zheng et al., 2018), the problem becomes a continuous optimization task, making it compatible with modern machine learning techniques (Pamfil et al., 2020).

In practice, researchers usually possess prior knowledge about partial causal relationships, which is essential for recovering more interpretable and insightful structures than relying on data alone, and should be actively incorporated into structure learning. To this end, Sun et al. (2023) proposed integrating causal knowledge with specific time lags into structure learning from time-series data. However, such precise lag-specific information is rarely available in real-world settings[2]. Instead, prior knowledge is more commonly available in the form of summarized causal relationships, without

---

\*Equal contribution.
†Corresponding authors: Xiangyu Wang (`sa312@ustc.edu.cn`) and Huanhuan Chen (`hchen@ustc.edu.cn`).

[1]Lag-specific causality refers to causal links with known, specific time lags (including instant causality).
[2]This is reflected in the scarcity of publicly available time-series data with annotated lag-specific structures.

specifying exact time lags (Marbach et al., 2010; Runge et al., 2019a). We refer to this more accessible form of knowledge as a *lag-agnostic prior* due to its invisibility to time lags of causality[3].

Incorporating lag-agnostic priors is to specify the presence (or absence) of a causal edge without knowledge of its exact time lag. Due to this ambiguity, such constraints cannot be directly imposed on specific structural parameters. This complicates the precise representation of prior knowledge, posing a challenge to preserve the intended information without introducing unintended bias toward specific lags during optimization. Moreover, the ambiguity introduces additional non-convexity[4], making the optimization landscape more difficult and prone to poor local optima resulting from suboptimal lag selections. In summary, structure learning from time-series data with lag-agnostic priors is a significant yet underexplored challenge in the field.

To address this gap, we propose a novel continuous optimization framework for incorporating lag-agnostic structural priors into structure learning from time-series data. We first show the issue of process-inequivalence in an intuitive maximum-based characterization of lag-agnostic priors. Despite the consequential equivalence (i.e., it ultimately enforces the same structural conditions as the lag-agnostic priors), the maximum-based formulation unexpectedly introduces bias toward specific lags during the optimization process, which is misaligned with the prior that the time lag is unknown. Targeting this issue, we introduce formulations that are both consequentially equivalent and process-equivalent, which are proven to maintain consistency with the intended semantics of the priors throughout optimization. This ensures that the prior constraints are not only satisfied at convergence but are also fully respected during the entire optimization trajectory.

For the optimization behavior, we show the increased non-convexity caused by incorporating lag-agnostic priors, which leads to more suboptimal local optima compared to the task without prior. Early optimization dynamics, and in particular, the initialization, can potentially bias which lag-specific edge satisfies the constraint, thus sticking into bad optima. To mitigate this, we introduce a data-driven initialization strategy that leverages unconstrained structure learning to guide early updates and promote convergence to favorable optima.

We conduct extensive experiments on both synthetic and real-world datasets. Results show that our method effectively leverages coarse-grained, lag-agnostic prior knowledge to significantly improve the recovery of fine-grained, lag-aware causal structures. This work provides practical insights and robust strategies for incorporating commonly available causal priors into dynamic causal discovery.

Our main contributions are listed as follows:

1. We introduce a continuous optimization framework for time-series structure learning to incorporate lag-agnostic structural priors, enabling the use of widely available coarse-grained knowledge to guide the recovery of temporally-resolved causal mechanisms.

2. We analyze the process-ineqivalent issue in the characterization of lag-agnostic priors, and propose formulations that are both consequentially and process-equivalent to the intended priors throughout optimization.

3. We analyze the increased non-convexity caused by lag-agnostic constraints, and propose a data-driven initialization strategy that facilitates the convergence to favorable optima[5]. It is validated by comprehensive experiments on synthetic and real-world datasets.

Related work, theoretical details, and complete experimental results are provided in the Appendix.

## 2 PRELIMINARIES

This section introduces the preliminaries of structure learning from time-series data and describes how to incorporate lag-specific edge constraints into the learning process.

**Notations**  For matrix A, we use $A_{s_1:e_1, s_2:e_2, \ldots}$ to denote slicing along multiple dimensions, where each $(s_k : e_k)$ specifies an inclusive start and end index for the $k$-th dimension. If a dimension

---

[3]See Appendix A.5 for a discussion of the sources and reliability of lag-agnostic prior.

[4]See Example 1 for details on the non-convexity induced by lag-agnostic priors.

[5]Note that our method does not guarantee finding the global optimum.

is omitted, it indicates selection of all elements along that axis. We write $\sum A$ for the sum of all elements in matrix $A$, and $|A|$ for the element-wise absolute value of $A$.

## 2.1 Structure Learning from Time Series

Let $X \in \mathbb{R}^{T \times d}$ be a multivariate time series, where $X_t \in \mathbb{R}^d$ denotes the observations of $d$ variables at time $t$. Temporal dependencies are modeled using a linear vector autoregressive (VAR) process:

$$X_t = \sum_{\tau=0}^{L} W_\tau X_{t-\tau} + \epsilon_t,$$

where $W_0$ captures intra-slice (instantaneous) dependencies, $W_\tau$ for $\tau > 0$ capture inter-slice (lagged) dependencies, and $\epsilon_t$ is zero-centered noise. The maximum lag $L$ defines the longest allowable delay in the model. Each $W_\tau \in \mathbb{R}^{d \times d}$ parameterizes directed edges from $X_{t-\tau}$ to $X_t$, defining a structure modeling the generation process of the time-series data.

Zheng et al. (2018) introduces a smooth constraint of acyclicity, which is used to ensure the acyclic intra-slice structure $W_0$ by Pamfil et al. (2020):

$$h(W_0) := \mathrm{Tr}\left(e^{W_0 \circ W_0}\right) - d = 0,$$

where $\circ$ denotes the Hadamard (element-wise) product.

The full modeling objective can be written as a regression task over time-series observations.

$$\min_\theta \mathcal{L}(X; \{W_\tau(\theta)\}_{\tau=0}^{L}), \quad \text{subject to } h(W_0(\theta)) = 0. \tag{1}$$

Here, $W_\tau(\theta)$ denotes the lag-$\tau$ structure derived from parameters $\theta$. The model can be either linear, where $\theta = \{W_\tau\}$, or nonlinear, where $\theta$ determines the lagged structure $W_\tau(\theta)$ via a predefined mapping (Sun et al., 2023; Zheng et al., 2020). Hence, constraints on the structure are directly applicable to the nonlinear settings with the defined mapping from $\theta$ to $W_\tau$.

The augmented Lagrangian approach (Zheng et al., 2018) is used to optimize this problem. After optimization, a thresholding step extracts the edge structure:

$$\hat{W}_\tau = \mathbb{I}(|W_\tau| > \delta) \circ W_\tau, \tag{2}$$

where $\delta > 0$ is the edge threshold, and $\mathbb{I}(\cdot)$ is the element-wise indicator function, which values 1 if the inner condition holds and values 0 otherwise.

## 2.2 Incorporating Lag-Specific Edge Constraints

To enforce the presence or absence of a specific lagged edge $(W_s)_{ij}$, there are two approaches, a hard constraint and a soft penalty. The hard way directly constrain the corresponding parameters:

$$\min_{\theta \in \Theta_{ij,s}(\delta)} \mathcal{L}\left(X; \{W_\tau(\theta)\}_{\tau=0}^{L}\right), \quad \text{subject to } h(W_0) = 0, \tag{3}$$

where $\Theta_{ij,s}(\delta) := \{\theta \mid |(W_s(\theta))_{ij}| \geq \delta\}$ enforces the presence of edge $(W_s)_{ij}$, or $\Theta_{ij,s}(0) := \{\theta \mid (W_s(\theta))_{ij} = 0\}$ enforces its absence. This strict constraint was used in the work by Sun et al. (2023).

Alternatively, one can add a loss term to softly encourage edge presence or absence:

$$\min_\theta \mathcal{L}\left(X; \{W_\tau(\theta)\}_{\tau=0}^{L}\right) + \mathrm{ReLU}\left(\delta - |(W_s(\theta))_{ij}|\right), \quad \text{subject to } h(W_0) = 0, \tag{4}$$

where the penalty $\mathrm{ReLU}(\delta - |(W_s(\theta))_{ij}|)$ encourages the presence of edge $(W_s)_{ij}$, while $|(W_s(\theta))_{ij}|$ alone can be used to encourage absence. The soft constraint allows a balance between prior enforcement and data fitting, but does not guarantee strict adherence.

**Discussion.** While the hard constraint strictly enforces prior knowledge, the soft constraint allows trade-offs with data fit. However, in the case of *lag-agnostic* priors, the exact lag of a causal edge is unknown, making parameter-level constraints ill-defined. Therefore, the hard constraint becomes unsuitable in this setting, and soft, flexible formulations are preferred.

**Remark 1.** *For simplicity, we assume a linear setting where $W_\tau(\theta) = W_\tau$ in the remainder of the paper. Nonetheless, the proposed methods and analysis also apply to nonlinear parameterizations.*

# 3 TIME-SERIES STRUCTURE LEARNING WITH LAG-AGNOSTIC STRUCTURAL CONSTRAINTS

## 3.1 PROBLEM FORMULATION

Let $X \in \mathbb{R}^{T \times d}$ be a multivariate time series, where $T$ is the number of time steps and $d$ the number of variables. Let $W_\tau \in \mathbb{R}^{d \times d}$ denote the lag-specific structure matrix at lag $\tau \in \{0, 1, \ldots, L\}$, where each entry $(W_\tau)_{ij}$ represents the causal influence from variable $j$ at time $t - \tau$ to variable $i$ at time $t$. We use $W_{0:L}$ (or simply $W$) to denote the set of all lagged structure matrices.

Let $\mathcal{C}_p, \mathcal{C}_a \in \{0, 1\}^{d \times d}$ be binary masks encoding lag-agnostic structural priors: $\mathcal{C}_p$ specifies the presence of edges, and $\mathcal{C}_a$ specifies their absence. A small threshold $\delta > 0$ is used to determine whether an edge is considered present.

**Definition 1** (Lag-Agnostic Edge Presence). *For a node pair $(i, j)$ with $(\mathcal{C}_p)_{ij} = 1$, the lag-agnostic presence constraint requires that edge $(i, j)$ at least one lag $\tau \in \{0, 1, \ldots, L\}$ exists, $|(W_\tau)_{ij}| > \delta$.*

**Definition 2** (Lag-Agnostic Edge Absence). *For a node pair $(i, j)$ with $(\mathcal{C}_a)_{ij} = 1$, the lag-agnostic absence constraint requires that $(W_\tau)_{ij} = 0$ for all $\tau \in \{0, 1, \ldots, L\}$.*

Given these constraints, the structure learning objective is formulated as:

$$\min_{W_{0:L}} \mathcal{L}(X; W_{0:L}), \quad \text{subject to } h(W_0) = 0, \ W_{0:L} \models \mathcal{C}_a, \mathcal{C}_p, \tag{5}$$

The adherence to lag-agnostic constraints is defined formally as:

$$W_{0:L} \models \mathcal{C}_a \iff \forall (i, j) \text{ with } (\mathcal{C}_a)_{ij} = 1, \quad (W_\tau)_{ij} = 0 \text{ for all } \tau, \tag{6}$$

$$W_{0:L} \models \mathcal{C}_p \iff \forall (i, j) \text{ with } (\mathcal{C}_p)_{ij} = 1, \quad \max_\tau |(W_\tau)_{ij}| > \delta. \tag{7}$$

The absence of a lag-agnostic edge can be directly enforced by zeroing the corresponding entries across all lags, as described in Section 2.2. Therefore, the focus of this work lies in how to properly formulate and enforce the presence of lag-agnostic edges during optimization.

## 3.2 PROCESS-INEQUIVALENCE IN CHARACTERIZING LAG-AGNOSTIC EDGE PRESENCE

We consider the problem of modeling lag-agnostic edge presence using a smooth, non-negative constraint function. Let $(p(W))_{ij} \geq 0$ represent the presence penalty for edge $(i, j)$, such that $(p(W))_{ij} = 0$ indicates satisfaction of the lag-agnostic presence prior. With this setup, we extend the objective in Equation (5) to include soft constraint penalties as follows:

$$\min_{W_{0:L}} \mathcal{L}(X; W_{0:L}) + \lambda_p \sum_{i,j} (\mathcal{C}_p \circ p(W))_{ij}, \quad \text{subject to } h(W_0) = 0, \tag{8}$$

Here, the prior constraint is relaxed into the objective via a weighted penalty, allowing for greater stability in the presence of the inherent ambiguity associated with lag-agnostic edge semantics.

**Maximum-Based Formulation and Process-Inequivalence**    A natural and direct approach for encoding lag-agnostic edge presence is to penalize the maximum magnitude across all lags, in line with the definition in Equation (7):

$$(p_{\max}(W))_{ij} = \text{ReLU} \left( \delta - \max_\tau |(W_\tau)_{ij}| \right). \tag{9}$$

Although this formulation is logically equivalent to the lag-agnostic prior at convergence, it is *not process-equivalent*. That is, it does not preserve the prior intention throughout optimization.

In particular, directly applying a penalty to $\max_\tau |(W_\tau)_{ij}|$ introduces a bias toward whichever lag happens to dominate early in training, leaving the rest lags unconsidered, as illustrated below:

**Proposition 1.** *Let $(i, j)$ be a lag-agnostic edge specified to be present, and assume the data-fitting loss satisfies $\nabla_{|(W_\tau)_{ij}|}\mathcal{L} \geq 0$ for all $\tau$ during optimization (push all $(W_\tau)_{ij}$ toward zero[6]). Suppose $\forall \tau \neq \tau_0, \delta > |(W_{\tau_0})_{ij}| > |(W_\tau)_{ij}|$ at initialization. Let $W^{opt}$ be the solution of the structure learning problem defined in Equation (8) using the maximum-based prior loss (Equation 9). Then, for sufficiently large $\lambda_p$, the optimum satisfies: $|(W^{opt}_{\tau_0})_{ij}| \geq \delta$ and $\forall \tau \neq \tau_0, |(W^{opt}_\tau)_{ij}| < \delta$.*

This scenario illustrates a typical trade-off between data fit and adherence to a lag-agnostic prior, where the data-fitting loss does not favor including the prior-specified edge $(i, j)$ at any lag. In this case, the maximum-based formulation selects the lag whose edge has the largest initial absolute weight and disregards other lag options. Once the prior constraint is satisfied by this single lag, the optimizer stops exploring alternative lags, thus incorrectly treating the lag-agnostic prior as lag-specific. We refer to this behavior as *process-inequivalence*.

**Definition 3** (Process Equivalence). *A lag-agnostic loss formulation is process-equivalent if it respects the prior's "lag-unknown" semantics throughout the entire optimization process. It must not introduce explicit bias toward any specific lag, but must instead evaluate all candidate lags holistically to select the one(s) most compatible with the data.*

The early commitment of the maximum-based formulation breaks this core idea of lag-agnostic priors, that no preference exists among candidate lags, and stops the model from correctly identifying the true lag. To address this issue, the process-equivalent formulation is needed to avoid such bias during optimization.

### 3.3 PROCESS-EQUIVALENT FORMULATIONS

The failure of the maximum-based formulation provides an insight that it is necessary to simultaneously impact on edge $(i, j)$ at every lag to avoid bias in specific lags, thus ensuring the process-equivalence to the lag-agnostic prior. Following this insight, we introduce a binary-masked formulation and a logic-dual formulation for the presence of lag-agnostic edges.

**Binary-Masked Formulation** A natural idea is to apply uniform constraints to all lagged versions of a given edge. This leads to the following loss function for a lag-agnostic edge $(i, j)$:

$$(p_{\text{bin}}(W))_{ij} = \mathbb{I}\left(\max_\tau |(W_\tau)_{ij}| < \delta\right) \cdot \sum_\tau \text{ReLU}\left(\delta - |(W_\tau)_{ij}|\right), \tag{10}$$

where $\mathbb{I}(\cdot)$ is the element-wise indicator function. The binary mask activates the loss only if all corresponding lag-specific edges are below the threshold $\delta$. When active, the penalty equally encourages all lag-specific edges to grow.

This design ensures that the constraint behaves in accordance with the intended semantics:

**Proposition 2.** *A lag-agnostic edge $(i, j)$ is present in $W_{0:L}$ if and only if $(p_{bin}(W_{0:L}))_{ij} = 0$, where $p_{bin}(\cdot)$ is defined by Equation (10).*

More importantly, this formulation also maintains consistency during optimization, as shown below:

**Proposition 3.** *Let $(i, j)$ be a lag-agnostic edge with known ordering of conflict degrees with respect to the data-fitting loss:*

$$0 \leq \nabla_{|(W_{\tau_1})_{ij}|}\mathcal{L} < \nabla_{|(W_{\tau_2})_{ij}|}\mathcal{L} < \cdots < \nabla_{|(W_{\tau_L})_{ij}|}\mathcal{L},$$

*where $\tau_1$ corresponds to the lag most aligned with data fit. Suppose all $(W_\tau)_{ij}$ are initialized with $\delta_0 < \delta$. Then, optimizing Equation (8) with the prior penalty $p_{bin}$ will result in edge $(i, j)$ appearing only at lag $\tau_1$.*

This result demonstrates that $p_{\text{bin}}(\cdot)$ selects the lag-specific edge most compatible with the data, at least under an ideally consistent ordering of loss gradients and identical initialization. In this sense, the binary-masked formulation seeks the optimal structure under the lag-agnostic presence constraint, maintaining both consequence and process equivalence with the original prior. This ensures the optimization remains equivalent to the intended causal semantics throughout training.

---

[6]This assumption ensures we analyze the non-trivial scenario where the data alone is insufficient to satisfy the prior constraint, thus forcing the lag-agnostic loss to actively influence the optimization path.

**Logic-Dual Formulation** We now introduce a second process-equivalent formulation by deriving the edge presence constraint as the logical dual of the edge absence under lag-agnostic priors:

$$(a(W))_{ij} = 0, \quad \text{where} \quad a(W) = \sum_\tau |W_\tau|.$$

Here, $|W_\tau|_{ij} = 0$ serves as the condition for the absence of a lag-specific edge, and the summation expresses an "AND" logic: all corresponding lag-specific edges must satisfy this absence condition.

In contrast, the presence of a lag-agnostic edge follows an "OR" logic: the edge is considered present if at least one lag-specific edge is active. We capture this behavior using a product-based formulation:

$$(p_{\text{or}}(W))_{ij} = \prod_\tau \text{ReLU}\left(\delta - |(W_\tau)_{ij}|\right), \tag{11}$$

where $\text{ReLU}(\delta - |(W_\tau)_{ij}|)$ evaluates to zero if any lag-specific edge exceeds the threshold $\delta$. The product thus encodes a logical "OR": the constraint is satisfied (i.e., the penalty vanishes) when any one of the edges is sufficiently strong.

**Proposition 4.** *A lag-agnostic edge $(i, j)$ is present in $W_{0:L}$ if and only if $(p_{or}(W_{0:L}))_{ij} = 0$, where $p_{or}(\cdot)$ is defined in Equation (11).*

This formulation, like the binary-masked version, simultaneously considers all lag-specific edges during optimization and remains process-equivalent to the original prior (cf. Proposition 3).

Compared to the binary-masked formulation, the logic-dual formulation is fully continuous, avoiding the discontinuities introduced by the indicator function. This results in smoother optimization dynamics. However, the product-based formulation suffers from scale sensitivity: as the number of candidate lags increases, the magnitude of the loss can decrease rapidly, even when the constraint is violated, leading to weaker optimization signals.

To mitigate this, we normalize the loss magnitude as follows:

$$(\bar{p}_{\text{or}}(W))_{ij} = \prod_\tau \frac{1}{\delta} \text{ReLU}\left(\delta - |(W_\tau)_{ij}|\right), \tag{12}$$

where each ReLU term is scaled by $\delta$ to standardize its contribution. This normalization ensures that, when all $(W_\tau)_{ij} = 0$, the loss yields a consistent penalty regardless of the number of lags, preserving balanced optimization across varying temporal horizons.

## 3.4 Data-Driven Initialization for Stable Optimization

The lag-agnostic edge presence constraint introduces additional non-convexity into the structure learning objective, as stated in the following result.

**Proposition 5.** *Let $\min_W \mathcal{L}(W; X)$ be a convex, linear VAR optimization problem. Then, the augmented problem*

$$\min_W \mathcal{L}(W; X) + \mathcal{C}_p \circ \bar{p}(W) \tag{13}$$

*is non-convex for a set of lag-agnostic edge-presence constraints $\mathcal{C}_p \in \{0, 1\}^{d \times d}$ and our proposed prior loss $\bar{p}(W)$.*

This added non-convexity can destabilize optimization and lead to convergence to suboptimal solutions. The following toy example illustrates this effect:

**Example 1** (Multiple Local Optima under Lag-Agnostic Constraint)**.** *Consider a time series with two variables, $X_1$ and $X_2$, over three time steps and maximum lag $L = 2$, with:*

$$X_{:,1} = [0, \ 0, \ 4], \quad X_{:,2} = [4, \ 2, \ 1].$$

*Assume a lag-agnostic prior indicating that $X_2 \to X_1$. We fit a linear model predicting $X_{3,1}$ as:*

$$\hat{X}_{3,1} = (W_0)_{2,1} X_{3,2} + (W_1)_{2,1} X_{2,2} + (W_2)_{2,1} X_{1,2},$$

*and optimize the following loss with a binary-masked lag-agnostic constraint[7]:*

$$\min_{W} \; \mathcal{L}(W) + 100 \cdot \mathbb{I}\left(\max_{\tau} |W_\tau|_{2,1} < 1\right) \sum_{\tau} ReLU(1 - |W_\tau|_{2,1}),$$

$$\mathcal{L}(W) = \left(4 - (W_0)_{2,1} - 2(W_1)_{2,1} - 4(W_2)_{2,1}\right)^2 + \|W\|_1. \tag{14}$$

*This setup yields at least three local optima:*

$$(i) \; W_{:,2,1} = [1, \; 0, \; 0.72], \quad (ii) \; W_{:,2,1} = [0, \; 1, \; 0.47], \quad (iii) \; W_{:,2,1} = [0, \; 0, \; 1],$$

*with corresponding losses $\mathcal{L} = 1.73, \; 1.48, \; 1.0$, respectively. Although all satisfy the lag-agnostic constraint, only (iii) is a good optima.*

This example (see Appendix B for proof, and see Appendix C for more discussions) demonstrates that incorporating lag-agnostic priors transforms the originally convex problem of unconstrained time-series structure learning (without the acyclicity constraint) into a non-convex one. As a result, different initializations may lead the optimization process to suboptimal solutions such as (i) or (ii), where the lag-agnostic constraint is prematurely satisfied by an incorrect lag. This underscores the importance of a principled initialization strategy to promote convergence to favorable optima.

**Two-Stage Optimization via Data-Driven Initialization**  A practical solution is to initialize using the outcome of the unconstrained data-fitting problem, which guides optimization toward the most data-consistent lag. For Example 1, this strategy naturally favors the correct lag, leading the optimization to recover solution (iii) and effectively avoid the suboptimal solutions (i) and (ii). Specifically, we adopt a two-stage process:

**Stage 1:** Solve the unconstrained structure learning problem:

$$\widehat{W}_{0:L}^{\text{data}} = \arg \min_{W_{0:L}} \mathcal{L}(X; W_{0:L}) \quad \text{subject to } h(W_0) = 0. \tag{15}$$

**Stage 2:** Use the result as initialization for the lag-agnostic constrained objective:

$$\min_{W_{0:L}} \; \mathcal{L}(X; W_{0:L}) + \lambda_p \sum \left(\mathcal{C}_p \circ p(W)\right), \; \text{subject to } h(W_0) = 0, \; \text{with } W_{0:L}^{(0)} = \widehat{W}_{0:L}^{\text{data}}. \tag{16}$$

This formulation preserves the original task structure, enabling existing optimization and thresholding methods to be applied directly. In our implementation, we use either the binary-masked loss $p_{\text{bin}}(W)$ or the logic-dual loss $p_{\text{or}}(W)$ to encode lag-agnostic edge presence constraints.

## 4 EXPERIMENTS

This section presents the main experimental results and analysis. We begin with experimental setup.

**Synthetic Data.**  We generate synthetic time-series data based on Erdős–Rényi (ER) random graphs, where each possible edge is included independently with equal probability. To control graph sparsity, we denote settings as ER-$k$, where $k$ is the ratio of the number of edges to the number of nodes $d$. Linear time-series samples are generated using a linear VAR process under two noise settings: Gaussian (Gauss) and Exponential (Exp). The sequence contains $T$ samples (time steps). For incorporating lag-agnostic priors, we randomly select a proportion $p\%$ of the true lag-agnostic edges as prior. The generation of nonlinear data is detailed in the specific corresponding experiments.

**Real-World Data.**  We evaluate on the DREAM4 gene regulatory network dataset (Marbach et al., 2009), a standard benchmark for time-series structure learning. It provides gene expression trajectories with known regulatory interactions under various perturbations. As the ground-truth specifies edges but not time lags, this dataset is well-suited for evaluating models with lag-agnostic structural priors.

---

[7]This example also applies to the logic-dual formulation.

**Metrics.** We report structural metrics including Structural Hamming Distance (SHD), True Positive Rate (TPR), False Discovery Rate (FDR), F1 score, and the recovery rate of prior lag-agnostic edges. We also use the area under the ROC curve (AUROC) for summary graph evaluation, and test-set regression loss to assess predictive accuracy when lag-specific ground truth is unavailable.

**Setup.** Default settings: prior percentage $p = 80$, loss weight $\lambda_p = 0.5$, edge threshold $\delta = 0.1$, maximum lag $L = 3$; others follow the backbone default. Backbones inlcude DYNOTEARS (Pamfil et al., 2020), LIN (Liu and Kuang, 2023), RHINO Gong et al. (2023), and NTS-NOTEARS (Sun et al., 2023). Our methods are denoted as Backbone& (binary-masked loss) and Backbone* (logic-dual loss). Initialization strategies include "Init 0" (zero), "Init Data" (data-driven), and "Init Random" (random). We also test the maximum-based formulation that is theoretically flawed, denoted as Backboneˆ. Experiments run on an AMD Ryzen 9 7950X (4.5 GHz) CPU and 32 GB RAM.

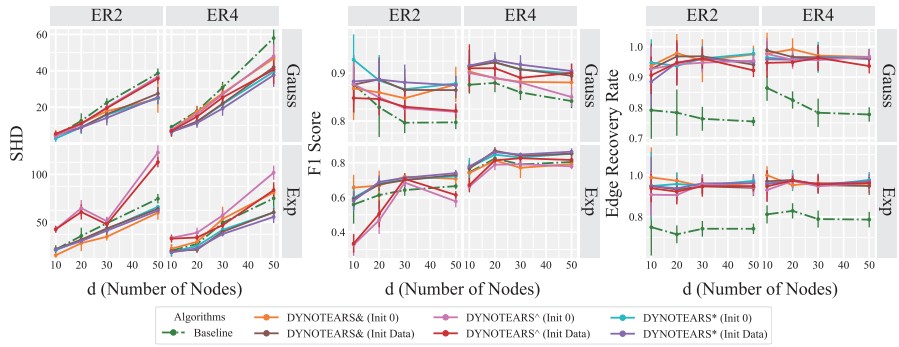

Figure 1: Comparison between time-series structure learning with and without lag-agnostic priors.

## 4.1 EVALUATION OF OVERALL PERFORMANCE

We compare our method, augmented with lag-agnostic edge presence priors, against the data-only baseline across varying node counts and graph settings. Figure 1 reports SHD, F1 score, and edge recovery rate on synthetic data with 250 time steps. A more comprehensive evaluation is available in Appendix E. We observe that both the logic-dual formulation (DYNOTEARS* with Init Data) and the binary-masked formulation (DYNOTEARS& with Init Data) consistently outperform the data-only baseline. Moreover, methods initialized with data-based strategies consistently outperform their zero-initialized counterparts, highlighting the benefit of data-driven initialization in improving optimization stability.

As for the formulation of prior losses, we observe that 1) The flawed maximum-based formulation (DYNOTEARSˆ, Init 0) underperforms the other process-equivalent methods (DYNOTEARS& and DYNOTEARS*, Init 0) in most cases, which is expected due to its inherent process-inequivalence. 2) DYNOTEARS* consistently outperforms DYNOTEARS and DYNOTEARS& under Init 0, suggesting that its fully continuous loss formulation leads to more stable convergence without a principled initialization. 3) Even when Init Data slightly masks the flow of the flawed formulation, our process-equivalent methods paired with Init Data still maintain the best performance stably.

## 4.2 COMPARISON WITH LAG-SPECIFIC PRIOR INTEGRATION

To demonstrate the robustness of our method, we compare it not only against a "no prior" baseline but also against a Lag-Specific (LS) baseline given imperfect information. In this setup, the LS baseline receives correct summary graph priors, but the associated lag is incorrect for $q\%$ of them (denoted LS-q). Table 1 presents partial results from experiments using 20-node ER4 graphs, Gaussian noise, and a maximum lag of 5, evaluated across varying percentages of prior knowledge. A more comprehensive evaluation is available in Appendix E.

The results in Table 1 are revealing. While the oracle baseline with perfect lag information (LS-0) achieves the best performance, our lag-agnostic method *performs remarkably close to this upper bound without requiring any specific lag knowledge*. However, LS's performance is brittle: With just

Table 1: Partial comparison results between our method with lag-specific prior integration methods.

| Method | 5% Priors | | 10% Priors | | 30% Priors | | 50% Priors | |
|---|---|---|---|---|---|---|---|---|
| | SHD↓ | F1↑ | SHD↓ | F1↑ | SHD↓ | F1↑ | SHD↓ | F1↑ |
| DYNOTEARS | 13.00±3.10 | 0.91±0.02 | 13.00±3.10 | 0.91±0.02 | 13.00±3.10 | 0.91±0.02 | 13.00±3.10 | 0.91±0.02 |
| LS-0, Perfect Lags | **10.33±2.73** | **0.93±0.02** | **8.00±2.37** | **0.95±0.02** | **4.83±1.94** | **0.97±0.01** | **4.83±1.33** | **0.97±0.01** |
| LS-10, 10% Error | 11.67±3.44 | 0.92±0.02 | 8.17±2.64 | **0.95±0.02** | 6.67±2.58 | 0.96±0.02 | 6.50±2.43 | 0.96±0.02 |
| LS-30, 30% Error | 12.50±3.62 | 0.92±0.03 | 9.50±3.27 | 0.94±0.02 | 8.83±1.33 | 0.94±0.01 | 8.33±2.16 | 0.95±0.02 |
| LS-50, 50% Error | 12.50±3.62 | 0.92±0.03 | 10.67±2.73 | 0.93±0.02 | 11.17±1.47 | 0.93±0.01 | 11.83±1.94 | 0.92±0.01 |
| DYNOTEARSˆ (Init 0) | 12.00±2.45 | 0.92±0.03 | 10.17±2.51 | 0.93±0.03 | 8.50±2.24 | 0.94±0.02 | 7.83±1.63 | 0.95±0.02 |
| DYNOTEARSˆ (Init Data) | 11.50±2.26 | 0.92±0.02 | 9.33±2.34 | 0.94±0.02 | 7.83±2.04 | 0.94±0.02 | 6.83±1.47 | 0.95±0.01 |
| DYNOTEARS& (Init 0) | 11.67±2.13 | 0.92±0.02 | 9.83±3.91 | 0.93±0.03 | 6.67±2.10 | 0.96±0.02 | 7.00±2.89 | 0.95±0.03 |
| DYNOTEARS& (Init Data) | 10.83±1.97 | **0.93±0.01** | 8.50±3.78 | 0.94±0.03 | 5.83±1.97 | **0.97±0.01** | 6.17±2.71 | 0.95±0.02 |
| DYNOTEARS* (Init 0) | 10.87±2.52 | **0.93±0.02** | 8.83±2.55 | 0.94±0.02 | 6.00±1.98 | 0.96±0.01 | 5.93±1.92 | 0.96±0.02 |
| DYNOTEARS* (Init Data) | 10.50±2.35 | **0.93±0.02** | 8.00±2.37 | **0.95±0.02** | 5.33±1.86 | **0.97±0.01** | 5.50±1.76 | 0.96±0.01 |

10% incorrect lag information (LS-10), our lag-agnostic method already exceeds its performance. As the noise in the lag increases (LS-30, LS-50), the LS method's performance degrades significantly, falling significantly behind our lag-agnostic approach. This highlights the practical value of our method in real-world scenarios where precise temporal information is often noisy or unavailable. Additionally, the process-equivalent formulations (& and *) consistently outperform the flawed, process-inequivalent formulation (ˆ), confirming that maintaining the intended semantics throughout the optimization process leads to superior and more reliable structural recovery.

## 4.3 GENERALIZATION TO NON-STATIONARY AND NON-LINEAR BACKBONES

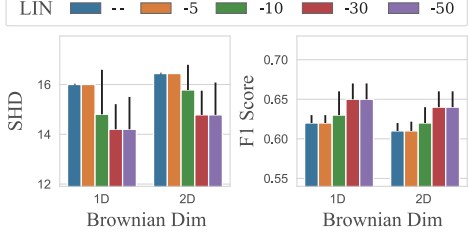
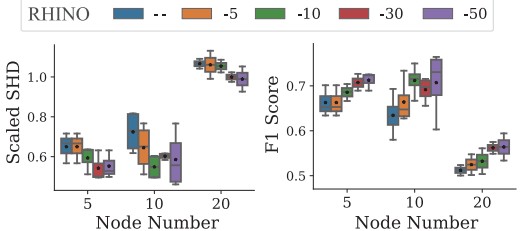

(a) Results for LIN with various prior amount.  (b) Results for RHINO with various prior amount.

Figure 2: Results of applying our method to LIN and RHINO for non-stationary, nonlinear data.

To demonstrate the versatility of our framework, we integrate it with state-of-the-art backbones designed for complex non-stationary and non-linear time-series data: LIN (Liu and Kuang, 2023) and RHINO (Gong et al., 2023). We use the logic-dual loss for lag-agnostic priors and the data-driven initialization. Our method with $p\%$ lag-agnostic priors is denoted as LIN-$p$ or RHINO-$p$ here. We adopted experimental setups from their respective papers to ensure a fair comparison: MLP-based nonlinear data; (LIN) 5-node ER graphs (density=0.5), 1 lag, 1D and 2D Brownian motion, 2 intervention nodes; (RHINO) {5, 10, 20}-node graphs, 2 lags, connection factor=2, spline product noise, intervention history=3. We report results with 5000 samples for LIN and RHINO in Figure 2, with more comprehensive evaluation available in Appendix E. The results show that our lag-agnostic prior framework consistently enhances the performance of both backbones, with improvements scaling directly with the amount of available prior knowledge. These findings provide strong evidence that our method serves as a general and effective plug-in for sophisticated, optimization-based discovery methods, confirming its utility for challenging data conditions.

## 4.4 PERFORMANCE ON REAL-WORLD DATA

We tested our framework on the DREAM4 dataset using various sample sizes ($m$). We supplied our method and a "random-lag" baseline with 25% and 75% of the true lag-agnostic edges as priors. As the lag-specific truth is unknown, we also test static NOTEARS with and without priors. Results of summary graph recovery quality and test-set regression loss are reported in Table 2.

Table 2: Results of data-only and prior-based methods on DREAM4 dataset.

| @Priors | Method | DREAM4-63 | | DREAM4-126 | | DREAM4-189 | |
|---|---|---|---|---|---|---|---|
| | | Loss↓ | AUROC↑ | Loss↓ | AUROC↑ | Loss↓ | AUROC↑ |
| 25% | NOTEARS | 8.20±0.73 | 0.54±0.03 | 8.02±0.92 | 0.55±0.03 | 7.89±0.73 | 0.55±0.03 |
| | NOTEARS+Prior | 7.57±0.52 | 0.66±0.02 | 7.57±0.72 | 0.67±0.02 | 7.66±0.61 | 0.68±0.03 |
| | DYNOTEARS | 7.81±1.14 | 0.58±0.03 | 5.61±1.04 | 0.62±0.04 | 4.08±0.60 | 0.65±0.05 |
| | DYNOTEARS-RandomLag | 7.26±0.63 | 0.64±0.03 | 6.01±0.83 | 0.68±0.04 | 4.94±0.61 | 0.71±0.06 |
| | DYNOTEARSˆ (Init 0) | **6.59±0.13** | 0.70±0.05 | 5.47±0.12 | 0.72±0.04 | 5.42±0.09 | 0.74±0.05 |
| | DYNOTEARSˆ (Init Data) | 7.58±0.11 | 0.58±0.03 | 5.43±0.12 | 0.63±0.03 | 4.35±0.07 | 0.66±0.06 |
| | DYNOTEARS& (Init 0) | 6.60±0.57 | **0.72±0.02** | 5.79±0.70 | **0.74±0.03** | 4.99±0.49 | **0.75±0.05** |
| | DYNOTEARS& (Init Data) | 7.24±0.91 | 0.69±0.02 | **5.19±0.88** | 0.73±0.04 | **3.80±0.53** | 0.74±0.04 |
| | DYNOTEARS* (Init 0) | 6.66±0.54 | 0.71±0.02 | 5.82±0.82 | **0.74±0.04** | 5.08±0.49 | **0.75±0.05** |
| | DYNOTEARS* (Init Data) | 7.19±1.01 | 0.69±0.02 | 5.27±0.92 | 0.73±0.03 | 3.85±0.54 | 0.74±0.05 |
| 75% | NOTEARS | 8.20±0.73 | 0.54±0.03 | 8.02±0.92 | 0.55±0.03 | 7.89±0.73 | 0.55±0.03 |
| | NOTEARS+Prior | 6.68±0.42 | 0.89±0.01 | 6.61±0.53 | 0.89±0.01 | 6.65±0.42 | 0.89±0.01 |
| | DYNOTEARS | 7.81±1.14 | 0.58±0.03 | 5.61±1.04 | 0.62±0.04 | 4.08±0.60 | 0.65±0.05 |
| | DYNOTEARS-RandomLag | 7.45±0.98 | 0.70±0.02 | 6.20±0.72 | 0.75±0.03 | 5.43±0.84 | 0.77±0.05 |
| | DYNOTEARSˆ (Init 0) | 7.60±0.14 | 0.89±0.02 | 5.49±0.14 | 0.90±0.02 | 4.41±0.09 | 0.90±0.02 |
| | DYNOTEARSˆ (Init Data) | 7.58±0.11 | 0.62±0.04 | 5.43±0.12 | 0.67±0.03 | 4.35±0.06 | 0.71±0.06 |
| | DYNOTEARS& (Init 0) | **5.24±0.57** | **0.91±0.01** | 4.70±0.70 | **0.92±0.01** | 4.29±0.64 | **0.92±0.02** |
| | DYNOTEARS& (Init Data) | 6.02±0.78 | 0.90±0.01 | **4.46±0.64** | 0.91±0.01 | **3.25±0.33** | **0.92±0.02** |
| | DYNOTEARS* (Init 0) | 5.28±0.84 | 0.90±0.01 | 4.66±0.74 | **0.92±0.01** | 4.27±0.39 | **0.92±0.02** |
| | DYNOTEARS* (Init Data) | 6.22±0.72 | 0.89±0.01 | 4.88±0.85 | 0.91±0.01 | 3.40±0.42 | **0.92±0.02** |

As shown, our method achieves the best performance across all conditions, outperforming baselines in both predictive accuracy (regression loss) and structure recovery (AUROC). Notably, while priors with imperfect random lags can appear to improve the summary causal graph, they severely degrade the model's predictive ability (126 and 189 samples). Our lag-agnostic approach, however, consistently improves both, demonstrating its practical value and robustness in the real-world setting. Besides, the process-equivalent formulations (& and *) outperforms the flawed formulation (ˆ) in nearly all cases, which demonstrates the necessity of the correct loss formulation.

### 4.5 Ablations, Supplementary Evaluation and Results

Experiments of comprehensive ablations, evaluation of lag-agnostic edge absence, comprehensive parameter analysis, qualitative results, and supplementary results, are presented in Appendix E.

## 5 Conclusion

This paper introduces a novel task: incorporating lag-agnostic structural priors into continuous structure learning from time-series. The goal is to leverage commonly available, coarse-grained prior knowledge to guide the recovery of fine-grained, lag-specific causal structures. We analyze the unique challenges this setting poses, particularly in loss formulation and optimization stability, and propose solutions with both theoretical guarantees and strong empirical performance. Results on synthetic and real-world datasets demonstrate the effectiveness and robustness of our approach. Future work may explore extending lag-agnostic priors beyond individual edges to more expressive forms such as causal paths or partial orders, which are widely used in static structure learning.

### Acknowledgements

This research was supported in part by the National Nature Science Foundation of China (No. 62406302, 62137002), in part by the Natural Science Foundation of Anhui Province (No. 2408085QF195), in part by the Fundamental Research Funds for the Central Universities under Grant WK2150110035, in part by the USTC Research Funds of the Double First-Class Initiative (Grant No. YD9110002085), and in part by the Research Funds of Centre for Leading Medicine and Advanced Technologies of IHM (Grant No. 2025IHM01030).

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

## STATEMENT ON LLM USAGE

In the preparation of this manuscript, a large language model (LLM), specifically Google's Gemini, was used as a writing assistance tool. The human authors first drafted and completed each section of the paper, establishing all core ideas, analyses, and conclusions. The LLM was then used solely to refine and polish the language of these pre-written drafts to improve clarity, grammar, and readability. The LLM's role was strictly that of an editing assistant; it did not contribute to the research ideation, experimental design, or the generation of any results or substantive content. All authors take full responsibility for the final content of this paper.

## A    RELATED WORK

### A.1    STRUCTURE LEARNING FROM STATIONARY TIME-SERIES DATA

Structure learning from stationary time-series data, whose fundamental statistical properties don't change over time, spans a broad set of methodologies, which can be mainly grouped into constraint-based, functional causal model (FCM)-based, gradient-based, Granger causality-based, and miscellaneous approaches.

**Constraint-based Methods.**    Constraint-based approaches, such as tsFCI (Entner and Hoyer, 2010), PCMCI (Runge et al., 2019b), and CDANs (Ferdous et al., 2023), rely on conditional independence testing to infer temporal dependencies. These methods aim to recover a partially directed acyclic graph (PDAG) by exploiting the statistical independencies implied by the causal Markov condition and faithfulness assumptions.

**FCM-based Methods.**    Functional Causal Model (FCM)-based methods, such as VarLiNGAM (Hyvärinen et al., 2010) and TiMINo (Peters et al., 2014), assume a structural equation model with functional dependencies and noise independence. These methods are identifiable under stronger functional assumptions, but may be restrictive when applied to complex nonlinear dynamics.

**Gradient-based Methods.**    Gradient-based approaches reformulate structure learning as a differentiable optimization problem. DYNOTEARS (Pamfil et al., 2020) extends the NOTEARS framework (Zheng et al., 2018) to time-series data by jointly learning intra-slice and inter-slice dependencies in linear VAR models. NTS-NOTEARS (Sun et al., 2023) further incorporates neural networks and soft supervision for dynamic causal structure discovery.

**Granger Causality-based Methods.**    Granger causality-based methods (Granger, 1969) evaluate causality through predictive performance: if the past of variable $X_j$ improves prediction of $X_i$, then $X_j$ is said to Granger-cause $X_i$. Granger causality assumes strictly inter-slice causality, and typically ignore instantaneous (intra-slice) effects. Modern machine learning methods are actively developing to recover granger causality from time-series data. Representative approaches include neural granger causality (Tank et al., 2018), generalized vector autoregression (GVAR) (Marcinkevičs and Vogt, 2021), neural additive vector autoregression (NAVAR) (Bussmann et al., 2021), and amortized causal discovery (ACD) (Löwe et al., 2022).

**Miscellaneous.**    Other methods, such as oCSE, TCDF (Nauta et al., 2019), NBCB, and PCTMI, include hybrid or specialized approaches tailored to specific domains. These may leverage mutual information, neural attention, or hybrid search strategies and are often evaluated in niche applications.

In summary, constraint-based methods are based on discrete search approaches, while FCM-based methods use analytical approaches to address structure learning from time-series. The gradient-based methods and neural granger causality approaches utilizes continuous optimization methods, which are compatible with our introduced task of incorporating lag-agnostic priors. Even our method is introduced based on the assumption of both intra-slice and inter-slice causality, it can be directly used for granger causality and benefit the community by leveraging rich lag-agnostic priors.

## A.2 Structure Learning from Non-Stationary Time-Series Data

Learning causal structures from real-world time-series data is often complicated by non-stationarity, where shifts in the underlying dynamics can lead to spurious discoveries if stationary models are used (Song et al., 2009; Zhang et al., 2017). To address this, a significant research effort has focused on methods that can adapt to changing data distributions. One prominent strategy is to model the process as a sequence of discrete causal regimes. This includes methods that explicitly detect changepoints to segment the data, like CASTOR (Rahmani and Frossard, 2025), as well as those that infer latent interventions to identify different causal models, such as LIN (Liu and Kuang, 2023).

An alternative line of work models more complex dynamics or leverages richer data. RHINO (Gong et al., 2023), for instance, captures continuously evolving systems by modeling history-dependent noise with deep neural networks. Others, like IDYNO (Gao et al., 2022), use interventional data from active experiments to robustly identify causal graphs that are otherwise unidentifiable from observational data alone.

Investigating the integration of widely available lag-agnostic prior knowledge in such cases is highly valuable but rarely studied. Our work complements these approaches by introducing a principled method to integrate widely available lag-agnostic priors, and we demonstrate its effectiveness on these complex backbones.

## A.3 Use of Prior Knowledge in Differentiable Structure Learning

Prior knowledge has long been recognized as a powerful asset in causal discovery and has been widely integrated into traditional combinatorial structure learning methods (Chen et al., 2016; Li and Beek, 2018; Constantinou et al., 2023). More recently, the differentiable structure learning community (Zheng et al., 2018; Yu et al., 2019; Zheng et al., 2020) has begun exploring the incorporation of various forms of prior knowledge, including edge constraints (Chen et al., 2025; Hasan and Gani, 2022), partial orders (Ban et al., 2024), and ancestral relationships (Wang et al., 2024). These constraint types reflect common structural priors and help connect traditional discrete methods with modern continuous optimization approaches in machine learning (Zhang et al., 2021).

In dynamic causal discovery, however, the integration of prior knowledge remains far less developed. A few recent efforts, such as NTS-NOTEARS (Sun et al., 2023), support hard supervision for known lag-specific edges. This study, however, assume that the exact time lag of a causal relationship is known, which is rarely the case in real-world applications. In practice, domain knowledge from fields such as biology, neuroscience, or industrial systems often indicates the presence of a causal relationship between variables, without specifying when the influence occurs (Marbach et al., 2010).

Despite the broad availability of this coarse-grained form of knowledge, no existing methods systematically support lag-agnostic structural priors, which specify the existence of a causal edge without identifying its exact time lag. This paper addresses that gap by proposing a continuous optimization framework for time-series structure learning that integrates lag-agnostic priors.

## A.4 Relationship to Static Structural Priors

Our work focuses on adapting coarse-grained knowledge, commonly available in static (non-temporal) settings, to the dynamic time-series domain. We formalize the relationship between our **lag-agnostic edge presence** prior and established constraints used in static structure learning (e.g., edge, path, and order constraints) by examining the hierarchy of constraints and the resulting optimization challenges.

**Connection: The Source of Knowledge.** The knowledge utilized in our framework—a high-level summary graph, such as $i \to j$ without a specific lag—often originates from the exact same domain experts or LLM sources used for specifying priors in static causal discovery. Our contribution is the first rigorous method to correctly interpret and enforce this common, lag-agnostic knowledge within a temporal, lag-aware structure learning pipeline.

**A Hierarchy of Constraints and Strength.** In structural discovery, priors exist in a hierarchy of strength, moving from the most restrictive to the least:

$$\text{Edge Presence } (i \to j) \implies \text{Path Existence } (i \rightsquigarrow j) \implies \text{Partial Order } (i \prec j)$$

Our work focuses on the strongest constraint, $i \rightarrow j$ (lag-agnostic edge presence), making it the most stringent form of structural prior introduced to the time-series setting to date.

**Comparison to Partial Orders ($i \prec j$).** A partial order constraint ($i \prec j$) (Ban et al., 2024), is fundamentally a forbidden-type constraint: it forbids any path from $j$ to $i$. This constraint is **gradient-consistent** with the DAG penalty. Both the partial order penalty and the instantaneous DAG constraint enforce acyclicity by pushing structural weights down (toward zero). This alignment results in a relatively stable optimization problem.

**Comparison to Path/Ancestral Constraints ($i \rightsquigarrow j$).** Path/ancestral constraints represent an existence-type prior. The fundamental challenge they present is directly analogous to our problem:

- **Key Similarity (Gradient Conflict):** Both path existence priors and our **lag-agnostic edge existence** prior introduce a penalty that must push weights up (away from zero) to satisfy the existence condition. This is in **direct conflict** with the DAG penalty and regularization, which push weights down. This conflict is the primary source of the optimization instability that both our work and recent ancestral constraint methods Ban et al. (2025) must solve (cf. Example 1 in Section 3.4).

- **Key Difference (Nature of Uncertainty):** While path constraints involve uncertainty over which set of edges forms the path (e.g., $i \rightarrow k \rightarrow j$ vs. $i \rightarrow l \rightarrow j$), our constraint involves uncertainty over the temporal dimension of a single edge, i.e., determining the correct lag among $\{(W_0)_{ij}, (W_1)_{ij}, \ldots, (W_L)_{ij}\}$.

In summary, our work introduces the strongest form of existence prior (edge) to the temporal domain. By doing so, we address a unique structural uncertainty (over lags) that shares the fundamental optimization difficulty of gradient conflict with modern existence-type static constraints. We posit that future work could build on this foundation by exploring the incorporation of weaker lag-agnostic path or order priors into dynamic structure learning.

### A.5 SOURCES AND PRACTICAL AVAILABILITY OF LAG-AGNOSTIC PRIORS

We identify the following sources and factors that contribute to the practical availability and reliability of lag-agnostic knowledge:

**Classical Source: Domain Expertise.** The primary and most established source for causal knowledge is domain expertise (Amirkhani et al., 2016). This expertise is almost always provided at a high-level, coarse-grained, semantic level. For instance, in fields like genetics (relevant to the DREAM4 experiments ) or neuroscience , an expert may confirm that "Gene A regulates Gene B" or "Region X influences Region Y," but they rarely specify the exact time lag of this interaction. This pervasive gap between high-level expert knowledge and the low-level temporal detail required by traditional methods (e.g., NTS-NOTEARS ) is the central motivation for our lag-agnostic framework.

**Emerging Source: Large Language Models.** A powerful, emerging source of high-level causal priors is Large Language Models (LLMs). Recent research has focused on developing reliable frameworks to extract summary causal relationships from LLMs. These methods provide inherently lag-agnostic and rich priors, bypassing the need for expensive and low-efficiency expert specification. The rapid rise of these data-derived and LLM-generated priors makes our work, which provides the formal mechanism to integrate them into time-series models, both timely and practically significant.

The reliability of the LLM-derived priors is ensured through both the derivation stage and the integration stage:

- Prior Derivation: Research shows that edge priors can be made more reliable by combining their derivation using LLM supervision with data-driven methods.

- Prior Integration: Our model's design ensures robustness even to the inherent imperfections in real-world knowledge. By adopting a soft penalty (as discussed in Section 3.2), our framework allows the model to weigh the prior against strong contradictory evidence from the time-series data, making it inherently robust to a degree of prior error.

## B    PROOF OF STATEMENTS

**Proposition 1.** *Let $(i, j)$ be a lag-agnostic edge specified to be present, and assume the data-fitting loss satisfies $\nabla_{|(W_\tau)_{ij}|}\mathcal{L} \geq 0$ for all $\tau$ during optimization (push all $(W_\tau)_{ij}$ toward zero[8]). Suppose $\forall \tau \neq \tau_0, \delta > |(W_{\tau_0})_{ij}| > |(W_\tau)_{ij}|$ at initialization. Let $W^{opt}$ be the solution of the structure learning problem defined in Equation (8) using the maximum-based prior loss (Equation 9). Then, for sufficiently large $\lambda_p$, the optimum satisfies: $|(W_{\tau_0}^{opt})_{ij}| \geq \delta$ and $\forall \tau \neq \tau_0, |(W_\tau^{opt})_{ij}| < \delta$.*

*Proof.* Let $w_\tau = (W_\tau)_{ij}$ and $m(W) = \max_\tau |w_\tau|$. The objective is

$$J(W) = \mathcal{L}(W) + \lambda_p \operatorname{ReLU}(\delta - m(W)) + \rho h(W_0).$$

Assume (i) $\nabla_{|w_\tau|}\mathcal{L} \geq 0$ for every $\tau$ (the data term always contracts each $|w_\tau|$) and (ii) $0 < |w_{\tau_0}| < \delta$ while $|w_{\tau_0}| > \max_{\tau \neq \tau_0}|w_\tau|$ at initialization. Note that the acyclicity loss consistently push all structural parameters toward 0. Hence, we regard $h(W_0)$ as part of $\mathcal{L}$ here as they both push all $(W_\tau)_{ij}$ toward 0 for the lag-agnostic edge $(i, j)$.

We have $m(W) = |w_{\tau_0}|$ and, while $|w_{\tau_0}| < \delta$, the penalty equals $\delta - |w_{\tau_0}|$ with sub-gradient $\partial_{w_{\tau_0}}p = -\lambda_p \operatorname{sgn}(w_{\tau_0})$ and $\partial_{w_\tau}p = 0$ for $\tau \neq \tau_0$. Because $\nabla_{w_\tau}\mathcal{L}$ is bounded on the compact set $\{|w_\tau| \leq \delta\}$, choose $\lambda_p$ larger than that bound. Then

$$\nabla_{|w_{\tau_0}|}J = \nabla_{|w_{\tau_0}|}\mathcal{L} - \lambda_p|w_{\tau_0}| < 0,$$

so a descent step increases $|w_{\tau_0}|$; for every $\tau \neq \tau_0$,

$$\nabla_{|w_\tau|}J = \nabla_{|w_\tau|}\mathcal{L} \geq 0,$$

and descent decreases $|w_\tau|$. Consequently $|w_{\tau_0}|$ grows monotonically and all other magnitudes shrink until the first time $t^*$ with $|w_{\tau_0}(t^*)| = \delta$. At that instant $p(W) = 0$ and remains zero as long as $|w_{\tau_0}| \geq \delta$; if $|w_{\tau_0}|$ ever tries to dip below $\delta$, the same penalty force $-\lambda_p \operatorname{sgn}(w_{\tau_0})$ reactivates and, because $\lambda_p$ still dominates the data term, immediately drives $|w_{\tau_0}|$ back up. Therefore eventually $|w_{\tau_0}| \geq \delta > |w_\tau|$ for every $\tau \neq \tau_0$, so the constraint $m(W) \geq \delta$ is satisfied solely through the single coefficient $w_{\tau_0}$, i.e., the edge at lag $\tau_0$ with the largest initial absolute value. $\qquad\square$

**Proposition 2.** *A lag-agnostic edge $(i, j)$ is present in $W_{0:L}$ if and only if $(p_{bin}(W_{0:L}))_{ij} = 0$, where $p_{bin}(\cdot)$ is defined by Equation (10).*

*Proof.* Write

$$m := \max_{0 \leq \tau \leq L} |(W_\tau)_{ij}|.$$

We first show that presence $\implies (p_{\text{bin}})_{ij} = 0$. Assume the edge is present, i.e. $m \geq \delta$. Then the indicator in (10) equals 0:

$$\mathbb{I}(m < \delta) = 0,$$

hence $(p_{\text{bin}}(W))_{ij} = 0 \cdot (\cdots) = 0$.

Then we show $(p_{\text{bin}})_{ij} = 0 \implies$ presence. Suppose $(p_{\text{bin}}(W))_{ij} = 0$. By (10) the product of two non-negative factors is zero, so at least one factor is zero.

Case 1: $\mathbb{I}(m < \delta) = 0$. Then $m \geq \delta$, hence $\exists \tau$ with $|(W_\tau)_{ij}| \geq \delta$; the edge is present.

Case 2: $\sum_\tau \operatorname{ReLU}(\delta - |(W_\tau)_{ij}|) = 0$. Each term of the sum is non-negative; therefore every term must be 0, which implies $|(W_\tau)_{ij}| \geq \delta$ for some $\tau$ (otherwise each absolute value would be $< \delta$ and

---

[8]This assumption ensures we analyze the non-trivial scenario where the data alone is insufficient to satisfy the prior constraint, thus forcing the lag-agnostic loss to actively influence the optimization path.

the indicator in the first factor would be 1, contradicting the product being 0). Hence the edge is again present.

Since presence implies $(p_{\text{bin}})_{ij} = 0$ and conversely $(p_{\text{bin}})_{ij} = 0$ implies presence, we have the desired equivalence and complete the proof. □

**Proposition 3.** *Let $(i, j)$ be a lag-agnostic edge with known ordering of conflict degrees with respect to the data-fitting loss:*

$$0 \leq \nabla_{|(W_{\tau_1})_{ij}|}\mathcal{L} < \nabla_{|(W_{\tau_2})_{ij}|}\mathcal{L} < \cdots < \nabla_{|(W_{\tau_L})_{ij}|}\mathcal{L},$$

*where $\tau_1$ corresponds to the lag most aligned with data fit. Suppose all $(W_\tau)_{ij}$ are initialized with $\delta_0 < \delta$. Then, optimizing Equation (8) with the prior penalty $p_{bin}$ will result in edge $(i, j)$ appearing only at lag $\tau_1$.*

*Proof.* We have that the data-fitting loss $\mathcal{L}(X; W_{0:L})$ satisfies the fixed ordering

$$0 \ \leq \ g_{\tau_1} < g_{\tau_2} < \cdots < g_{\tau_L}, \qquad g_\tau \ := \ \nabla_{|(W_\tau)_{ij}|}\mathcal{L}. \tag{17}$$

Denote $w_\tau := |(W_\tau)_{ij}|$ $(\tau = 0, \ldots, L)$ and let

$$m := \max_\tau w_\tau.$$

Because all $w_\tau < \delta$ at initialisation, $\mathbb{I}(m < \delta) = 1$ and the prior penalty for the pair $(i, j)$ equals

$$\lambda_p \sum_{\tau=0}^{L} (\delta - w_\tau) \quad \text{(all terms positive)}.$$

The negative gradient of the total objective (restricted to edge $(i, j)$) is

$$-\frac{\partial}{\partial w_\tau}\left(\mathcal{L} + \lambda_p p_{\text{bin}}\right) = -g_\tau \ + \ \lambda_p \qquad (\tau = 0, \ldots, L). \tag{18}$$

Because $g_{\tau_1} < g_{\tau_2} < \cdots < g_{\tau_L}$ by (17), we have

$$-g_{\tau_1} + \lambda_p > -g_{\tau_2} + \lambda_p > \cdots > -g_{\tau_L} + \lambda_p. \tag{19}$$

Thus $w_{\tau_1}$ receives the largest positive ascent among all lags, while every $w_\tau$ starts from the same value. Consequently $w_{\tau_1}$ becomes the unique maximiser $m$ after an arbitrarily small descent step.

As long as $m < \delta$ (so the indicator in $p_{\text{bin}}$ stays 1) the update for every lag keeps the form (18). Inequality (19) remains valid because the ordering (17) is assumed fixed. Hence $w_{\tau_1}$ grows strictly faster than every other $w_\tau$ and reaches $\delta$ first. Let $t^\star$ be the first iteration where $w_{\tau_1} = \delta$. At this moment $m = \delta$ and the indicator in $p_{\text{bin}}$ flips to zero, so the penalty vanishes.

For iterations $t > t^\star$ the objective reduces to $\mathcal{L}$ alone, whose gradients $-g_\tau$ are non-positive (each $g_\tau \geq 0$ by assumption). Thus $\dot{w}_\tau \leq 0$ for every lag. For $\tau \neq \tau_1$ we still have $w_\tau(t) \leq w_\tau(0) < \delta$; no mechanism makes them increase. $w_{\tau_1}(t)$ can decrease but remains $\geq \delta$ in a neighbourhood of $t^\star$, so the presence condition is fulfilled and the prior penalty never re-activates.

Gradient flow therefore converges to a stationary point where

$$w_{\tau_1} \ \geq \ \delta, \qquad w_\tau = 0 \ (\tau \neq \tau_1),$$

which is exactly the result of proposition 3. We complete the proof. □

**Proposition 4.** *A lag-agnostic edge $(i, j)$ is present in $W_{0:L}$ if and only if $(p_{or}(W_{0:L}))_{ij} = 0$, where $p_{or}(\cdot)$ is defined in Equation (11).*

*Proof.* Define $a_\tau := |(W_\tau)_{ij}|$ and write

$$p_{\text{or}} = \prod_{\tau=0}^{L} \text{ReLU}(\delta - a_\tau).$$

($\Rightarrow$) If $p_{\text{or}} = 0$, then the edge is present. Because a product of non–negative numbers is zero only when at least one factor is zero, there exists an index $\tau^\star$ with

$$\text{ReLU}(\delta - a_{\tau^\star}) = 0 \implies \delta - a_{\tau^\star} \leq 0 \implies a_{\tau^\star} \geq \delta.$$

Hence $|(W_{\tau^\star})_{ij}| \geq \delta$; the edge is present at lag $\tau^\star$.

($\Leftarrow$) If the edge is present, then $p_{\text{or}} = 0$. Assume there exists $\tau^\dagger$ with $a_{\tau^\dagger} \geq \delta$. Then

$$\text{ReLU}(\delta - a_{\tau^\dagger}) = 0,$$

so the product contains a zero factor and therefore $p_{\text{or}} = 0$.

Since each direction holds, the equivalence is proved. $\qquad\square$

**Example 1** (Multiple Local Optima under Lag-Agnostic Constraint). *Consider a time series with two variables, $X_1$ and $X_2$, over three time steps and maximum lag $L = 2$, with:*

$$X_{:,1} = [0,\ 0,\ 4], \quad X_{:,2} = [4,\ 2,\ 1].$$

*Assume a lag-agnostic prior indicating that $X_2 \to X_1$. We fit a linear model predicting $X_{3,1}$ as:*

$$\hat{X}_{3,1} = (W_0)_{2,1} X_{3,2} + (W_1)_{2,1} X_{2,2} + (W_2)_{2,1} X_{1,2},$$

*and optimize the following loss with a binary-masked lag-agnostic constraint[9]:*

$$\min_W\ \mathcal{L}(W) + 100 \cdot \mathbb{I}\left(\max_\tau |W_\tau|_{2,1} < 1\right) \sum_\tau ReLU(1 - |W_\tau|_{2,1}),$$

$$\mathcal{L}(W) = \left(4 - (W_0)_{2,1} - 2(W_1)_{2,1} - 4(W_2)_{2,1}\right)^2 + \|W\|_1. \tag{14}$$

*This setup yields at least three local optima:*

*(i) $W_{:,2,1} = [1,\ 0,\ 0.72]$,   (ii) $W_{:,2,1} = [0,\ 1,\ 0.47]$,   (iii) $W_{:,2,1} = [0,\ 0,\ 1]$,*

*with corresponding losses $\mathcal{L} = 1.73,\ 1.48,\ 1.0$, respectively. Although all satisfy the lag-agnostic constraint, only (iii) is a good optima.*

*Proof.* We will show that the three points (with exact values) listed in Example 1,

$$\begin{aligned}
\text{(i) } W^{(i)} &= (1,\ 0,\ \frac{23}{32}), \\
\text{(ii) } W^{(ii)} &= (0,\ 1,\ \frac{15}{32}), \\
\text{(iii) } W^{(iii)} &= (0,\ 0,\ 1),
\end{aligned} \tag{20}$$

are local minimisers of the objective

$$\begin{aligned}
F(W) &= \underbrace{\left(4 - (W_0)_{2,1} - 2(W_1)_{2,1} - 4(W_2)_{2,1}\right)^2}_{\mathcal{L}(W)} + \underbrace{\|W\|_1}_{L_1 \text{ sparsity}} \\
&\quad + 100\,\mathbb{I}\left(\max_{\tau=0,1,2} |W_\tau|_{2,1} < 1\right) \sum_{\tau=0}^{2} \text{ReLU}\left(1 - |W_\tau|_{2,1}\right).
\end{aligned} \tag{21}$$

---

[9]This example also applies to the logic-dual formulation.

Throughout, write $w_\tau := (W_\tau)_{2,1}$ for brevity and consider perturbations $w_\tau + \varepsilon_\tau$ with $\varepsilon := (\varepsilon_0, \varepsilon_1, \varepsilon_2)$ arbitrarily small in Euclidean norm.

First note that for every point in (20), $\max_\tau |w_\tau| = 1$, so the indicator $\mathbb{I}(\max_\tau |w_\tau| < 1)$ is zero and the penalty term is inactive.

Let $P(\varepsilon)$ the perturbed penalty indicator:

$$P(\varepsilon) = \mathbb{I}\Big(\max_\tau |w_\tau + \varepsilon_\tau| < 1\Big).$$

For any perturbation sufficiently small, we have two disjoint cases.

**Case A:** The penalty is triggered $\big(P(\varepsilon) = 1\big)$. Then

$$F\big(W + \varepsilon\big) \;\geq\; 100 \sum_{\tau=0}^{2} \big(1 - |w_\tau + \varepsilon_\tau|\big) \;>\; 100,$$

while $F(W^{(i)}) = 1.73$, $F(W^{(ii)}) = 1.48$, $F(W^{(iii)}) = 1$. This penalty value (100) is larger than every baseline value,

$$F\big(W + \varepsilon\big) - F\big(W^{(k)}\big) \;>\; 0 \quad (k = i, ii, iii).$$

**Case B:** The penalty remains zero $\big(P(\varepsilon) = 0\big)$. This implies $\max_\tau |w_\tau + \varepsilon_\tau| \geq 1$. Because exactly one component in each $W^{(k)}$ equals 1 and the others are below 1, we must have $\varepsilon_{\hat\tau} \geq 0$ for that maximising index $\hat\tau$, while $|\varepsilon_\tau|$ is arbitrarily small for the others.

Consider the case of $W^{(i)} = (1, 0, \frac{23}{32})$, we have that $\varepsilon_0 > 0$. In this region, $F(W)$ is fully continuous, and we only need to prove that for arbitrarily small $\varepsilon_0 > 0$ and $|\varepsilon_1|, |\varepsilon_2|$:

$$F(w_0 + \varepsilon_0, w_1, w_2) \geq F(w_0, w_1, w_2), \tag{22}$$
$$F(w_0, w_1 + \varepsilon_1, w_2) \geq F(w_0, w_1, w_2), \tag{23}$$
$$F(w_0, w_1, w_2 + \varepsilon_2) \geq F(w_0, w_1, w_2). \tag{24}$$

Let

$$t \equiv (4 - w_0 - 2w_1 - 4w_2) = \frac{1}{8}.$$

We have:

$$F(w_0 + \varepsilon_0, w_1, w_2) - F(w_0, w_1, w_2) = (-2t + 1)\varepsilon_0 = \frac{3}{4}\varepsilon_0 > 0,$$

which proves (22).

Besides, we have:

$$F(w_0, w_1 + \varepsilon_1, w_2) - F(w_0, w_1, w_2) = \begin{cases} (-4t + 1)\varepsilon_1 = \frac{1}{2}\varepsilon_1 > 0 & \varepsilon_1 > 0 \\ (-4t - 1)\epsilon_1 = -\frac{3}{2}\varepsilon_1 > 0 & \varepsilon_1 < 0 \end{cases},$$

which proves (23).

Finally, we have:

$$F(w_0, w_1, w_2 + \varepsilon_2) - F(w_0, w_1, w_2) = (-8t + 1)\epsilon_2 = 0 \geq 0,$$

which proves (24).

Hence, we have shown that $F(W^{(i)} + \varepsilon) \geq F(W^{(i)})$ in case B. Parallel calculations for $W^{(ii)}$ and $W^{(iii)}$ give the same results as well.

Both Case A and Case B show that for sufficiently small non-zero perturbations $F(W + \varepsilon) - F(W) \geq 0$. Thus each reference point $W^{(i)}, W^{(ii)}, W^{(iii)}$ satisfies the definition of a local minimiser. $\qquad \square$

## C  Ambiguity-Induced Non-Convexity of Lag-Agnostic Priors

This section discusses why the lag-agnostic priors introduce additional non-convexity to the objective, and illustrates that this is a unique challenge for lag-agnostic prior, which is not encountered by lag-specific priors.

In lag-specific structure learning a prior singles out one weight, e.g. "$(W_1)_{ij} \geq \delta$", and the objective remains convex (quadratic data-loss + $\ell_1$ + linear constraint). A lag-agnostic prior, instead, states only that some lagged edge must be present; the optimisation then decides which lag de-activates the penalty. That logical OR introduces pieces that switch on/off at different locations, yielding a non-convex surface even in the linear-VAR setting.

Now we consider the following example that replace the lag-agnostic constraint in Example 1 with a lag-specific one.

Consider two variables $(X_1, X_2)$ observed at three time steps ($L = 2$):

$$X_{:,1} = [0, 0, 4], \quad X_{:,2} = [4, 2, 1].$$

We learn weights $w_\tau = (W_\tau)_{2,1}$ in

$$\hat{X}_{3,1} = w_0 X_{3,2} + w_1 X_{2,2} + w_2 X_{1,2},$$

using either lag-agnostic prior $p_{\text{LA}}$ in Example 1, or lag-specific prior $p_{\text{LS}}$ on the presence of edge in lag 1, .e., parameter $w_1$ in this formulation. Then we consider the corresponding losses:

$$F_{\text{LA}}(w) = \left(4 - w_0 - 2w_1 - 4w_2\right)^2 + |w_0| + |w_1| + |w_2| + 100\,\mathbb{I}\big(\max_\tau |w_\tau| < 1\big) \sum_{\tau=0}^{2} \text{ReLU}(1 - |w_\tau|),$$

$$F_{\text{LS}}(w) = \left(4 - w_0 - 2w_1 - 4w_2\right)^2 + |w_0| + |w_1| + |w_2| + 100\,\text{ReLU}(1 - |w_1|).$$

Both use edge threshold $\delta = 1$.

Next, we consider the convexity of the lag-specific objective. In $F_{\text{LS}}$, the squared error is a convex quadratic, $\ell_1$ is convex, $\text{ReLU}(1 - |w_1|)$ is convex in $w_1$. Hence $F_{\text{LS}}$ is convex in $(w_0, w_1, w_2)$.

In comparison, the lag-agnostic loss $F_{\text{LA}}$ has been proven to be non-convex, which has the following three local optima:

$$A = (1, 0, 0.72), \quad B = (0, 1, 0.47), \quad D = (0, 0, 1).$$

Because the penalty switches off as soon as any $w_\tau$ reaches 1, each axis direction creates a separate flat basin. Gradient descent starting near $A$, $B$, or $D$ converges to these points, all stationary and satisfying the lag-agnostic prior. Only $D$ minimizes the overall loss, while $A$, $B$ persist as (suboptimal) local minima, precisely the phenomenon highlighted in Example 1.

In summary, without the acyclicity constraint the lag-specific objective $F_{\text{LS}}$ is convex and has a unique (lasso) solution. Replacing the single-lag prior with the ambiguous OR-constraint $p_{\text{LA}}$ produces a non-convex surface with multiple disconnected minima. This non-convexity is intrinsic to lag-agnostic priors and absent in earlier lag-specific formulations, motivating careful initialization and the process-equivalent losses proposed in this paper.

## D  Limitations and Broader Impact

**Limitations.**  While our method demonstrates strong performance across synthetic and real-world datasets while remaining identical computational efficiency of the used backbone model, several limitations remain. First, the method currently assumes no latent confounders or missing variables; its effectiveness under partial observability remains untested. Second, the two-stage optimization (data fit followed by prior-guided refinement) may not always guarantee global optima due to residual non-convexity in the loss landscape. Finally, our method assumes the correctness of the provided priors, which may harm the structure learning if with low prior quality.

**Broader Impact.** This work contributes to improving structure learning in time-series settings where precise temporal annotations are scarce but higher-level causal insights are available, an increasingly common situation in domains such as biology, neuroscience, finance, and sensor systems. By enabling the use of coarse-grained prior knowledge, our method supports more practical and accessible causal discovery pipelines. However, as with all data-driven causal inference tools, there is a risk of overinterpreting or misapplying recovered structures, especially in high-stakes fields like healthcare or policy-making. We emphasize that the results should be validated with domain expertise and, where possible, interventional or experimental studies.

Overall, we hope this work encourages the development of structure learning methods that bridge the gap between real-world prior availability and fine-grained causal inference in dynamic systems.

# E COMPLETE EXPERIMENT RESULTS

This section presents a series of supplementary analyses. We begin by evaluating key modules, testing our method on the non-linear NTS-NOTEARS backbone and integration of edge absence priors. We then present comprehensive parameter analysis results, followed qualitative results including experiments with an unknown maximum lag, and an analysis of varying structural weights. Finally, we provide supplementary results for all experiments.

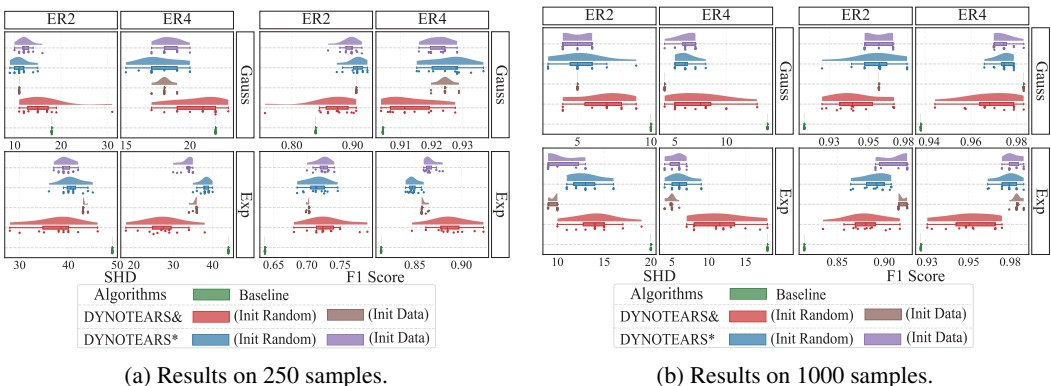

(a) Results on 250 samples.       (b) Results on 1000 samples.

Figure 3: Comparison between data-based initialization and random initialization.

## E.1 ABLATIONS ON INITIALIZATION STRATEGIES

We assess the effectiveness of data-driven initialization by comparing it with random initialization. Specifically, we generate 12 random $W_{0:L}$ initializations and evaluate both two-stage optimization using data-driven initialization (Init Data) and direct optimization from these random initialization (Init Random). Results on 30-node graphs with 250 and 1000 samples are shown in Figure 3.

For DYNOTEARS&, Init Data consistently generally yields better performance than Init Random, reducing the variability across runs and leading to more stable outcomes. For DYNOTEARS*, the difference is less pronounced in the low-sample setting due to its inherently stable, fully continuous loss formulation. However, in the high-sample regime, Init Data clearly outperforms Init Random. These results confirm that data-driven initialization enhances both performance and stability when incorporating lag-agnostic structural priors.

## E.2 ABLATIONS ON PROCESS-EQUIVALENT LOSS FORMULATION

We evaluate the impact of process-equivalence in loss design by comparing our logic-dual formulation (DYNOTEARS*) with the maximum-based formulation, which is denoted as DYNOTEARSˆ. Results are shown in Figure 4. DYNOTEARS* consistently outperforms DYNOTEARSˆ, with the performance gap particularly pronounced under Init 0. This supports our analysis that the maximum-based formulation is highly sensitive to initialization, as it tends to commit early to a single lag based on initial parameter values, violating the intent of lag-agnostic priors.

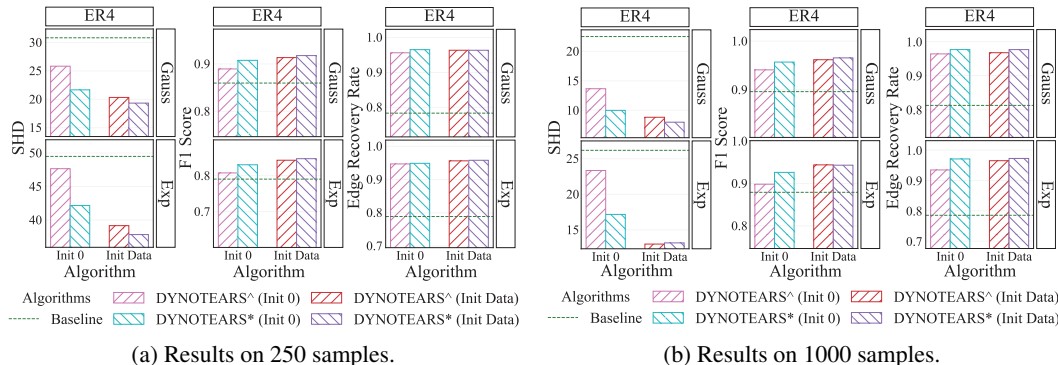

(a) Results on 250 samples.          (b) Results on 1000 samples.

Figure 4: Comparison between process-equivalent and -inequivalent loss formulation.

Table 3: Results on nonlinear data with backbone algorithm NTS-NOTEARS.

| | Sample Count | 200 | | | 1000 | | |
|---|---|---|---|---|---|---|---|
| Model | Method | F1-score↑ | SHD↓ | Recovery↑ | F1-score↑ | SHD↓ | Recovery↑ |
| ANM | NTSNOTEARS | 0.63±0.08 | 48.00±11.73 | 0.60±0.15 | 0.75±0.06 | 31.33±7.76 | 0.68±0.14 |
| ANM | NTS-NOTEARS*-20 | 0.65±0.06 | 47.17±8.45 | 1.00±0.00 | 0.76±0.05 | 31.17±7.03 | 1.00±0.00 |
| ANM | NTS-NOTEARS*-40 | 0.68±0.05 | **45.17±7.03** | 1.00±0.00 | 0.79±0.03 | **28.33±4.93** | 1.00±0.00 |
| ANM | NTS-NOTEARS*-60 | 0.67±0.07 | 49.00±10.22 | 1.00±0.00 | 0.78±0.06 | 31.00±8.88 | 1.00±0.00 |
| ANM | NTS-NOTEARS*-80 | 0.70±0.05 | 46.00±7.64 | 1.00±0.00 | 0.81±0.04 | **28.33±6.83** | 1.00±0.00 |
| ANM | NTS-NOTEARS*-100 | **0.70±0.07** | 48.17±13.18 | **1.00±0.00** | **0.81±0.04** | 28.67±6.22 | **1.00±0.00** |
| AIM | NTSNOTEARS | 0.60±0.04 | 49.17±5.12 | 0.61±0.14 | 0.85±0.04 | 20.00±6.29 | 0.77±0.06 |
| AIM | NTS-NOTEARS*-20 | 0.64±0.02 | 47.33±4.68 | 1.00±0.00 | 0.87±0.04 | 17.67±5.96 | 1.00±0.00 |
| AIM | NTS-NOTEARS*-40 | 0.67±0.02 | 46.00±6.45 | 1.00±0.00 | 0.88±0.03 | 17.67±5.61 | 1.00±0.00 |
| AIM | NTS-NOTEARS*-60 | 0.70±0.05 | 44.83±10.17 | 1.00±0.00 | 0.88±0.05 | 18.33±8.31 | 1.00±0.00 |
| AIM | NTS-NOTEARS*-80 | 0.73±0.04 | 42.50±8.96 | 1.00±0.00 | 0.88±0.04 | 18.17±7.25 | 1.00±0.00 |
| AIM | NTS-NOTEARS*-100 | **0.75±0.04** | **39.67±9.24** | **1.00±0.00** | **0.89±0.04** | **16.50±7.29** | **1.00±0.00** |

### E.3 EVALUATION ON NONLINEAR DATA

This section evaluates our method in nonlinear time-series settings using synthetic data and the NTS-NOTEARS backbone. We first describe the data generation process for two nonlinear structural equation models (SEMs), then present experimental results under varying lag-agnostic priors and finally the analysis of the observations.

**Synthetic Nonlinear Data**  We construct two types of nonlinear time-series datasets to test the robustness of our approach. In both settings, each variable is generated as a nonlinear function of its lagged parents, plus additive noise. The Additive Noise Model (ANM) generates each sample via

$$x = \sigma(XW)W + z,$$

where $\sigma(\cdot)$ denotes the element-wise sigmoid function and $z$ is Gaussian noise. The Additive Index Model (AIM) introduces more complex nonlinearity:

$$x = \tanh(XW) + \cos(XW) + \sin(XW) + z.$$

**Experimental Setting and Results**  We apply the logic-dual formulation (NTS-NOTEARS*) with data-driven initialization (Init Data) under both nonlinear models. For each setting, we vary the percentage of available lag-agnostic edge priors and measure performance using SHD and F1 score on ER4 graph with 3 maximum lags and 20 nodes. The results are reported in Table 3.

**Analysis**  As the proportion of prior edges increases, we observe a steady improvement in F1 score, indicating more accurate recovery of true dependencies. SHD exhibits mild variability across settings but generally remains low, suggesting that performance gains are not achieved by overfitting or

inflating edge counts. These results demonstrate that our method effectively leverages coarse-grained structural knowledge to improve structure learning even in nonlinear time-series settings, confirming its general applicability beyond the linear case.

Table 4: Results of DYNOTEARS with absence and presence constraints of lag-agnostic edges.

| ER2 | | 250 Samples | | | 1000 Samples | | |
|---|---|---|---|---|---|---|---|
| Metric | Method | Node=20 | Node=30 | Node=50 | Node=20 | Node=30 | Node=50 |
| SHD ↓ | Baseline | 12.67±4.03 | 22.33±2.25 | 37.67±2.25 | 10.50±3.02 | 15.33±2.94 | 19.67±1.75 |
| | DYNOTEARS+Absence | 9.83±3.13 | 16.50±3.08 | **24.33±2.58** | 9.83±2.79 | 14.17±1.83 | 19.50±1.76 |
| | DYNOTEARS& (Init 0) | 11.17±3.43 | 18.00±4.20 | **24.33±6.68** | 7.17±2.04 | 6.33±3.27 | 7.33±1.21 |
| | DYNOTEARS& (Init Data) | 9.33±2.42 | 16.00±3.58 | 27.33±2.50 | **4.33±1.03** | **5.33±1.86** | **5.67±2.42** |
| | DYNOTEARS* (Init 0) | 10.00±3.10 | 15.00±2.00 | 24.83±3.54 | 6.50±2.59 | **5.33±1.63** | 7.00±1.10 |
| | DYNOTEARS* (Init Data) | **9.17±2.14** | **14.00±3.79** | 25.50±2.43 | 5.50±2.07 | 6.00±1.41 | 7.33±2.58 |
| F1 ↑ | Baseline | 0.83±0.06 | 0.80±0.02 | 0.80±0.01 | 0.85±0.05 | 0.85±0.03 | 0.89±0.01 |
| | DYNOTEARS+Absence | 0.86±0.05 | 0.85±0.03 | 0.87±0.02 | 0.86±0.05 | 0.87±0.02 | 0.89±0.01 |
| | DYNOTEARS& (Init 0) | 0.86±0.05 | 0.85±0.03 | 0.88±0.03 | 0.90±0.03 | 0.94±0.03 | 0.96±0.01 |
| | DYNOTEARS& (Init Data) | 0.88±0.03 | 0.87±0.03 | 0.86±0.01 | **0.94±0.01** | **0.95±0.02** | **0.97±0.01** |
| | DYNOTEARS* (Init 0) | 0.87±0.04 | 0.87±0.02 | **0.88±0.02** | 0.91±0.04 | 0.95±0.01 | 0.96±0.01 |
| | DYNOTEARS* (Init Data) | **0.88±0.03** | **0.88±0.03** | 0.87±0.01 | 0.93±0.03 | 0.95±0.01 | 0.96±0.01 |

| ER4 | | 250 Samples | | | 1000 Samples | | |
|---|---|---|---|---|---|---|---|
| Metric | Method | Node=20 | Node=30 | Node=50 | Node=20 | Node=30 | Node=50 |
| SHD ↓ | Baseline | 17.00±2.68 | 31.17±2.86 | 57.00±6.90 | 13.00±3.10 | 22.67±5.39 | 33.33±4.27 |
| | DYNOTEARS+Absence | 14.00±3.22 | 24.00±3.16 | 40.33±4.68 | 12.50±3.02 | 22.33±5.05 | 32.33±4.63 |
| | DYNOTEARS& (Init 0) | 17.33±4.76 | 27.00±6.16 | 46.83±10.01 | 6.83±0.98 | 10.33±2.16 | 14.83±2.64 |
| | DYNOTEARS& (Init Data) | 11.83±3.19 | 21.83±1.94 | 40.50±11.64 | **6.33±1.86** | **6.83±2.93** | **11.50±2.07** |
| | DYNOTEARS* (Init 0) | 13.50±2.81 | 22.50±4.09 | 40.50±7.64 | 6.67±2.16 | 10.17±3.66 | 14.17±1.33 |
| | DYNOTEARS* (Init Data) | **11.50±4.51** | **20.00±2.00** | **38.83±10.78** | 6.50±1.87 | 7.33±2.50 | 13.33±3.44 |
| F1 ↑ | Baseline | 0.88±0.02 | 0.86±0.01 | 0.84±0.02 | 0.91±0.02 | 0.90±0.03 | 0.91±0.01 |
| | DYNOTEARS+Absence | 0.91±0.02 | 0.89±0.02 | 0.89±0.01 | 0.91±0.02 | 0.90±0.03 | 0.91±0.01 |
| | DYNOTEARS& (Init 0) | 0.89±0.03 | 0.89±0.03 | 0.88±0.03 | 0.96±0.01 | 0.96±0.01 | 0.96±0.01 |
| | DYNOTEARS& (Init Data) | 0.92±0.02 | 0.91±0.01 | 0.90±0.03 | **0.96±0.01** | **0.97±0.01** | **0.97±0.01** |
| | DYNOTEARS* (Init 0) | 0.91±0.02 | 0.90±0.02 | 0.90±0.02 | 0.96±0.01 | 0.96±0.02 | 0.96±0.00 |
| | DYNOTEARS* (Init Data) | **0.93±0.03** | **0.91±0.01** | **0.90±0.03** | 0.96±0.01 | 0.97±0.01 | 0.97±0.01 |

### E.4 EVALUATION OF LAG-AGNOSTIC EDGE ABSENCE CONSTRAINTS

This experiment evaluates the impact of incorporating lag-agnostic edge absence constraints on structure learning from time-series data. Specifically, we enforce the absence of a lag-agnostic edge $(i, j)$ by constraining the corresponding lagged edge weights to zero across all lags:

$$\forall \tau \in \{0, 1, \ldots, L\}, \quad (W_\tau)_{ij} = 0.$$

Given a binary mask $\mathcal{C}_a \in \{0, 1\}^{d \times d}$, where $(\mathcal{C}_a)_{ij} = 1$ denotes the absence of the lag-agnostic edge $(i, j)$, the optimization problem becomes:

$$\min_{W_{0:L}} \mathcal{L}(X; W_{0:L}), \quad \text{subject to } h(W_0) = 0, \quad \sum_\tau \mathcal{C}_a \circ W_\tau = 0.$$

We use DYNOTEARS as the backbone solver and denote the model with lag-agnostic absence constraints as DYNOTEARS+Absence.

In our setup, we randomly select 80% of absent lag-agnostic edges as priors. We compare three configurations: (1) DYNOTEARS (data-only baseline), (2) DYNOTEARS+Absence, and (3) DYNOTEARS with 80% lag-agnostic edge presence priors. Results under varying node counts, sample sizes, and graph densities (with Gaussian noise) are summarized in Table 4.

**Results and Analysis**   We observe that incorporating lag-agnostic edge absence priors improves structure learning performance over the data-only baseline. However, the degree of improvement is smaller than that achieved by incorporating lag-agnostic edge presence constraints at the same proportion. This difference arises because presence constraints actively guide the model toward recovering missing true edges, enhancing both recall and structural accuracy. In addition, by encouraging the inclusion of informative edges, presence priors can help remove erroneous edges through the regularization, further improving model performance. In contrast, absence constraints merely restrict spurious connections and do not directly assist in identifying the correct ones. As a result, their influence on structure recovery is more passive and less impactful.

These findings highlight that while absence priors are useful for improving precision and suppressing false positives, presence priors contribute more significantly to overall structure recovery, especially in terms of recall and correct identification of lagged dependencies. This underscores the practical importance of integrating lag-agnostic edge presence information when available.

### E.5    EVALUATION UNDER VARYING PRIOR RATES

This section studies how the proportion of available lag-agnostic priors influences the structure learning performance of our method. We systematically vary the prior rate and evaluate its effect on several metrics, including F1 score, Structural Hamming Distance (SHD), recall (TPR), accuracy (1–FDR), and the recovery rate of the prior edges.

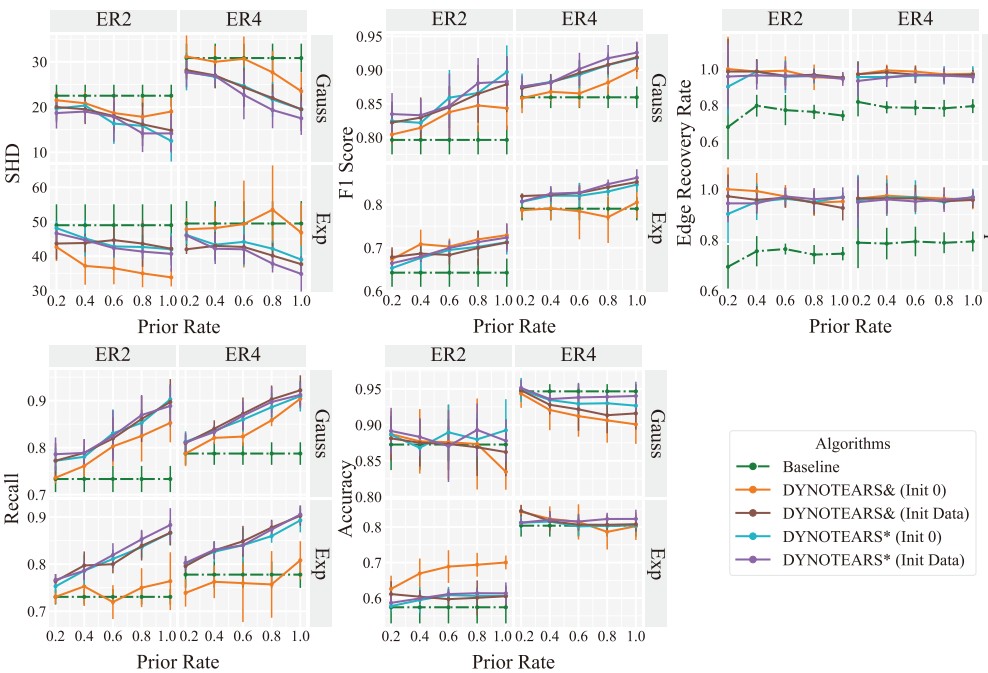

Figure 5: Results under varying prior rates on 250 samples.

**Experimental Setup**   We use synthetic time-series data generated from ER-2 and ER-4 graphs with 30 nodes, under both Gaussian and Exponential noise settings. For each configuration, we compare DYNOTEARS, DYNOTEARS* (logic-dual), and DYNOTEARS& (binary-masked), each evaluated with two initialization strategies: zero initialization (Init 0) and data-driven initialization (Init Data). The lag-agnostic prior rate is varied from low to high values, and performance results are summarized in Figure 5 for 250 samples and Figure 6 for 1000 samples.

**Results and Observations**   As the prior rate increases, we observe consistent improvements in F1 score, SHD, and recall across all methods and settings. This confirms that incorporating more prior knowledge improves the model's ability to recover the true causal structure. Notably, accuracy (1–FDR) for some Init 0 methods exhibits a slight decline in certain cases, likely due to unstable

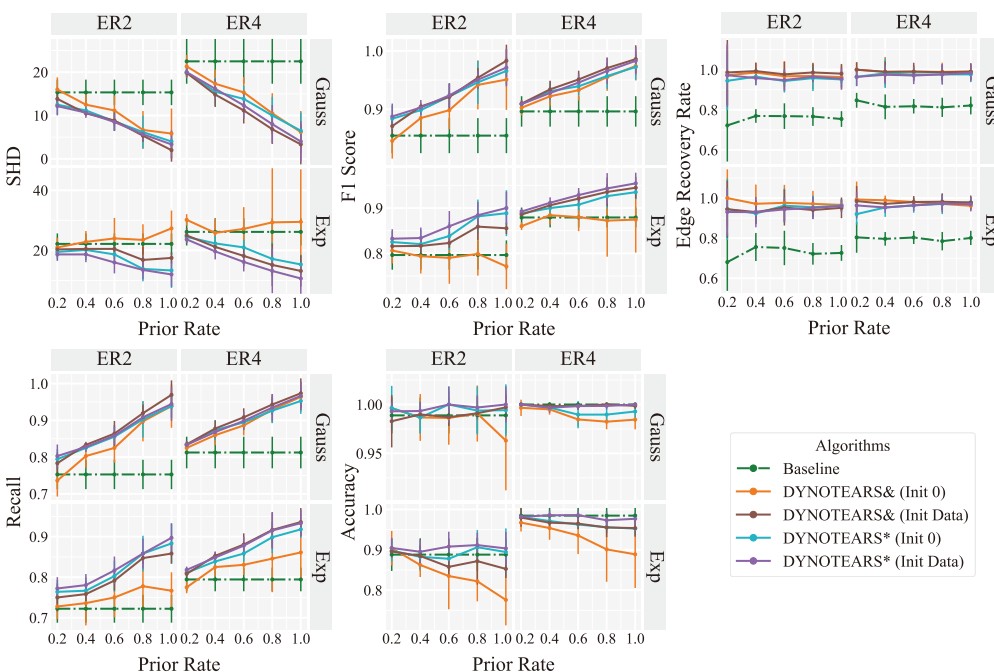

Figure 6: Results under varying prior rates on 1000 samples.

convergence or early overfitting to incorrect lag selections. In contrast, Init Data methods maintain higher and more stable accuracy, demonstrating the benefit of principled initialization under ambiguous prior constraints. These findings highlight the robustness of our approach and reinforce the importance of both informative priors and strong initialization in achieving reliable structure recovery from time-series data.

### E.6 EVALUATION WITH VARYING MAXIMUM LAG NUMBERS

To investigate the influence of the maximum allowable lag $L$ our method, we generate synthetic time-series data with true lag-specific structures ranging from $L = 1$ to $L = 10$. We evaluate the methods, DYNOTEARS, DYNOTEARS*, and DYNOTEARS&, with both Init 0 and Init Data strategies, using metrics: SHD, F1 score, recall (TPR), accuracy (1–FDR), and edge recovery rate.

**Experimental Setup** Experiments are conducted on synthetic datasets with 30 nodes and two different sample sizes: 250 and 1000. The results on 250 samples are presented in Figure 7, while those on 1000 samples are shown in Figure 8. These experiments assess the robustness of each method under increasing temporal complexity.

**Results and Analysis** On datasets with 250 samples, performance generally decreases as the maximum lag increases. This is expected, as longer lags increase the effective model complexity and introduce more potential for overfitting or misidentification of spurious lagged dependencies, especially under limited data. In contrast, when 1000 samples are available, performance improves as $L$ increases, suggesting that the additional temporal depth can be beneficial when supported by sufficient data. Among the methods, DYNOTEARS* (both Init 0 and Init Data) and DYNOTEARS& (Init Data) consistently outperform the data-only baseline across lag values. However, DYNOTEARS& with Init 0 underperforms relative to the baseline when $L > 5$. We attribute this to the increasing non-convexity of the optimization landscape as the number of lag candidates grows, which introduces more local optima. The non-differentiability of the binary-masked loss exacerbates this issue when the model is initialized poorly, leading to unstable optimization. These results underscore the importance of both smooth loss design and principled initialization for scaling lag-agnostic priors to longer temporal horizons.

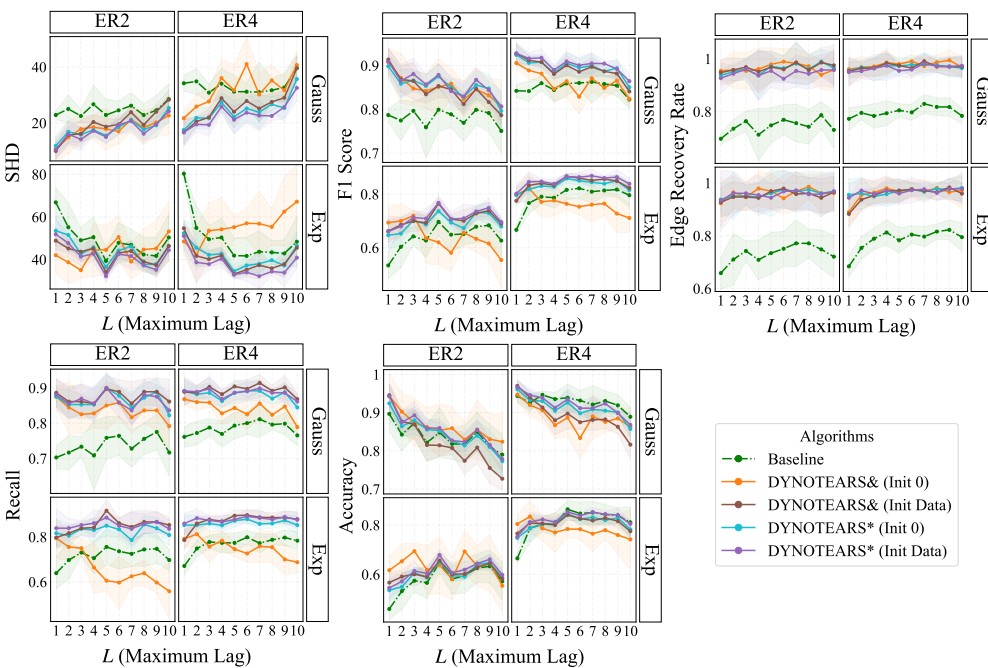

Figure 7: Results under varying maximum lags on 250 samples

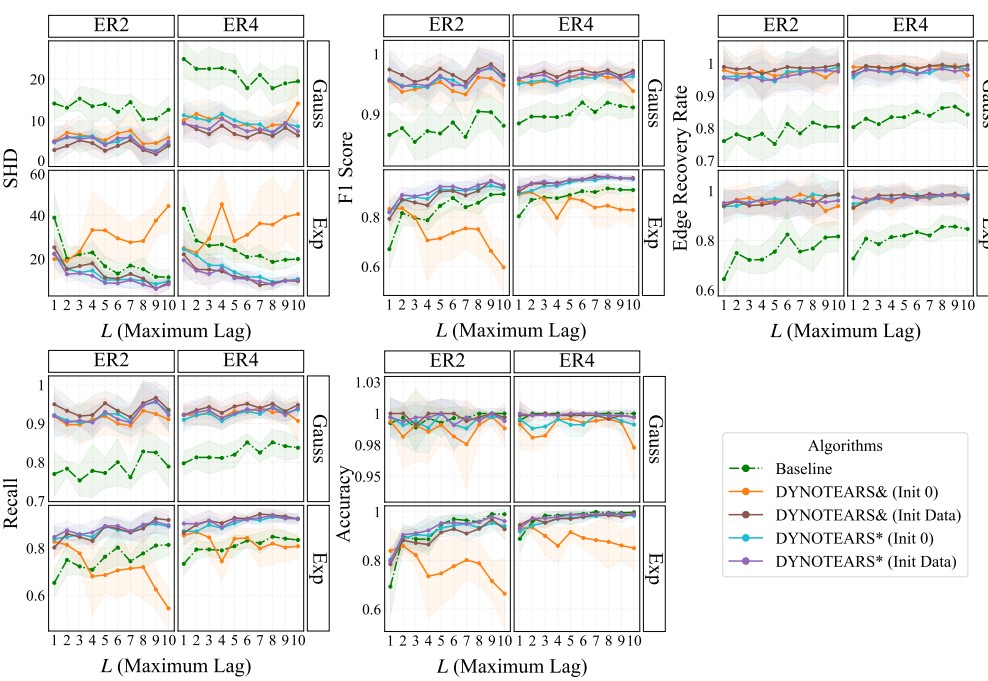

Figure 8: Results under varying maximum lags on 1000 samples.

## E.7    EVALUATION WITH VARYING PRIOR LOSS WEIGHTS

To analyze the sensitivity of our method to the strength of the lag-agnostic prior constraint, we conduct experiments by varying the weight $\lambda_p$ of the prior loss term from 0.01 to 1.0. We evaluate all five model variants, DYNOTEARS, DYNOTEARS*, and DYNOTEARS&, with both Init 0 and Init

Data strategies, using five evaluation metrics: SHD, F1 score, recall (TPR), accuracy (1–FDR), and edge recovery rate.

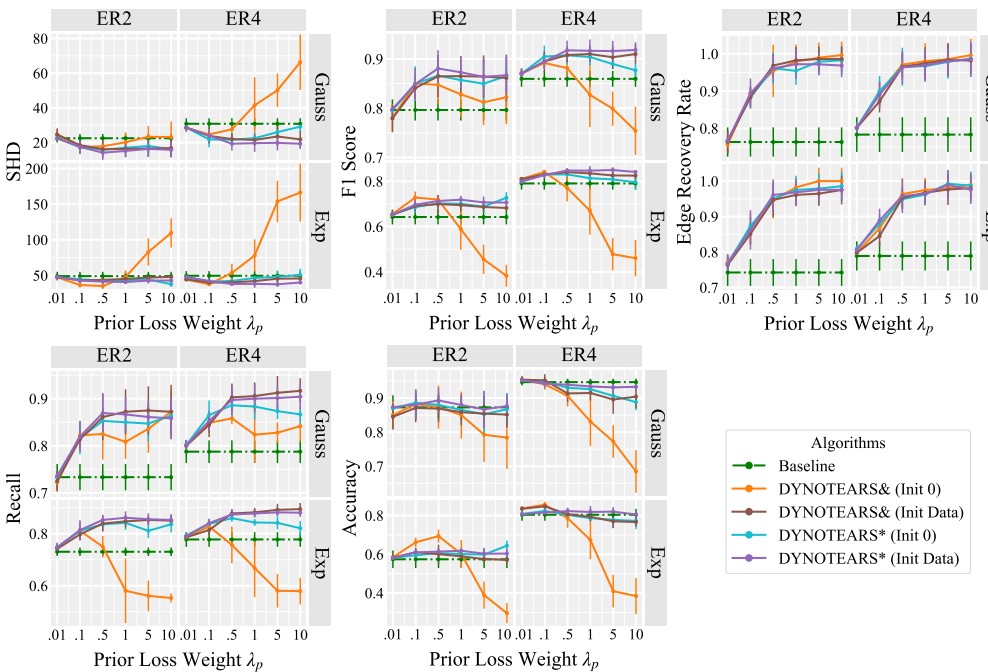

Figure 9: Results under varying prior loss weights on 250 samples.

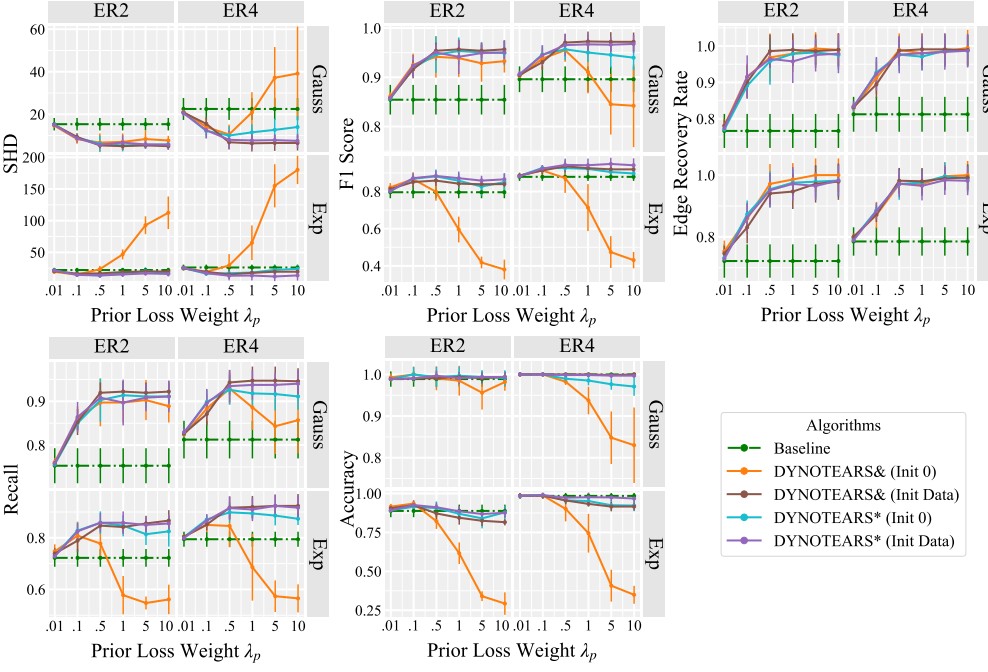

Figure 10: Results under varying prior loss weights on 1000 samples.

**Experimental Setup** Experiments are conducted on synthetic time-series datasets with 30 nodes under two sample size conditions: 250 and 1000 samples. For each value of $\lambda_p$, we apply the lag-agnostic constraint to the structure learning objective and compare performance across methods.

Results on 250 samples are reported in Figure 9, and results on 1000 samples are reported in Figure 10.

**Results and Analysis**   We observe a clear difference in behavior between the two initialization strategies. For methods with Init 0, performance tends to improve initially as $\lambda_p$ increases, but then deteriorates when the loss weight becomes too large. This pattern aligns with our theoretical understanding: as the influence of the prior grows, the optimization becomes increasingly dominated by the ambiguous lag-agnostic constraint, which can distort the loss surface and lead to premature convergence to poor local optima, especially in the absence of a guiding initialization.

In contrast, the Init Data variants show a more stable trend. Performance improves with increasing $\lambda_p$ and then plateaus at a high level, indicating that data-driven initialization effectively mitigates the non-convexity introduced by the lag-agnostic formulation. These results reinforce the importance of principled initialization, especially when prior constraints are strongly enforced.

This experiment confirms that the stability of optimizing the problem with lag-agnostic prior constraints can be significantly improved with informed initialization, maintaining good performance even with large prior strengths.

### E.8   EVALUATION UNDER LARGER MAXIMUM LAGS THAN GROUND TRUTH

This experiment investigates how our method behaves when the model's assumed maximum lag $L_{\text{model}}$ is larger than the true maximum lag $L_{\text{true}}$, as we may not know the specific maximum lag in practice. Specifically, we set $L_{\text{true}} = 5$ and set the used model with $L_{\text{model}} = 10$. The underlying graph is an ER-2 network with 30 nodes, and Gaussian noise is used to simulate the time-series data.

**Metrics and Evaluation**   We report two key evaluation metrics: (1) the data-fitting loss (including regularization) computed using the recovered parameters up to lag $k$ (i.e., lag-0 to lag-$k$), and (2) the maximum absolute edge weight within each lag-specific matrix $W_k$, which reflects the relative contribution of each lag to the fitted model.

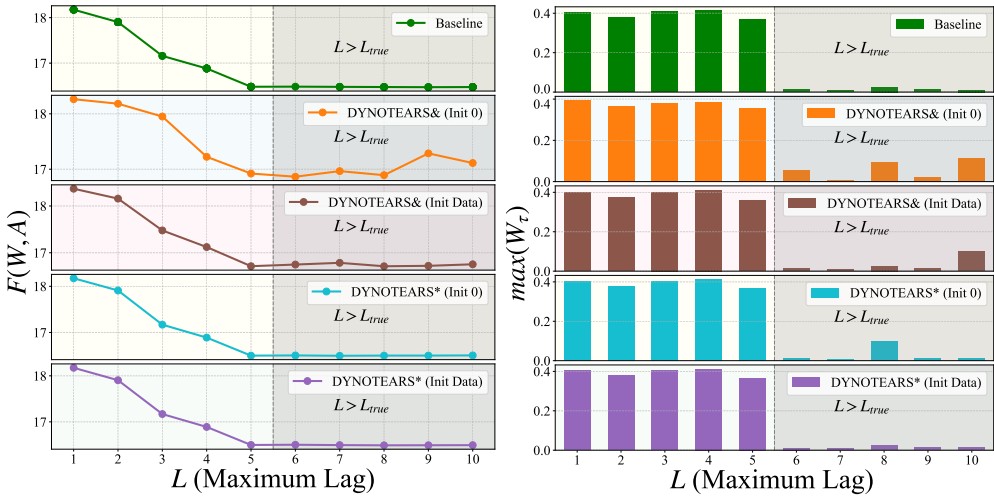

Figure 11: Results under larger maximum lags than ground truth.

**Results and Observations**   Figure 11 presents the results for all five methods. We observe that the data-fitting loss decreases as more lags are included up to the true maximum lag $L_{\text{true}} = 5$, and then remains relatively stable as $k$ increases beyond this point. This indicates that the methods correctly prioritize learning causal dependencies within the correct temporal horizon and do not overfit to spurious long-lag dependencies.

In terms of the maximum edge weight per lag, all methods identify significant causal strengths for lags 0 through 5, while edge magnitudes beyond lag 5 drop sharply and remain close to zero. This

behavior suggests that even when the model is allowed to consider more lags than necessary, it can accurately isolate the true temporal dependencies and avoid overestimating long-lag effects.

Among the methods, DYNOTEARS* shows more stable performance than DYNOTEARS&, particularly when $k > L_{true}$. This is likely due to its fully differentiable loss design, which enables smoother optimization across lagged parameters. Additionally, Init Data consistently outperforms Init 0 for both loss functions, resulting in more stable data fit and sparser edge recovery in the extra lag dimensions. Specifically, the better-performing methods exhibit lower fitting loss and maintain near-zero maximum edge weights in higher lag levels, confirming their ability to resist overfitting when the model includes excessive lags.

### E.9 VISUALIZATION OF LAG-SPECIFIC EDGE WEIGHTS

Figure 12: Visualized results of the lag-specific edge weights.

To qualitatively assess the effectiveness of incorporating lag-agnostic priors, we visualize the recovered lag-specific edge weights in a representative example. We consider a synthetic dataset with 5 nodes and a maximum lag of 3. The true lagged structures, the structure learned by the baseline method (DYNOTEARS, using data only), and the structure learned by our lag-agnostic prior-based method (DYNOTEARS*) are visualized and compared. The results are shown in Figure 12.

**Results and Observations**   From the visualization, we observe that the model incorporating lag-agnostic priors (DYNOTEARS*) more accurately recovers both the *locations* and *magnitudes* of the lag-specific edges compared to the data-only baseline. Many true causal connections that are missed by the baseline are correctly recovered under the guidance of lag-agnostic priors. Importantly, the recovered weights not only reflect the correct lags but also closely match the ground-truth edge strengths, demonstrating the precision of the proposed method.

These results visually support our quantitative findings, showing that even coarse-grained prior knowledge can lead to more accurate and interpretable recovery of fine-grained, lag-specific temporal structures.

### E.10 EVALUATION OF PRIOR ROBUSTNESS

We assess the practical robustness of integrating lag-agnostic priors when the provided knowledge is imperfect or noisy.

### E.10.1 ROBUSTNESS TO IMPERFECT EDGE PRESENCE

We evaluated our method's performance (DYNOTEARS* with Init Data) on a 20-node ER4 graph (Max Lag $L = 5$, Gaussian noise) while increasing the percentage of **incorrect** edge presence priors ($q$). The results demonstrate that our framework is highly robust due to its soft-constraint design.

Table 5: Robustness to Imperfect Lag-Agnostic Edge Presence Priors (20-node ER4, L=5). ($\checkmark$) denotes outperformance over baseline.

| Method | Error Rate ($q$) | Performance | | Correct Priors | | Wrong Priors | |
|---|---|---|---|---|---|---|---|
| | | SHD $\downarrow$ | F1 $\uparrow$ | Num. | Recov. (%) | Num. | Recov. (%) |
| Baseline | - | 15.0 | 0.897 | - | - | - | - |
| Ours | 0% $\checkmark$ | **7.2 $\pm$ 1.6** | **0.953 $\pm$ 0.011** | 57 | 96.20% $\pm$ 2.05% | 0 | - |
| | 5% $\checkmark$ | **7.8 $\pm$ 1.5** | **0.949 $\pm$ 0.010** | 54 | 95.99% $\pm$ 2.73% | 3 | 5.56% $\pm$ 13.61% |
| | 10% $\checkmark$ | **9.0 $\pm$ 2.6** | **0.941 $\pm$ 0.017** | 51 | 96.41% $\pm$ 2.29% | 6 | 11.11% $\pm$ 8.61% |
| | 30% $\checkmark$ | **12.8 $\pm$ 2.2** | **0.916 $\pm$ 0.016** | 40 | 97.50% $\pm$ 1.58% | 17 | 16.67% $\pm$ 4.43% |
| | 50% $\times$ | 15.833 $\pm$ 3.9 | 0.898 $\pm$ 0.023 | 29 | 98.85% $\pm$ 1.78% | 28 | 19.05% $\pm$ 11.66% |

The results show our method ($\checkmark$) **outperforms the baseline with up to 30% prior error**. This robustness stems from successfully **leveraging "good" information** (recovering 96–99% of correct priors) while actively **rejecting "bad" information** (e.g., rejecting $> 83\%$ of wrong priors at the 30% error level).

### E.10.2 SENSITIVITY TO INCORRECT EDGE ABSENCE

We also evaluated robustness to imperfect edge **absence** priors. We found that while **correct** absence priors are helpful (SHD improves from 15.0 to 13.5), the process is **highly sensitive to incorrect** (false absence) priors. Practitioners must be highly confident in absence priors, as enforcing a spurious absence constraint is extremely disruptive and leads to rapid performance degradation (Table 6).

Table 6: Sensitivity to Incorrect Lag-Agnostic Edge Absence Priors (20-node ER4, L=5).

| Algorithm | Num. Correct Priors | Num. Wrong Priors | SHD $\downarrow$ | F1-Score $\uparrow$ |
|---|---|---|---|---|
| DYNOTEARS (Baseline) | - | - | 15.0 | 0.897 |
| DYNOTEARS-Absence | 163 | 0 | 13.5 $\pm$ 0.5 | 0.908 $\pm$ 0.004 |
| DYNOTEARS-Absence | 155 | 8 | 22.2 $\pm$ 3.0 | 0.843 $\pm$ 0.022 |
| DYNOTEARS-Absence | 147 | 16 | 37.7 $\pm$ 6.1 | 0.729 $\pm$ 0.042 |
| DYNOTEARS-Absence | 114 | 49 | 83.8 $\pm$ 5.4 | 0.325 $\pm$ 0.042 |
| DYNOTEARS-Absence | 81 | 74 | 111.7 $\pm$ 1.9 | 0.000 $\pm$ 0.000 |

### E.11 INTEGRATION WITH GROUP SPARSITY BASELINES

The effectiveness of our framework is independent of the base regularization strategy. To validate this, we conducted an experiment integrating our lag-agnostic prior into the Group Sparsity (GS) baseline, which is a variant of DYNOTEARS that applies sparsity regularization at the group (lag-agnostic) level rather than the individual edge level.

We replaced the standard DYNOTEARS backbone with the GS variant and applied both the 'oracle' lag-specific (LS) priors and our lag-agnostic (DYNOTEARS*) priors on top. We tested on a 20-node ER4 graph ($L = 5$, Gaussian noise).

The results in Table 7 demonstrate the substantial value of integrating our proposed prior.

Observation: Our lag-agnostic prior provides a massive structural improvement to the GS baseline, reducing the SHD from 21.19 (data-only) to 14.23 (with 50% priors). Furthermore, the performance of our lag-agnostic method nearly matches the oracle, perfect-lag-information baseline (LS-0), demonstrating that our method effectively complements existing regularization strategies.

Table 7: Performance on Group Sparsity (GS) Baseline (20-node ER4, $L = 5$, Gaussian).

| Method | 5% Priors | | 10% Priors | | 30% Priors | | 50% Priors | |
|---|---|---|---|---|---|---|---|---|
| | SHD↓ | F1↑ | SHD↓ | F1↑ | SHD↓ | F1↑ | SHD↓ | F1↑ |
| DYNOTEARS (Group) | $21.19 \pm 2.55$ | $0.80 \pm 0.02$ | $21.19 \pm 2.55$ | $0.80 \pm 0.02$ | $21.19 \pm 2.55$ | $0.80 \pm 0.02$ | $21.19 \pm 2.55$ | $0.80 \pm 0.02$ |
| LS-0, 0% error (Group) | **$19.04 \pm 2.71$** | **$0.82 \pm 0.02$** | **$16.98 \pm 2.45$** | **$0.85 \pm 0.01$** | **$13.67 \pm 1.98$** | **$0.88 \pm 0.01$** | **$13.29 \pm 2.15$** | **$0.89 \pm 0.01$** |
| LS-10, 10% error (Group) | $20.10 \pm 3.11$ | $0.81 \pm 0.03$ | $18.04 \pm 2.63$ | $0.84 \pm 0.02$ | $16.04 \pm 2.22$ | $0.86 \pm 0.02$ | $15.33 \pm 2.04$ | $0.87 \pm 0.01$ |
| LS-30, 30% error (Group) | $20.10 \pm 2.95$ | $0.81 \pm 0.03$ | $18.56 \pm 2.81$ | $0.83 \pm 0.02$ | $16.69 \pm 2.40$ | $0.85 \pm 0.02$ | $16.33 \pm 2.31$ | $0.86 \pm 0.02$ |
| LS-50, 50% error (Group) | $20.10 \pm 3.05$ | $0.81 \pm 0.03$ | $18.56 \pm 2.76$ | $0.83 \pm 0.02$ | $18.48 \pm 2.58$ | $0.83 \pm 0.02$ | $19.69 \pm 2.99$ | $0.82 \pm 0.03$ |
| **Ours** | $19.21 \pm 2.66$ | $0.82 \pm 0.02$ | $17.25 \pm 2.18$ | $0.84 \pm 0.01$ | $14.29 \pm 1.95$ | $0.88 \pm 0.01$ | $14.23 \pm 2.05$ | $0.88 \pm 0.01$ |

Table 8: Comparison of Process-Equivalent Formulations (Softmax vs. Proposed). Data: 20-node ER4, $L = 5$, Gaussian Noise.

| Method | 5% Priors | | 10% Priors | | 30% Priors | | 50% Priors | |
|---|---|---|---|---|---|---|---|---|
| | SHD↓ | F1↑ | SHD↓ | F1↑ | SHD↓ | F1↑ | SHD↓ | F1↑ |
| DYNOTEARS | $13.00\pm3.10$ | $0.91\pm0.02$ | $13.00\pm3.10$ | $0.91\pm0.02$ | $13.00\pm3.10$ | $0.91\pm0.02$ | $13.00\pm3.10$ | $0.91\pm0.02$ |
| LS-0, Perfect Lags | **$10.33\pm2.73$** | **$0.93\pm0.02$** | **$8.00\pm2.37$** | **$0.95\pm0.02$** | **$4.83\pm1.94$** | **$0.97\pm0.01$** | **$4.83\pm1.33$** | **$0.97\pm0.01$** |
| LS-10, 10% Error | $11.67\pm3.44$ | $0.92\pm0.02$ | $8.17\pm2.64$ | $0.95\pm0.02$ | $6.67\pm2.58$ | $0.96\pm0.02$ | $6.50\pm2.43$ | $0.96\pm0.02$ |
| LS-30, 30% Error | $12.50\pm3.62$ | $0.92\pm0.03$ | $9.50\pm3.27$ | $0.94\pm0.02$ | $8.83\pm1.33$ | $0.94\pm0.01$ | $8.33\pm2.16$ | $0.95\pm0.02$ |
| LS-50, 50% Error | $12.50\pm3.62$ | $0.92\pm0.03$ | $10.67\pm2.73$ | $0.93\pm0.02$ | $11.17\pm1.47$ | $0.93\pm0.01$ | $11.83\pm1.94$ | $0.92\pm0.01$ |
| DYNOTEARSˆ (Flawed, Init Data) | $11.50 \pm 2.26$ | $0.92 \pm 0.02$ | $9.33 \pm 2.34$ | $0.94 \pm 0.02$ | $7.83 \pm 2.04$ | $0.94 \pm 0.02$ | $6.83 \pm 1.47$ | $0.95 \pm 0.01$ |
| **DYNOTEARS-softmax** (Init Data) | $10.83 \pm 2.88$ | $0.93 \pm 0.02$ | **$8.17 \pm 2.32$** | **$0.95 \pm 0.02$** | $6.33 \pm 1.94$ | $0.96 \pm 0.01$ | $5.67 \pm 2.16$ | $0.96 \pm 0.01$ |
| DYNOTEARS& (Init Data) | $10.83 \pm 1.97$ | $0.93 \pm 0.01$ | $8.50 \pm 3.78$ | $0.94 \pm 0.03$ | $5.83 \pm 1.97$ | **$0.97 \pm 0.01$** | $6.17 \pm 2.71$ | $0.95 \pm 0.02$ |
| **Ours (DYNOTEARS*)** (Init Data) | $10.50 \pm 2.35$ | $0.93 \pm 0.02$ | **$8.00 \pm 2.37$** | **$0.95 \pm 0.02$** | $5.33 \pm 1.86$ | **$0.97 \pm 0.01$** | $5.50 \pm 1.76$ | $0.96 \pm 0.01$ |

### E.12 COMPARISON WITH SOFTMAX-WEIGHTED PRIOR FORMULATION

The continuous optimization approach for structure learning naturally invites comparisons with alternative differentiable surrogates. To fully explore the landscape of process-equivalent formulations, we implemented a new baseline, DYNOTEARS-softmax, to use a softmax-weighted sum over lags. This approach can be viewed as a weighted variant of our binary-masked loss ($p_{\text{bin}}$), where the penalty for unsatisfied edges is distributed according to the relative magnitude of each lagged weight.

The penalty function for the softmax-weighted prior is defined as:

$$p_{\text{softmax}} = \sum \left( \mathcal{C}_p \circ \mathbb{I}(\max_\tau |W_\tau| < \delta) \circ \left( \sum_\tau \left( \text{softmax}_\tau(|W|) \circ \text{ReLU}(\delta - |W_\tau|) \right) \right) \right)$$

where the inner $\text{softmax}_\tau(|W|)$ is calculated element-wise across the lag dimension ($\tau = 0..L$) for a specific edge $(i, j)$. Like our proposed formulations, this loss is also process-equivalent as it addresses all lags simultaneously when the constraint is active.

We compared DYNOTEARS-softmax (Init Data) against our main baselines (Table 8, appended to the Appendix). As expected, the softmax-weighted approach performs significantly better than the flawed max-based formulation (DYNOTEARSˆ), achieving results on par with our proposed process-equivalent formulations (DYNOTEARS& and DYNOTEARS*). This further confirms that resolving the issue of process-inequivalence is the decisive factor for performance gains.

### E.13 EVALUATION OF LAG SCALE SENSITIVITY IN LOGIC-DUAL FORMULATION

The Logic-Dual formulation ($p_{\text{or}}$) is fully continuous but it suffers from theoretical **scale sensitivity** when the maximum lag $L$ is large. Specifically, as the number of factors in the product term increases (i.e., as $L$ grows), the penalty magnitude can decrease, yielding a weaker optimization signal even when the constraint is violated. Our solution to this was the normalized form, $\bar{p}_{\text{or}}$.

To test the practical impact and effectiveness of this normalization under long temporal horizons, we ran an experiment on a small 10-node, ER-4 graph, varying the maximum lag $L$ from 5 up to 50. We compare the performance of the data-only baseline (DYNOTEARS) against our normalized Logic-Dual method ($\bar{p}_{\text{or}}$).

Table 9: Performance Stability of Normalized Logic-Dual Loss ($\overline{p}_{\text{or}}$) under increasing Maximum Lag $L$. Data: 10-node ER4.

| Method | $L = 5$ | | $L = 10$ | | $L = 20$ | |
|---|---|---|---|---|---|---|
| | SHD↓ | F1↑ | SHD↓ | F1↑ | SHD↓ | F1↑ |
| Baseline | $8.00 \pm 1.41$ | $0.89 \pm 0.02$ | $4.00 \pm 0.89$ | $0.95 \pm 0.01$ | $6.00 \pm 1.26$ | $0.92 \pm 0.02$ |
| Ours ($\overline{p}_{\text{or}}$) | $\mathbf{2.83 \pm 0.75}$ | $\mathbf{0.96 \pm 0.01}$ | $\mathbf{1.00 \pm 0.63}$ | $\mathbf{0.99 \pm 0.01}$ | $\mathbf{1.17 \pm 0.75}$ | $\mathbf{0.99 \pm 0.01}$ |
| Method | $L = 30$ | | $L = 40$ | | $L = 50$ | |
| | SHD↓ | F1↑ | SHD↓ | F1↑ | SHD↓ | F1↑ |
| Baseline | $9.00 \pm 1.64$ | $0.88 \pm 0.03$ | $7.00 \pm 1.34$ | $0.90 \pm 0.02$ | $8.00 \pm 1.58$ | $0.89 \pm 0.03$ |
| Ours ($\overline{p}_{\text{or}}$) | $\mathbf{4.50 \pm 0.96}$ | $\mathbf{0.94 \pm 0.01}$ | $\mathbf{3.50 \pm 0.84}$ | $\mathbf{0.95 \pm 0.01}$ | $\mathbf{3.00 \pm 0.63}$ | $\mathbf{0.96 \pm 0.01}$ |

The results confirm the stability provided by the normalization. The product-based prior loss continues to provide strong, consistent performance gains over the baseline across the entire range, from $L = 5$ up to $L = 50$. This indicates that our normalization effectively mitigates the theoretical scale sensitivity issue, ensuring the method's practical utility for analyzing data with large maximum lags.

### E.14 SCALABILITY AND COMPUTATIONAL EFFICIENCY

We performed both a theoretical and empirical analysis to address the scalability and practical runtime of integrating our lag-agnostic prior, especially for large-scale graphs ($d \geq 100$).

**Theoretical Complexity.** The per-iteration complexity for calculating our proposed prior loss (both $p_{\text{bin}}$ and $p_{\text{or}}$) is $\mathcal{O}(Ld^2)$. This cost is highly efficient and is negligible compared to the standard VAR-based data-fit loss, which has a complexity of $\mathcal{O}(TLd^2)$, where $T$ is the number of time steps. Therefore, the prior loss itself does not introduce a computational bottleneck to the per-iteration cost.

**Empirical Scaling.** We tested the overall runtime on large synthetic graphs (ER4, Gaussian noise, $L = 3$, 30% priors) with $d = 100$ and $d = 200$ nodes. The results below compare the runtime of the data-only baseline against our method (DYNOTEARS*).

Table 10: Scalability and Runtime Comparison on Large Graphs ($d \geq 100$). Our method (Ours*) achieves large performance gains despite increased optimization complexity.

| Nodes | Algorithm | SHD ↓ | F1-Score ↑ | Runtime (s) |
|---|---|---|---|---|
| 100 | Baseline (DYNOTEARS) | 95 | 0.815 | 46.43 |
| | **Ours (DYNOTEARS*)** | **65** | **0.894** | 237.89 |
| 200 | Baseline (DYNOTEARS) | 164 | 0.778 | 216.08 |
| | **Ours (DYNOTEARS*)** | **112** | **0.863** | 1364.46 |

Observation: Our method scales well, achieving significant structural gains (SHD reduced by $\approx 35\%$) on large graphs. The observed 5x–6x runtime increase is not from the per-iteration loss calculation, but rather from the optimization process. The combined non-convexity introduced by the DAG constraint and our prior requires the optimizer to take significantly more iterations to converge to a quality solution.

### E.15 ABLATION ON INITIALIZATION PRE-TRAINING

We compared a new, simpler data-based initialization strategy in this experiment: Init VAR (New): Pre-training a simple VAR model by optimizing only the data-fit loss $\mathcal{L}(X; W_{0:L})$, omitting the DAG constraint $h(W_0)$.

The experiment was conducted on a 20-node, ER4 graph ($L = 5$, Gaussian noise), testing both our proposed formulations (DYNOTEARS& and DYNOTEARS*). Results are reported in Table 11.

Table 11: Ablation Study on Initialization Strategies (20-node ER4, $L = 5$).

| Loss Function | Initialization Strategy | SHD ↓ | F1-Score ↑ | Accuracy ↑ | Recall ↑ |
|---|---|---|---|---|---|
| Baseline | (No Prior) | 20.0 | 0.859 | 0.984 | 0.762 |
| DYNOTEARS& (Binary Masked) | Init 0 | 7.50 | 0.951 | 0.985 | 0.921 |
| | Init VAR | 5.50 | 0.964 | **1.000** | 0.931 |
| | **Init Data (Ours)** | **5.17** | **0.967** | **1.000** | **0.935** |
| DYNOTEARS* (Logic Dual) | Init 0 | 6.67 | 0.957 | 0.987 | 0.929 |
| | Init VAR | 6.17 | 0.960 | **1.000** | 0.923 |
| | **Init Data (Ours)** | **5.50** | **0.964** | **1.000** | **0.931** |

The results confirm our design intuition. Init VAR consistently outperforms Init 0, demonstrating that starting from a state with good data-fit (even without the DAG constraint) is superior to a cold start. Crucially, the Init Data strategy is consistently the best performer across both loss formulations (DYNOTEARS& and DYNOTEARS*). This superiority is attributed to its ability to solve the non-convex DAG constraint first, providing a stable, acyclic instantaneous structure ($\mathbf{W_0}$) before introducing the second non-convex prior loss. This effectively avoids the initial gradient conflict and promotes convergence to a better local optimum.

### E.16 SUPPLEMENTARY RESULTS

This section presents supplementary experimental results that extend the findings reported in the main text. These additional results cover a broader range of evaluation metrics, graph configurations, and data sample sizes to provide a more comprehensive view of method performance. The specific settings for each experiment are detailed in the corresponding figure or table captions.

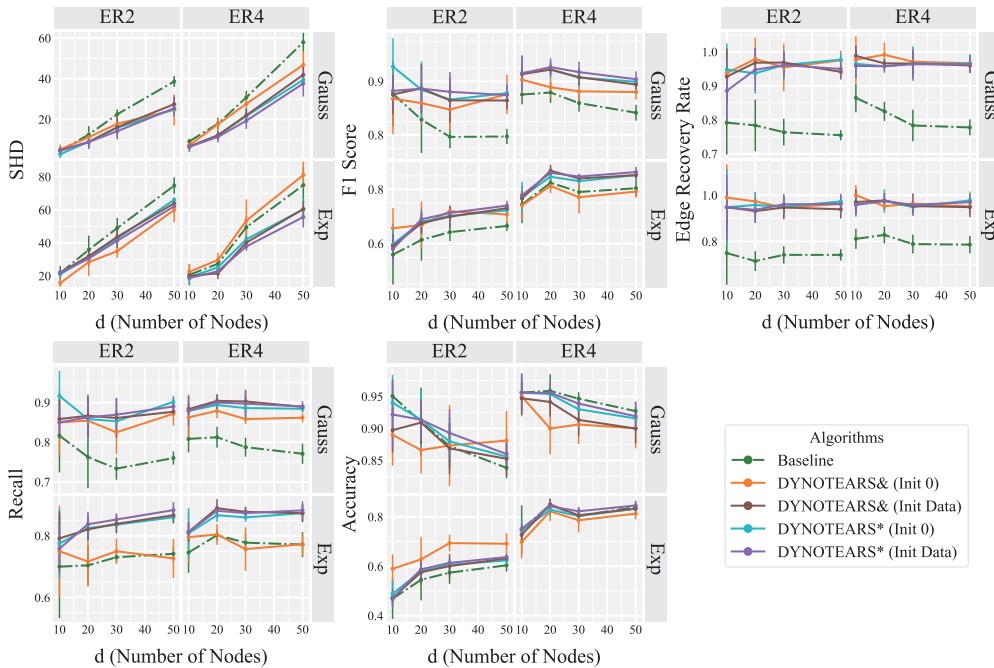

Figure 13: Supplementary results of the overall comparison on 250 samples.

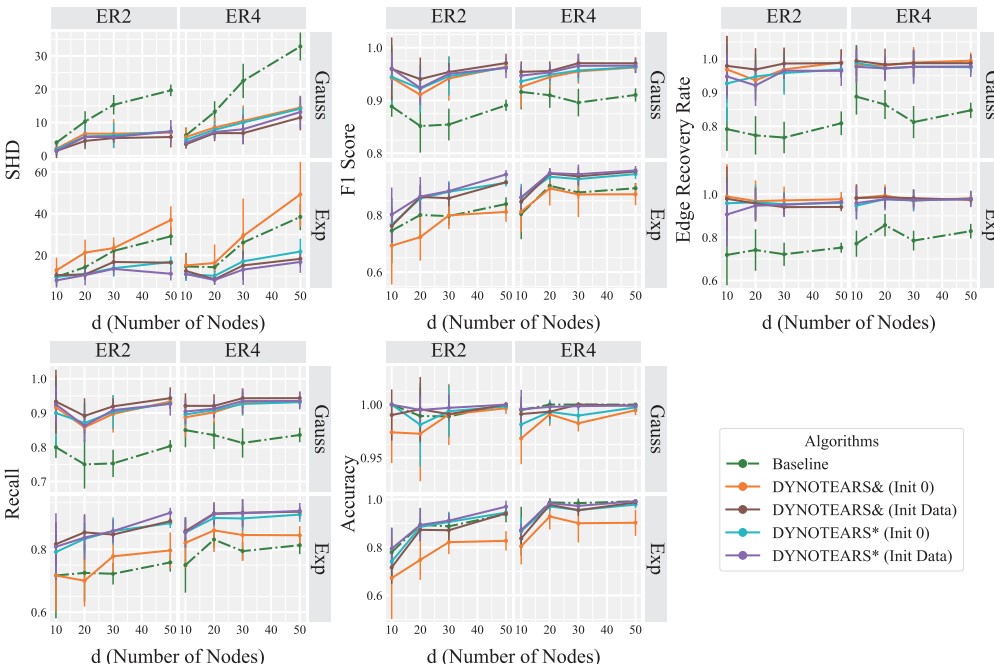

Figure 14: Supplementary results of the overall comparison on 1000 samples.

Table 12: Supplementary results of the overall comparison with lag-specific prior integration on 200 samples.

| GT | NT | Method | 5% Priors | | 10% Priors | | 30% Priors | | 50% Priors | |
|---|---|---|---|---|---|---|---|---|---|---|
| | | | SHD ↓ | F1 ↑ | SHD ↓ | F1 ↑ | SHD ↓ | F1 ↑ | SHD ↓ | F1 ↑ |
| ER-2 | Exp | DYNOTEARS | 40.00±6.63 | 0.58±0.06 | 40.00±6.63 | 0.58±0.06 | 40.00±6.63 | 0.58±0.06 | 40.00±6.63 | 0.58±0.06 |
| | | LS-0, Perfect Lags | **37.17±6.05** | **0.61±0.05** | **36.33±5.68** | **0.62±0.05** | **30.17±5.00** | **0.70±0.04** | **31.17±6.18** | **0.69±0.05** |
| | | LS-10, 10% Error | 39.17±6.24 | 0.60±0.05 | 37.33±5.57 | 0.62±0.05 | 34.17±5.91 | 0.66±0.05 | 33.50±5.01 | 0.67±0.04 |
| | | LS-30, 30% Error | 39.17±6.24 | 0.60±0.05 | 38.33±4.13 | 0.61±0.03 | 35.83±5.49 | 0.64±0.04 | 37.00±5.83 | 0.63±0.04 |
| | | LS-50, 50% Error | 39.17±6.24 | 0.60±0.05 | 38.33±4.13 | 0.61±0.03 | 38.00±4.00 | 0.62±0.03 | 41.17±2.71 | 0.60±0.02 |
| | | Ours (Lag Agnostic) | 37.83±5.46 | 0.61±0.04 | 37.00±4.82 | 0.62±0.04 | 34.17±4.26 | 0.66±0.03 | 33.50±6.25 | 0.66±0.05 |
| ER-2 | Gauss | DYNOTEARS | 17.33±4.84 | 0.77±0.07 | 17.33±4.84 | 0.77±0.07 | 17.33±4.84 | 0.77±0.07 | 17.33±4.84 | 0.77±0.07 |
| | | LS-0, Perfect Lags | **16.50±4.37** | **0.78±0.06** | **14.50±4.64** | **0.81±0.07** | **10.00±3.16** | **0.88±0.04** | **9.17±3.43** | **0.89±0.04** |
| | | LS-10, 10% Error | 17.17±4.26 | 0.77±0.06 | 15.83±4.54 | 0.79±0.06 | 12.33±4.89 | 0.84±0.07 | 12.50±5.36 | 0.85±0.07 |
| | | LS-30, 30% Error | 17.17±4.26 | 0.77±0.06 | 16.33±4.80 | 0.79±0.07 | 13.00±3.52 | 0.84±0.05 | 13.33±3.50 | 0.84±0.04 |
| | | LS-50, 50% Error | 17.17±4.26 | 0.77±0.06 | 16.33±4.80 | 0.79±0.07 | 15.33±4.97 | 0.81±0.06 | 15.17±3.19 | 0.81±0.03 |
| | | Ours (Lag Agnostic) | 16.50±3.99 | 0.78±0.06 | 15.67±5.54 | 0.80±0.08 | 12.33±5.20 | 0.85±0.06 | 9.50±2.26 | 0.88±0.03 |
| ER-4 | Exp | DYNOTEARS | 36.50±5.54 | 0.77±0.03 | 36.50±5.54 | 0.77±0.03 | 36.50±5.54 | 0.77±0.03 | 36.50±5.54 | 0.77±0.03 |
| | | LS-0, Perfect Lags | **32.83±6.21** | **0.79±0.03** | **29.33±7.37** | **0.82±0.04** | **26.00±6.90** | **0.84±0.04** | **25.83±5.20** | **0.84±0.03** |
| | | LS-10, 10% Error | 33.83±6.37 | 0.79±0.04 | 30.00±6.60 | 0.81±0.04 | 28.83±6.62 | 0.83±0.03 | 31.17±6.46 | 0.81±0.04 |
| | | LS-30, 30% Error | 36.00±5.48 | 0.77±0.03 | 32.17±6.97 | 0.80±0.04 | 33.83±6.97 | 0.80±0.04 | 35.33±4.84 | 0.79±0.03 |
| | | LS-50, 50% Error | 36.00±5.48 | 0.77±0.03 | 32.83±7.88 | 0.80±0.04 | 36.33±7.55 | 0.78±0.04 | 41.83±7.03 | 0.76±0.04 |
| | | Ours (Lag Agnostic) | 33.17±7.19 | 0.79±0.04 | 29.33±5.13 | 0.82±0.03 | 26.50±6.86 | 0.84±0.04 | 27.00±4.82 | 0.84±0.03 |
| ER-4 | Gauss | DYNOTEARS | 23.00±4.47 | 0.85±0.03 | 23.00±4.47 | 0.85±0.03 | 23.00±4.47 | 0.85±0.03 | 23.00±4.47 | 0.85±0.03 |
| | | LS-0, Perfect Lags | **20.67±4.18** | **0.86±0.03** | **16.83±4.07** | **0.89±0.03** | **12.00±3.03** | **0.92±0.02** | **11.83±2.99** | **0.93±0.02** |
| | | LS-10, 10% Error | 22.67±4.84 | 0.85±0.03 | 19.67±5.13 | 0.87±0.03 | 17.33±3.93 | 0.89±0.02 | 16.67±4.59 | 0.90±0.03 |
| | | LS-30, 30% Error | 23.17±4.07 | 0.85±0.03 | 19.33±5.16 | 0.87±0.03 | 17.67±2.88 | 0.89±0.02 | 18.83±4.22 | 0.88±0.02 |
| | | LS-50, 50% Error | 23.17±4.07 | 0.85±0.03 | 21.33±5.28 | 0.86±0.03 | 20.67±3.20 | 0.87±0.02 | 23.33±4.46 | 0.85±0.02 |
| | | Ours (Lag Agnostic) | 21.83±4.45 | 0.85±0.03 | 17.33±4.41 | 0.89±0.03 | 13.17±4.26 | 0.92±0.03 | 13.33±3.56 | 0.92±0.02 |

Table 13: Supplementary results of the overall comparison with lag-specific prior integration on 1000 samples.

| GT | NT | Method | 5% Priors | | 10% Priors | | 30% Priors | | 50% Priors | |
|----|----|--------|-----------|--|------------|--|------------|--|------------|--|
| | | | SHD ↓ | F1 ↑ | SHD ↓ | F1 ↑ | SHD ↓ | F1 ↑ | SHD ↓ | F1 ↑ |
| ER-2 | Exp | DYNOTEARS | 14.17±5.91 | 0.80±0.09 | 14.17±5.91 | 0.80±0.09 | 14.17±5.91 | 0.80±0.09 | 14.17±5.91 | 0.80±0.09 |
| | | LS-0, Perfect Lags | **12.17±5.88** | **0.83±0.09** | 11.83±5.12 | 0.84±0.07 | 8.17±5.91 | **0.90±0.08** | 7.67±4.27 | **0.90±0.05** |
| | | LS-10, 10% Error | 13.00±5.87 | 0.82±0.09 | 12.33±5.13 | 0.83±0.07 | 12.00±6.26 | 0.84±0.09 | 11.00±5.06 | 0.86±0.07 |
| | | LS-30, 30% Error | 13.00±5.87 | 0.82±0.09 | 14.17±5.19 | 0.81±0.08 | 13.17±5.64 | 0.83±0.08 | 12.17±5.00 | 0.85±0.06 |
| | | LS-50, 50% Error | 13.00±5.87 | 0.82±0.09 | 14.17±5.19 | 0.81±0.08 | 14.17±4.92 | 0.82±0.07 | 17.00±8.22 | 0.79±0.10 |
| | | Ours (Lag Agnostic) | 12.33±6.09 | 0.83±0.09 | 12.00±5.14 | 0.84±0.07 | 8.33±5.32 | 0.89±0.07 | 9.83±5.04 | 0.87±0.06 |
| ER-2 | Gauss | DYNOTEARS | 10.50±3.02 | 0.85±0.05 | 10.50±3.02 | 0.85±0.05 | 10.50±3.02 | 0.85±0.05 | 10.50±3.02 | 0.85±0.05 |
| | | LS-0, Perfect Lags | **9.00±3.10** | **0.87±0.05** | 7.33±1.97 | 0.90±0.03 | 2.83±2.32 | 0.96±0.03 | 3.50±1.52 | 0.95±0.02 |
| | | LS-10, 10% Error | 10.00±2.45 | 0.86±0.04 | 8.50±3.02 | 0.88±0.05 | 4.83±1.94 | 0.94±0.03 | 4.33±1.21 | 0.94±0.02 |
| | | LS-30, 30% Error | 10.00±2.45 | 0.86±0.04 | 8.83±2.99 | 0.87±0.05 | 6.17±2.40 | 0.92±0.04 | 5.83±1.83 | 0.92±0.03 |
| | | LS-50, 50% Error | 10.00±2.45 | 0.86±0.04 | 8.83±2.99 | 0.87±0.05 | 7.50±1.97 | 0.90±0.03 | 8.17±1.94 | 0.89±0.03 |
| | | Ours (Lag Agnostic) | 9.00±2.97 | 0.87±0.05 | 8.00±2.68 | 0.89±0.04 | 4.00±1.67 | 0.95±0.02 | 4.00±1.67 | 0.95±0.02 |
| ER-4 | Exp | DYNOTEARS | 15.00±1.90 | 0.90±0.01 | 15.00±1.90 | 0.90±0.01 | 15.00±1.90 | 0.90±0.01 | 15.00±1.90 | 0.90±0.01 |
| | | LS-0, Perfect Lags | **11.83±2.64** | **0.92±0.02** | 8.67±2.66 | 0.94±0.02 | 6.50±1.52 | 0.96±0.01 | 7.00±2.45 | 0.95±0.02 |
| | | LS-10, 10% Error | 13.50±2.43 | 0.91±0.02 | 11.50±3.94 | 0.92±0.03 | 11.67±6.09 | 0.92±0.04 | 13.00±2.68 | 0.92±0.02 |
| | | LS-30, 30% Error | 14.33±2.80 | 0.90±0.02 | 13.83±2.79 | 0.91±0.02 | 15.17±4.71 | 0.90±0.03 | 17.50±1.38 | 0.89±0.01 |
| | | LS-50, 50% Error | 14.33±2.80 | 0.90±0.02 | 14.83±2.99 | 0.90±0.02 | 17.17±3.82 | 0.89±0.02 | 23.33±4.76 | 0.85±0.03 |
| | | Ours (Lag Agnostic) | 13.17±2.56 | 0.91±0.02 | 11.00±3.29 | 0.93±0.02 | 10.50±4.97 | 0.93±0.03 | 9.83±2.79 | 0.94±0.02 |
| ER-4 | Gauss | DYNOTEARS | 13.00±3.10 | 0.91±0.02 | 13.00±3.10 | 0.91±0.02 | 13.00±3.10 | 0.91±0.02 | 13.00±3.10 | 0.91±0.02 |
| | | LS-0, Perfect Lags | **10.33±2.73** | **0.93±0.02** | 8.00±2.37 | 0.95±0.02 | 4.83±1.94 | 0.97±0.01 | 4.83±1.33 | 0.97±0.01 |
| | | LS-10, 10% Error | 11.67±3.44 | 0.92±0.02 | 8.17±2.64 | 0.95±0.02 | 6.67±2.58 | 0.96±0.02 | 6.50±2.43 | 0.96±0.02 |
| | | LS-30, 30% Error | 12.50±3.62 | 0.92±0.03 | 9.50±3.27 | 0.94±0.02 | 8.83±1.33 | 0.94±0.01 | 8.33±2.16 | 0.95±0.02 |
| | | LS-50, 50% Error | 12.50±3.62 | 0.92±0.03 | 10.67±2.73 | 0.93±0.02 | 11.17±1.47 | 0.93±0.01 | 11.83±1.94 | 0.92±0.01 |
| | | Ours (Lag Agnostic) | 10.50±2.35 | 0.93±0.02 | 8.00±2.37 | 0.95±0.02 | 5.33±1.86 | 0.97±0.01 | 5.67±2.16 | 0.96±0.01 |

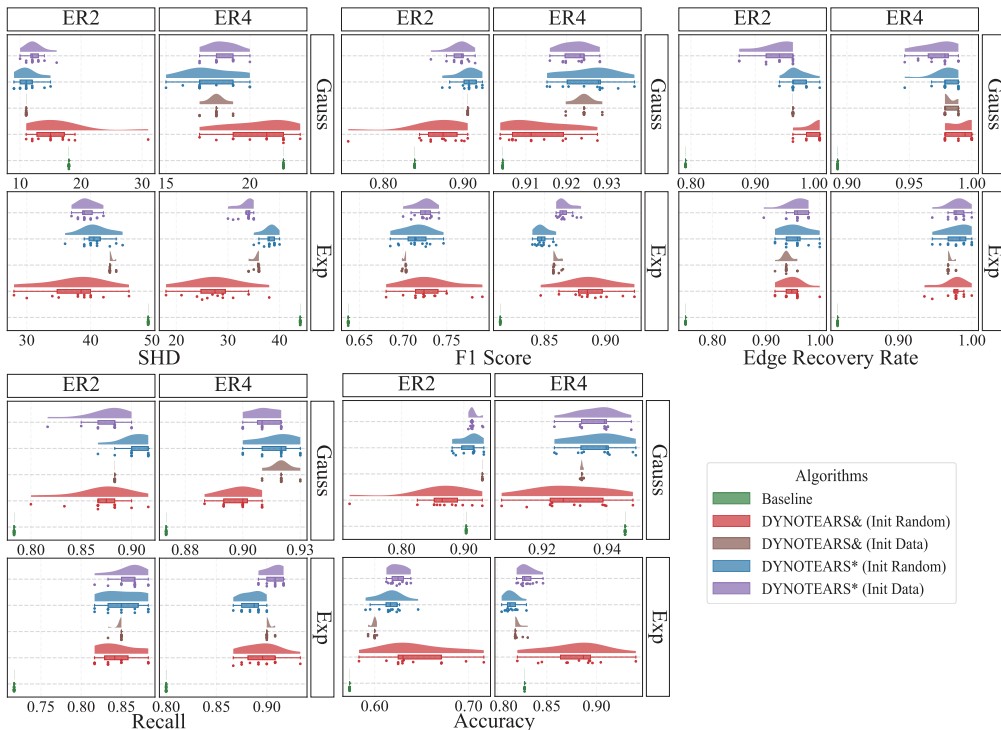

Figure 15: Supplementary results of various initialization strategies on 250 samples.

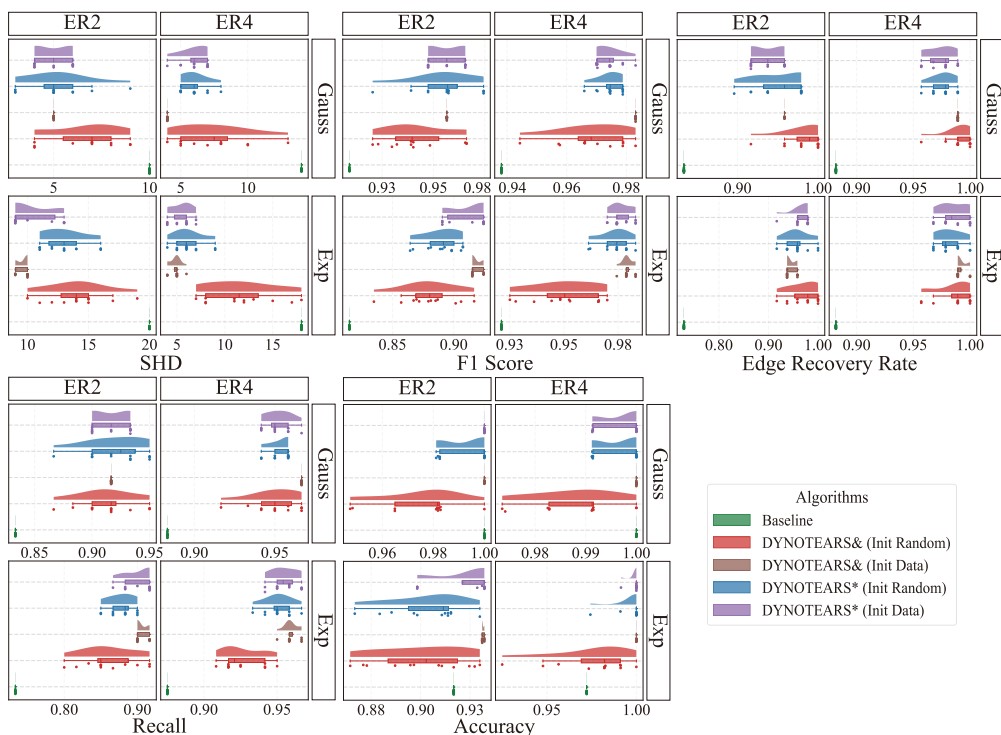

Figure 16: Supplementary results of various initialization strategies on 1000 samples.

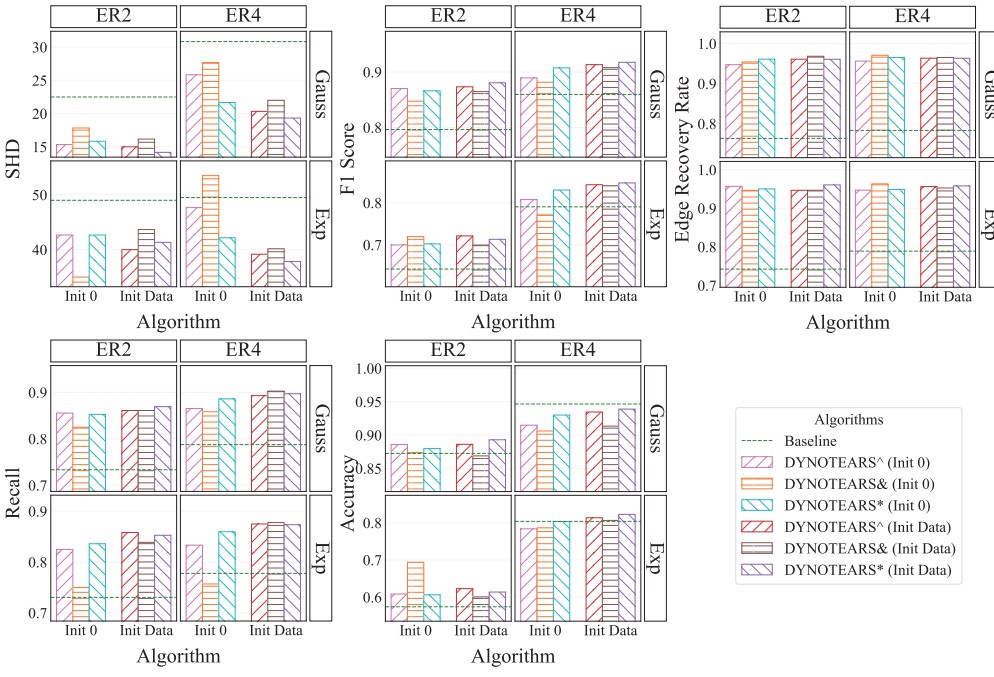

Figure 17: Supplementary results of comparison to maximum-based formulation on 250 samples.

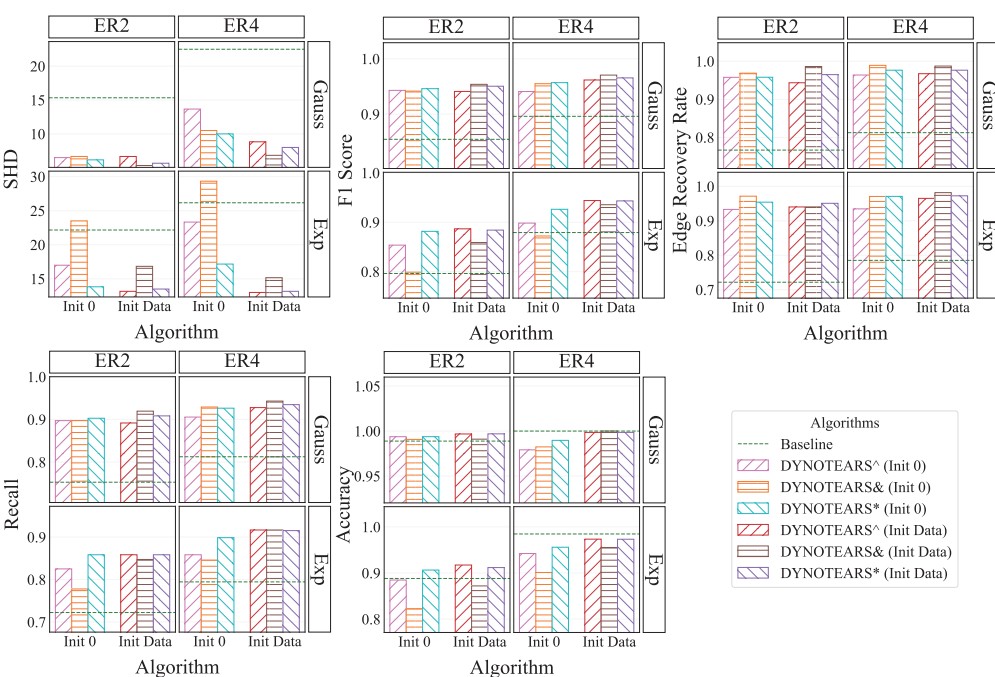

Figure 18: Supplementary results of comparison to maximum-based formulation on 1000 samples.

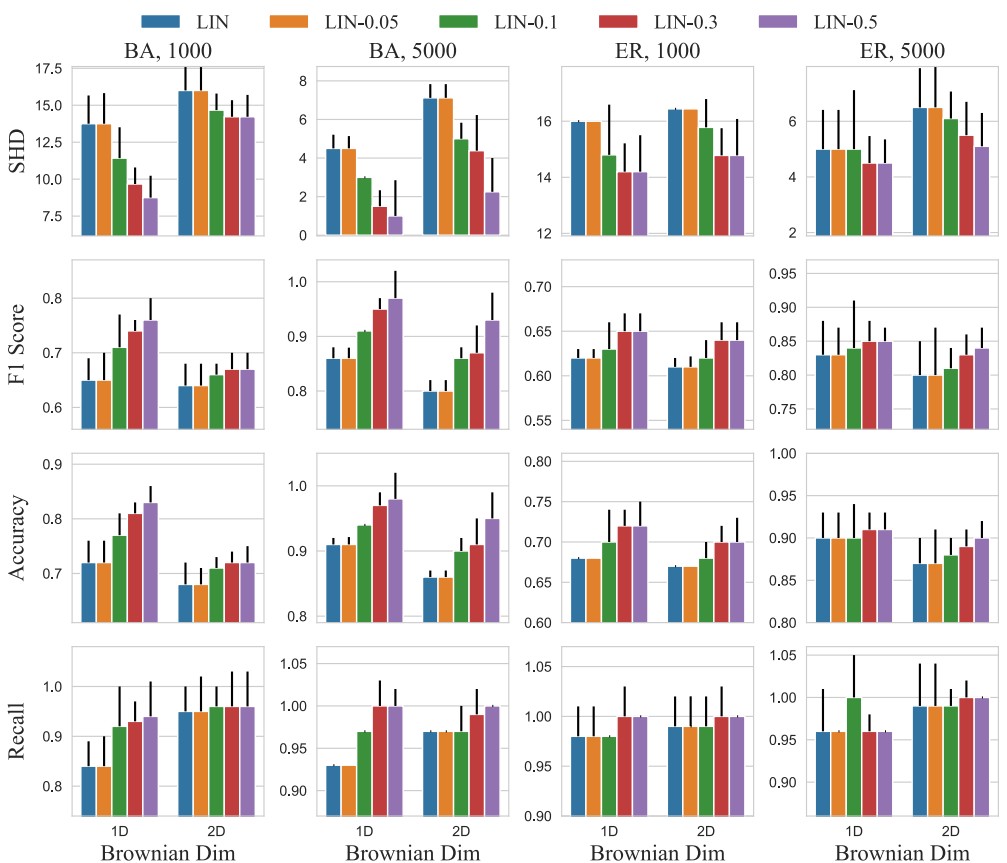

Figure 19: Results on non-stationary and non-linear time-series data with the backbone algorithm LIN. This figure illustrates the performance of the LIN algorithm under various prior rates, evaluated on simulated data with sample sizes of 1000 and 5000. The data was generated using 1D and 2D Brownian motion on Barabási-Albert (BA) and Erdős-Rényi (ER) graph structures with 5 nodes. The evaluation metrics include Structural Hamming Distance (SHD), F1 Score, Accuracy, and Recall

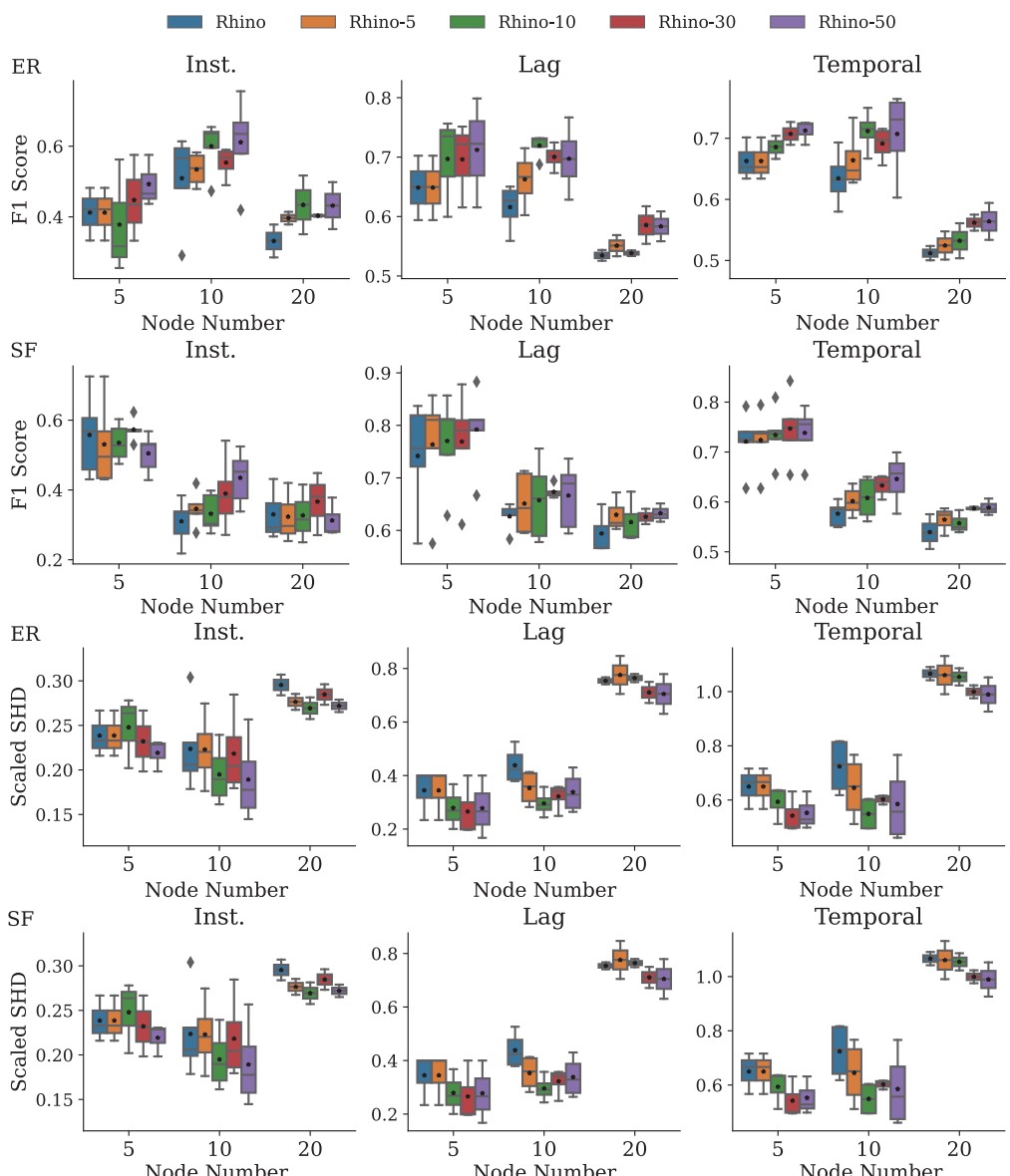

Figure 20: Results on non-stationary and non-linear time-series data with the backbone algorithm RHINO. This figure illustrates the causal discovery capabilities of the RHINO algorithm under various prior rates on simulated data generated from Erdős-Rényi (ER) and Scale-Free (SF) networks with 5, 10, and 20 nodes. The simulation involved a sample size of 5000, a lag order of 2, a noise level of 0.5, and utilized spline functions to model historical noise. The evaluation specifically assesses the algorithm's accuracy in identifying three distinct graph structures: the Instantaneous graph (Inst), which represents causal relationships among variables within the current time point (t); the Lagged graph (Lag), which details causal influences from past time points (t-k) on the current time point (t); and the comprehensive Temporal graph, which combines both instantaneous and lagged relationships. All results are measured using the Scaled Structural Hamming Distance (Scaled SHD) and the F1 Score.

Table 14: Supplementary results on the DERAM4 dataset.

| Sample size | | DREAM4-63 | | DREAM4-126 | | DREAM4-189 | |
|---|---|---|---|---|---|---|---|
| Prior Rate | Method | Loss | AUROC | Loss | AUROC | Loss | AUROC |
| | NOTEARS | 8.20±0.73 | 0.54±0.03 | 8.02±0.92 | 0.55±0.03 | 7.89±0.73 | 0.55±0.03 |
| | NOTEARS+Prior | 7.57±0.52 | 0.66±0.02 | 7.57±0.72 | 0.67±0.02 | 7.66±0.61 | 0.68±0.03 |
| | DYNOTEARS | 7.81±1.14 | 0.58±0.03 | 5.61±1.04 | 0.62±0.04 | 4.08±0.60 | 0.65±0.05 |
| 25% | DYNOTEARS-RandomLag | 7.26±0.63 | 0.64±0.03 | 6.01±0.83 | 0.68±0.04 | 4.94±0.61 | 0.71±0.06 |
| | DYNOTEARS& (Init 0) | **6.60±0.57** | **0.72±0.02** | 5.79±0.70 | 0.74±0.03 | 4.99±0.49 | **0.75±0.05** |
| | DYNOTEARS& (Init Data) | 7.24±0.91 | 0.69±0.02 | **5.19±0.88** | 0.73±0.04 | **3.80±0.53** | 0.74±0.04 |
| | DYNOTEARS* (Init 0) | 6.66±0.54 | 0.71±0.02 | 5.82±0.82 | **0.74±0.04** | 5.08±0.49 | 0.75±0.05 |
| | DYNOTEARS* (Init Data) | 7.19±1.01 | 0.69±0.02 | 5.27±0.92 | 0.73±0.03 | 3.85±0.54 | 0.74±0.05 |
| | NOTEARS | 8.20±0.73 | 0.54±0.03 | 8.02±0.92 | 0.55±0.03 | 7.89±0.73 | 0.55±0.03 |
| | NOTEARS+Prior | 7.13±0.49 | 0.77±0.01 | 7.16±0.62 | 0.78±0.01 | 7.20±0.50 | 0.78±0.02 |
| | DYNOTEARS | 7.81±1.14 | 0.58±0.03 | 5.61±1.04 | 0.62±0.04 | 4.08±0.60 | 0.65±0.05 |
| 50% | DYNOTEARS-RandomLag | 7.25±0.76 | 0.66±0.03 | 6.16±0.93 | 0.71±0.04 | 5.12±0.66 | 0.72±0.06 |
| | DYNOTEARS& (Init 0) | **5.75±0.53** | **0.81±0.01** | 5.09±0.59 | **0.82±0.02** | 4.56±0.47 | **0.83±0.02** |
| | DYNOTEARS& (Init Data) | 6.67±0.82 | 0.79±0.01 | **4.90±0.58** | 0.81±0.02 | **3.52±0.33** | 0.83±0.02 |
| | DYNOTEARS* (Init 0) | 6.01±0.48 | 0.80±0.01 | 5.37±0.61 | 0.82±0.01 | 4.54±0.41 | 0.83±0.03 |
| | DYNOTEARS* (Init Data) | 6.96±0.90 | 0.79±0.01 | 5.08±0.83 | 0.80±0.02 | 3.65±0.45 | 0.82±0.02 |
| | NOTEARS | 8.20±0.73 | 0.54±0.03 | 8.02±0.92 | 0.55±0.03 | 7.89±0.73 | 0.55±0.03 |
| | NOTEARS+Prior | 6.68±0.42 | 0.89±0.01 | 6.61±0.53 | 0.89±0.01 | 6.65±0.42 | 0.89±0.01 |
| | DYNOTEARS | 7.81±1.14 | 0.58±0.03 | 5.61±1.04 | 0.62±0.04 | 4.08±0.60 | 0.65±0.05 |
| 75% | DYNOTEARS-RandomLag | 7.45±0.98 | 0.70±0.02 | 6.20±0.72 | 0.75±0.03 | 5.43±0.84 | 0.77±0.05 |
| | DYNOTEARS& (Init 0) | **5.24±0.57** | **0.91±0.01** | 4.70±0.70 | **0.92±0.01** | 4.29±0.64 | 0.92±0.02 |
| | DYNOTEARS& (Init Data) | 6.02±0.78 | 0.90±0.01 | **4.46±0.64** | 0.91±0.01 | **3.25±0.33** | 0.92±0.02 |
| | DYNOTEARS* (Init 0) | 5.28±0.84 | 0.90±0.01 | 4.66±0.74 | 0.92±0.01 | 4.27±0.39 | **0.92±0.02** |
| | DYNOTEARS* (Init Data) | 6.22±0.72 | 0.89±0.01 | 4.88±0.85 | 0.90±0.01 | 3.40±0.42 | 0.92±0.02 |
| | NOTEARS | 8.20±0.73 | 0.54±0.03 | 8.02±0.92 | 0.55±0.03 | 7.89±0.73 | 0.55±0.03 |
| | NOTEARS+Prior | 6.27±0.39 | 0.99±0.00 | 6.32±0.50 | 1.00±0.00 | 6.30±0.31 | 1.00±0.00 |
| | DYNOTEARS | 7.81±1.14 | 0.58±0.03 | 5.61±1.04 | 0.62±0.04 | 4.08±0.60 | 0.65±0.05 |
| 100% | DYNOTEARS-RandomLag | 7.19±0.79 | 0.75±0.03 | 6.13±0.92 | 0.77±0.04 | 5.73±0.76 | 0.79±0.03 |
| | DYNOTEARS& (Init 0) | **4.91±0.57** | **1.00±0.00** | 4.42±0.63 | **1.00±0.00** | 3.97±0.39 | **1.00±0.00** |
| | DYNOTEARS& (Init Data) | 5.67±0.58 | 0.99±0.00 | **4.29±0.53** | 0.99±0.00 | **3.16±0.36** | 0.99±0.00 |
| | DYNOTEARS* (Init 0) | 5.07±0.55 | 1.00±0.00 | 4.42±0.75 | 0.99±0.00 | 4.18±0.36 | 0.99±0.00 |
| | DYNOTEARS* (Init Data) | 6.30±0.80 | 0.98±0.00 | 4.63±0.68 | 0.99±0.00 | 3.31±0.38 | 0.99±0.00 |

