# OpenReview forum: "Structure Learning from Time-Series Data with Lag-Agnostic Structural Prior"
_ICLR.cc/2026/Conference — ICLR 2026 Poster_

### Official Review · Reviewer_54Cj · 2025-10-27

**Soundness:** 3
**Presentation:** 3
**Contribution:** 2
**Rating:** 6
**Confidence:** 4

**Summary:**

This paper considers  the integration of coarse-grained lag-agnostic causal priors.
The main claim is that lag-agnostic priors can enable the discovery of lag-specific causal links.
The main effort of the paper is put on how to integrate such prior during the practical optimization procedure, and it provides both theoretical and empirical analysis.

**Strengths:**

- lag-agnostic structural priors is an interesting formulation, and it can be useful in practice
- The theoretical analysis is clear and rigorous. It reveals the challenges during the optimization and why the proposed  logic-dual Formulation is essential.

**Weaknesses:**

- In Section 4.2 and Figure 1, it would be more comprehensive if a baseline using equation (9) can be added. Such empirical results would further justify the discussion about process-equivalent approach.
- The concrete research problem is not sufficiently introduced. For example, the concept of "process equivalence" lacks a more formal definition. I suggest a minor adjustment to emphasize this part.

**Questions:**

- What is the connection and differences between the Lag-Agnostic Structural Prior and those in the related work? Especially the "order" of causal variables, like Partial Orders [1] and Causal Orders [2]. I suggest an additional discussion.
- Please consider improving the notation about $\Theta\_{ij,s}\:=\\{\\theta\\mid \|(W\_s(\\theta))\_{ij}\| \\geq \\delta \\}$. Readers may expect $\\Theta\_{ij,s}(0)\:=\\{\\theta\\mid\|(W\_s(\\theta))\_{ij}|\\geq 0\\}$, which does not match the actual definition of $\Theta_{ij,s}(0)$.

[1] Differentiable Structure Learning with Partial Orders

[2] Causal Order: the Key to Leverage Imperfect Experts in Causal Inference

---

> ### Author Response · Authors · 2025-11-16
> **Part 1 of Response**
>
> Thanks for your throughout review and valuable comments. Here are our responses.
>
> ## Weakness 1
>
> Thank you for this excellent suggestion. We agree this is a crucial comparison and have now added the max-based formulation (from Equation (9) ) as a baseline, denoted **DYNOTEARS^**, to Figure 1, Table 1, and Table 2.
>
> These new results justify our discussion on process-equivalence, as they reveal the empirical failure of the flawed max-based approach, especially on real-world data.
>
> Partial results of the new Table 2, on the realistic data, are presented below.
>
> | @Priors | Method                  | DREAM4-63     |               | DREAM4-126    |               | DREAM4-189    |               |
> | ------- | ----------------------- | ------------- | ------------- | ------------- | ------------- | ------------- | ------------- |
> |         |                         | Loss↓         | AUROC↑        | Loss↓         | AUROC↑        | Loss↓         | AUROC↑        |
> | 75%     | DYNOTEARS               | 7.81±1.14     | 0.58±0.03     | 5.61±1.04     | 0.62±0.04     | 4.08±0.60     | 0.65±0.05     |
> |         | DYNOTEARS^  (Init Data) | 7.58±0.11     | 0.62±0.04     | 5.43±0.12     | 0.67±0.03     | 4.35±0.06     | 0.71±0.06     |
> |         | DYNOTEARS& (Init Data)  | **6.02±0.78** | **0.90±0.01** | **4.46±0.64** | **0.91±0.01** | **3.25±0.33** | **0.92±0.02** |
> |         | DYNOTEARS* (Init Data)  | 6.22±0.72     | 0.89±0.01     | 4.88±0.85     | 0.90±0.01     | 3.40±0.42     | **0.92±0.02** |
>
> We also present partial results for Table 1 below (Figure 1 is omitted here).
>
> | **Method**              | **5% Priors** |             | **10% Priors** |             | **30% Priors** |             | **50% Priors** |             |
> | :---------------------- | :------------ | :---------- | :------------- | :---------- | :------------- | :---------- | :------------- | :---------- |
> |                         | **SHD↓**      | **F1↑**     | **SHD↓**       | **F1↑**     | **SHD↓**       | **F1↑**     | **SHD↓**       | **F1↑**     |
> | DYNOTEARS               | 13.00±3.10    | 0.91±0.02   | 13.00±3.10     | 0.91±0.02   | 13.00±3.10     | 0.91±0.02   | 13.00±3.10     | 0.91±0.02   |
> | DYNOTEARS^ (Init Data)  | 11.50 ± 2.26  | 0.92 ± 0.02 | 9.33 ± 2.34    | 0.94 ± 0.02 | 7.83 ± 2.04    | 0.94 ± 0.02 | 6.83 ± 1.47    | 0.95 ± 0.01 |
> | DYNOTEARS& (Init Data)  | 10.83 ± 1.97  | 0.93 ± 0.01 | 8.50 ± 3.78    | 0.94 ± 0.03 | 5.83 ± 1.97    | 0.97 ± 0.01 | 6.17 ± 2.71    | 0.95 ± 0.02 |
> | DYNOTEARS*  (Init Data) | 10.50 ± 2.35  | 0.93 ± 0.02 | 8.00 ± 2.37    | 0.95 ± 0.02 | 5.33 ± 1.86    | 0.97 ± 0.01 | 5.50 ± 1.76    | 0.96 ± 0.01 |
>
> We will update the main paper with these new results.
>
> ## Weakness 2
>
> Thank you for this suggestion. We agree that "process equivalence" is a core concept and benefits from a more formal definition to clarify our research problem.
>
> We have revised the paper to explicitly add this definition, which formally introduces the property we analyze.
>
> > **Definition (Process Equivalence):** A lag-agnostic loss formulation is **process-equivalent** if it respects the prior's "lag-unknown" semantics throughout the entire optimization process. It must not introduce explicit bias toward any specific lag, but must instead evaluate all candidate lags holistically to select the one(s) most compatible with the data.
>
> This definition now serves as the formal criterion that we show the naive max-based loss fails, and which our proposed loss functions are designed to satisfy.

---

> ### Author Response · Authors · 2025-11-16
> **Part 2 of Response**
>
> ## Question 1
>
> Thank you for this excellent question. It allows us to clarify the precise relationship between our work and other forms of structural priors.
>
> Here is our detailed discussion of the connections and differences.
>
>
>
> **1. Connection: The Source of Knowledge**
>
> First, the **connection** is that our "lag-agnostic prior" often comes from the **exact same source** as priors used in static (non-temporal) causal discovery.
>
> Domain experts can often provide a "summary graph" (e.g., "gene A regulates gene B," "component A affects component B") without any information on time lags. In static causal discovery, this is treated as a simple edge/path/order prior. Our work is the first to formalize how to *correctly* use this same, common, lag-agnostic knowledge in the more complex *temporal, lag-aware* setting.
>
>
>
> **2. Difference: A Hierarchy of Constraints**
>
> The priors, edge, path, and order, exist in a **hierarchy of strength**:
>
> **Edge Presence ($i \rightarrow j$)** $\implies$ **Path Existence ($i \leadsto j$)** $\implies$ **Partial Order ($i \prec j$)**
>
> **Our work focuses on $i \rightarrow j$:** We tackle "lag-agnostic **edge** presence," which is the strongest constraint in this hierarchy.
>
>
>
> **3. Comparison to Partial Orders [1]**
>
> - **What it is:** A partial order ($i \prec j$) is primarily a **forbidden-type constraint**. It states that *no path* from $j$ to $i$ can exist. It is *weaker* than an edge/path existence prior; for example, $i \prec j$ allows for the case where $i$ and $j$ are not causally related at all.
>
> - **Key Difference (Optimization):** This constraint is **gradient-consistent** with the DAG penalty. Both the partial order penalty and the DAG penalty work by pushing weights *down* (toward 0) to enforce acyclicity. This makes optimization relatively stable.
>
>
>
> **4. Comparison to Path/Ancestral Constraints [2]**
>
> - **What it is:** A path existence prior ($i \leadsto j$) is an **existence-type constraint**.
>
> - **Key Similarity (The Core Challenge):** This is the most crucial connection. Both "path existence" [2] and our "lag-agnostic edge existence" are **gradient-conflicting** with the DAG penalty.
>
>   - Both priors introduce a penalty that must push weights *up* (away from 0) to "create" a path or an edge.
>   - This is in direct conflict with the DAG penalty, which pushes weights *down* to 0.
>   - This conflict is the source of the optimization instability that both our work and [2] must solve.
>
> - **Key Difference (The Nature of Uncertainty):**
>
>   - In **Path Constraints**, the uncertainty is over the *set of edges* that form the path (e.f., $i \rightarrow k \rightarrow j$ vs. $i \rightarrow l \rightarrow j$).
>
>   - In **Our Constraint**, the uncertainty is over the *set of lags* for a *single edge* (e.g., $(W_1)_{ij}$ vs. $(W_2)_{ij}$).
>
>
>
> **In summary:** Our work introduces the **strongest** form of this prior (edge existence) to the **temporal** domain. By doing so, we identify a problem that shares the fundamental optimization challenge (gradient conflict) with path/ancestral constraints, but which has a unique structural uncertainty (over lags, not paths).
>
> Future work could indeed explore incorporating the weaker lag-agnostic path or order priors into the temporal setting, building on the foundation we have set here.
>
>
>
> [1] Differentiable Structure Learning with Partial Orders
>
> [2] Differentiable Structure Learning with Ancestral Constraints
>
>
>
> ## Question 2
>
> Thank you for this helpful suggestion. To improve clarity, we have revised Section 2.2 to use explicit, unambiguous notation. For your convenience, the updated text in the manuscript now reads:
>
> > To enforce the presence or absence of a specific lagged edge $(W_s)_{ij}$, there are two approaches, a hard constraint and a soft penalty. The hard way directly constrains the corresponding parameters. For example, enforcing presence can be written as:
> >
> > \min_{\theta\in\Theta^{\text{pres}}_{ij,s}(\delta)} \mathcal{L}(X;\{W_{\tau}(\theta)\}_{\tau=0}^{L}) \quad , \text{ subject to } h(W_0)=0, \quad (3)
> >
> > where we use explicit notation for the constraint sets. The presence set is \Theta^{\text{pres}}_{ij,s}(\delta) := \{\theta \mid |(W_s(\theta))_{ij}| \ge \delta\}, which enforces that the edge (W_s)_{ij} exists with a magnitude of at least $\delta$. The absence set is \Theta^{\text{abs}}_{ij,s} := \{\theta \mid (W_s(\theta))_{ij} = 0\}, which enforces its removal. This strict constraint approach was used in the work by Sun et al. (2023).

---

> > ### Comment · Reviewer_54Cj · 2025-11-25
> >
> > Thanks for the response. My concerns have been sufficiently addressed. Has updated the score accordingly.

---

### Official Review · Reviewer_2t4w · 2025-10-29

**Soundness:** 3
**Presentation:** 3
**Contribution:** 3
**Rating:** 6
**Confidence:** 3

**Summary:**

This paper addresses the problem of causal structure learning from time-series data when only lag-agnostic prior knowledge is available. The authors propose a continuous optimization framework that integrates such priors into time-series structure learning. The paper first identifies the process-inequivalence issue in naive maximum-based formulations for lag-agnostic priors, which causes bias toward specific lags during optimization (Section 3.2, Proposition 1). To address this, it introduces two process-equivalent formulations, a binary-masked formulation (Eq. 10) and a logic-dual formulation (Eq. 11), that preserve the semantics of lag-agnostic priors throughout optimization (Section 3.3). The authors further analyze the non-convexity induced by lag-agnostic constraints and propose a data-driven initialization strategy (Section 3.4) to mitigate convergence to poor local optima. Finally, through comprehensive experimental validation on synthetic data, non-linear and non-stationary datasets (using LIN and RHINO backbones), and real-world DREAM4 gene regulatory networks (Sections 4.1–4.4), the paper demonstrates that the proposed framework improves causal recovery and stability compared to both data-only and lag-specific prior methods, particularly when temporal information is noisy or incomplete.

**Strengths:**

Originality: The notion of lag-agnostic structural priors is novel and fills a clear gap between lag-specific causal discovery (e.g., Sun et al., 2023) and coarse-grained prior-based static structure learning (e.g., Zheng et al., 2018). The formal distinction between consequence equivalence and process equivalence is particularly insightful and provides new conceptual clarity (Proposition 1–4, Section 3).

Technical quality: The theoretical analysis is rigorous. The proofs of process-equivalence (Appendix B) and the illustration of increased non-convexity via Example 1 are convincing. The data-driven initialization strategy is well-motivated and empirically validated.

Clarity: The paper is well-structured, with clear notation and well-separated sections. Figures 1–2 and Tables 1–2 (pages 8–9) are clear and they effectively support the claims.

Significance: The work provides a general, modular mechanism that can be integrated into various differentiable structure learning frameworks (DYNOTEARS, LIN, RHINO). The experiments demonstrate consistent improvement across backbones, suggesting wide applicability.

**Weaknesses:**

Scalability considerations: The computational complexity of the binary-masked and logic-dual penalties is not analyzed. As both require operations across all lags and node pairs, their efficiency and scaling to large graphs (e.g., d > 100) remain unclear.

Interpretability of lag selection: Although the process-equivalent formulations prevent early bias, the final lag assignments are driven primarily by data fitting. The paper could elaborate more on how reliably the method identifies the true lag rather than simply satisfying priors (discussion in Section 3.3).

Ablation clarity: While Appendix E reportedly contains ablations, the main text provides limited discussion of which component (formulation type vs. initialization) contributes most to performance gains.

**Questions:**

Complexity and scalability: How does the proposed penalty (especially the product-based logic-dual term) scale with increasing lag length and variable count?

Initialization sensitivity: How sensitive is the method to the choice of unconstrained pre-training in Eq. (14)? Does using different unconstrained learners (e.g., VAR vs. neural backbones) affect performance?

Interpretability of lag selection: Can the authors quantify how often the method identifies the correct lag (when available) rather than merely satisfying the lag-agnostic constraint?

Practical deployment: Are there specific guidelines for tuning λₚ when validation ground truth is unavailable?

---

> ### Author Response · Authors · 2025-11-17
> **Part 1 of Response**
>
> ## Weakness 1
>
> Thank you for raising this important point. We have performed both a theoretical and an empirical analysis to address the scalability of our method.
>
> 1. Theoretical Complexity:
>
> The per-iteration complexity of our prior loss (both $p_{bin}$ and $p_{or}$) is $O(L d^2)$. This is negligible compared to the data-fit loss of $O(T L d^2)$ and does not create a computational bottleneck.
>
> 2. Empirical Scaling (New Experiment):
>
> We ran a new experiment on large graphs (100 and 200 nodes, ER4graph, Gaussian noise, 30% priors) to test practical scaling. We report the results of the product-based loss formulation, logic-dual loss, below.
>
> | **Nodes** | **Algorithm**          | **SHD ↓** | **F1-Score ↑** | **Runtime (s)** |
> | --------- | ---------------------- | --------- | -------------- | --------------- |
> | **100**   | Baseline (DYNOTEARS)   | 95        | 0.815          | 46.43           |
> | **100**   | **Ours (DYNOTEARS\*)** | **65**    | **0.894**      | 237.89          |
> | **200**   | Baseline (DYNOTEARS)   | 164       | 0.778          | 216.08          |
> | **200**   | **Ours (DYNOTEARS\*)** | **112**   | **0.863**      | 1364.46         |
>
> The results show:
>
> - **Performance:** Our method scales well, achieving significant SHD and F1 improvements on large graphs.
> - **Runtime:** The 5-6x runtime increase is not from the *loss calculation*, but from the **optimization process**. The combined non-convexity of the DAG and our prior requires more iterations to converge.
>
> This analysis confirms our method's scalability in terms of graph quality, and we will add these results to the paper.
>
>
>
> ## Weakness 2
>
> Thank you for this sharp observation. You are correct that our process-equivalent loss (Sec 3.3) is *designed* to prevent bias and allow the data-fitting term to make the final lag selection. As you note, the key question is: **how reliably does this data-driven choice find the true lag?**
>
> To answer this directly, we explicitly measure this. We tracked a set of 23 true lag-agnostic edges (on a 20-node, ER4, L=5, Gaussian setup) and measured not only if our method *satisfied* the prior (i.e., found an edge at *any* lag) but if it found the **ground-truth lag-specific edge**.
>
> The results show our method is reliable:
>
> | **Method**               | **SHD ↓** | **F1 ↑**  | **Priors Provided** | **Priors Satisfied (any lag)** | **Priors Recovered (at True Lag)** |
> | ------------------------ | --------- | --------- | ------------------- | ------------------------------ | ---------------------------------- |
> | **Baseline** (Data-Only) | 12.0      | 0.919     | 0                   | 12.0 / 23                      | **13.5** / 24.83                   |
> | **Ours** (Prior-Guided)  | **2.67**  | **0.983** | 23                  | 21.3 / 23                      | **22.8** / 24.83                   |
>
> *(Std. dev. omitted for clarity.)*
>
> Our Method, when guided by the 23 *lag-agnostic* priors, not only satisfied the prior (21.3 / 23) but, more importantly, it pinpointed the ground-truth lag for **22.8 out of 24.83** edges (a lag-agnostic prior may contain multiple lag-specific edges).
>
> This experiment provides strong empirical evidence for the theory in Proposition 3. It confirms our framework does not just "satisfy the prior"; it uses the prior as a guide, and the data-fitting term then **reliably selects the correct lag**. We will add this metric and discussion to the paper.

---

> > ### Author Response · Authors · 2025-11-17
> > **Part 2 of Response**
> >
> > ## Weakness 3
> >
> > Thank you for your comment. You are right that disentangling the contributions of the **loss formulation** vs. the **initialization strategy** is critical. We have conducted new experiments and demonstrates that:
> >
> > 1. The correct, process-equivalent formulation is the foundation and the most critical component.
> > 2. The data-driven initialization is a crucial enabler\. It navigates the new non-convex landscape introduced by the prior loss of lag-agnostic edge presence, allowing the optimizer to find a superior local optimum.
> >
> > **1. Evidence from Real-World Data (DREAM4)**
> >
> > We ran a full ablation on the DREAM4 dataset. The results are very revealing. The Flawed Loss (DYNOTEARS^) is the process-inequivalent max-based formulation.
> >
> > | Method              | Init      | DREAM4-63 |         | DREAM4-126 |         | DREAM4-189 |         |
> > | ------------------- | --------- | --------- | ------- | ---------- | ------- | ---------- | ------- |
> > |                     |           | Loss ↓    | AUROC ↑ | Loss ↓     | AUROC ↑ | Loss ↓     | AUROC ↑ |
> > | Baseline            | -         | 7.81      | 0.58    | 5.61       | 0.62    | 4.08       | 0.65    |
> > | Flawed Loss (max)   | Init 0    | 7.6       | 0.89    | 5.49       | 0.90    | 4.41       | 0.9     |
> > | Flawed Loss (max)   | Init Data | 7.58      | 0.62    | 5.43       | 0.67    | 4.35       | 0.71    |
> > | Our Loss (Binary &) | Init 0    | 5.24      | 0.91    | 4.7        | 0.92    | 4.29       | 0.92    |
> > | Our Loss (Binary &) | Init Data | 6.02      | 0.90    | 4.46       | 0.91    | 3.25       | 0.92    |
> > | Our Loss (Logic *)  | Init 0    | 5.28      | 0.90    | 4.66       | 0.92    | 4.27       | 0.92    |
> > | Our Loss (Logic *)  | Init Data | 6.22      | 0.89    | 4.88       | 0.91    | 3.40       | 0.92    |
> >
> > *(Std. dev. omitted for table clarity)*
> >
> > Analysis of DREAM4 Data:
> >
> > - Formulation is Most Critical: Our proposed formulations (both & and *) achieve superior AUROC and significantly better test-set loss compared to the Flawed Loss when using Init 0.
> >
> > - The Flawed Loss with Init Data is significantly worse than with Init 0.
> >
> > - For the correct formulations, Init Data provides a significant performance boost with more data (DREAM4-189), it drops the test loss from ~4.2 to ~3.3, a >20% improvement.
> >
> >
> >
> > **2. Evidence from Synthetic Data**
> >
> > This ablation on synthetic data (20-node ER4, L=5, Gaussian) reinforces the findings.
> >
> > | Method              | Init      | 5% Priors |      | 10% Priors |      | 30% Priors |      | 50% Priors |      |
> > | ------------------- | --------- | --------- | ---- | ---------- | ---- | ---------- | ---- | ---------- | ---- |
> > |                     |           | SHD↓      | F1↑  | SHD↓       | F1↑  | SHD↓       | F1↑  | SHD↓       | F1↑  |
> > | Baseline            | -         | 13.00     | 0.91 | 13.00      | 0.91 | 13.00      | 0.91 | 13.00      | 0.91 |
> > | Flawed Loss (max)   | Init 0    | 12.00     | 0.92 | 10.17      | 0.93 | 8.50       | 0.94 | 7.83       | 0.95 |
> > | Flawed Loss (max)   | Init Data | 11.50     | 0.92 | 9.33       | 0.94 | 7.83       | 0.94 | 6.83       | 0.95 |
> > | Our Loss (Binary &) | Init 0    | 11.67     | 0.92 | 9.83       | 0.93 | 6.67       | 0.96 | 7.00       | 0.95 |
> > | Our Loss (Binary &) | Init Data | 10.83     | 0.93 | 8.50       | 0.94 | 5.83       | 0.97 | 6.17       | 0.95 |
> > | Our Loss (Logic *)  | Init 0    | 10.87     | 0.93 | 8.83       | 0.94 | 6.00       | 0.96 | 5.93       | 0.96 |
> > | Our Loss (Logic *)  | Init Data | 10.50     | 0.93 | 8.00       | 0.95 | 5.33       | 0.97 | 5.50       | 0.96 |
> >
> > Analysis of Synthetic Data:
> >
> > - The correct loss formulation outperforms the flawed loss formulation for both init 0 and init data.
> > - Init Data provides a more significant boost to the **binary-masked (&)** loss than the logic-dual (*) loss. This supports our analysis: the fully continuous * formulation is inherently more stable and less dependent on a good start than the non-differentiable & formulation.
> >
> > Conclusion:
> >
> > Both components are vital, but the correct, process-equivalent formulation is the fundamental pre-requisite for good performance. Data-initialization is then a powerful "enabler" that helps our process-equivalent (but non-convex) formulation find a superior solution. We will add this detailed ablation to the paper.
> >
> >
> >
> > ## Question 1
> >
> > This point is addressed in our reply to Weakness 1.

---

> > > ### Author Response · Authors · 2025-11-17
> > > **Part 3 of Response**
> > >
> > > ## Question 2
> > >
> > > Thank you for this insightful question. In response, we ran a new experiment. We now compare **three** initialization strategies:
> > >
> > > 1. **Init 0:** Standard zero initialization.
> > > 2. **Init VAR (New):** The strategy you suggested. We pre-train a simple VAR model by optimizing *only* the data-fit loss $\mathcal{L}(X;W_{0:L})$, with **no DAG constraint** ($h(W_0)$ is ignored).
> > > 3. **Init Data (Our Paper's Method):** The solution to Eq. (14), which optimizes *both* the data-fit loss $\mathcal{L}$ *and* the non-convex DAG constraint $h(W_0)$.
> > >
> > > The results (on 20-node, ER4, L=5, Gaussian) are shown below:
> > >
> > > | **Initialization Strategy** | **SHD ↓** | **F1-Score ↑** | **Accuracy ↑** | **Recall ↑** |
> > > | --------------------------- | --------- | -------------- | -------------- | ------------ |
> > > | Baseline (No Prior)         | 20.0      | 0.859          | 0.984          | 0.762        |
> > > | **Init 0** (Our Loss &)     | 7.50      | 0.951          | 0.985          | 0.921        |
> > > | **Init VAR** (Our Loss &)   | 5.50      | 0.964          | 1.000          | 0.931        |
> > > | **Init Data** (Our Loss &)  | **5.17**  | **0.967**      | **1.000**      | **0.935**    |
> > > |                             |           |                |                |              |
> > > | **Init 0** (Our Loss *)     | 6.67      | 0.957          | 0.987          | 0.929        |
> > > | **Init VAR** (Our Loss *)   | 6.17      | 0.960          | 1.000          | 0.923        |
> > > | **Init Data** (Our Loss *)  | **5.50**  | **0.964**      | **1.000**      | **0.931**    |
> > >
> > > *(Note: Prior satisfaction was high (>94%) for all methods and is omitted for clarity)*
> > >
> > > This experiment clearly isolates the source of our performance gains:
> > >
> > > 1. **Init VAR > Init 0:** Using a simple VAR pre-training *does* improve performance. This shows that starting from a good *data-fit* region is better than starting from zero.
> > > 2. **Init Data > Init VAR:** Our proposed Init Data strategy is consistently the best.
> > >
> > > The Init Data strategy is superior because it solves the non-convex DAG constraint first, providing a high-quality, acyclic $W_0$ *before* introducing the second non-convex prior loss. This avoids the initial conflict and leads to a better final optimum. Nevertheless, using a simpler initialization like the VAR also contributes to the results.
> > >
> > >
> > >
> > > ## Question 3
> > >
> > > This point is addressed in our reply to Weakness 2.
> > >
> > >
> > >
> > > ## Question 4
> > >
> > > Thank you for this key practical question.A general approach is to monitor the **number of priors satisfied** as $\lambda_p$ is gradually increased. A practitioner can select $\lambda_p$ from the "elbow" of this curve, i.e., the point where the number of satisfied priors stops increasing significantly.
> > >
> > > A new experiment varying $\lambda_p$ from 0.001 to 100 shows that performance immediately **plateaus at a near-optimal level** for any $\lambda_p \ge 0.5$.
> > >
> > > *(20-node ER4, L=5, Gaussian, 57 Priors)*
> > >
> > > | **Method**      | $\lambda_p$ | **SHD ↓** | **F1-Score ↑** | **Priors Satisfied** |
> > > | --------------- | ----------- | --------- | -------------- | -------------------- |
> > > | Baseline        | -           | 28.0      | 0.868          | 43 / 57              |
> > > | **Ours**        | 0.001       | 28.0      | 0.868          | 43 / 57              |
> > > | **(Init Data)** | 0.1         | 14.5      | 0.936          | 52 / 57              |
> > > |                 | **0.5**     | **8.8**   | **0.962**      | **55.5 / 57**        |
> > > |                 | **1.0**     | **8.7**   | **0.963**      | **55.4 / 57**        |
> > > |                 | **10.0**    | **8.2**   | **0.965**      | **56.5 / 57**        |
> > > |                 | **100.0**   | **9.3**   | **0.960**      | **56.6 / 57**        |
> > >
> > > We found that our recommended method (DYNOTEARS* with Init Data) is **remarkably robust** to this hyperparameter. Precise tuning is not required. A user can simply set a sufficiently large weight (e.g., $\lambda_p=1.0$) to ensure priors are enforced. The method is not sensitive to this choice, which is a significant practical advantage.

---

> > > > ### Comment · Reviewer_2t4w · 2025-11-25
> > > >
> > > > Thank you for engaging in the rebuttal. My score remains unchanged.

---

### Official Review · Reviewer_rVC5 · 2025-10-31

**Soundness:** 3
**Presentation:** 3
**Contribution:** 3
**Rating:** 4
**Confidence:** 2

**Summary:**

The paper addresses structure learning for multivariate time series when practitioners have coarse, lag-agnostic causal priors (edge exists but unknown lag). Authors show a straightforward "max over lags" penalty is consequentially equivalent to the desired prior but process inequivalent (induces early bias to a single lag). They propose two process-equivalent penalties that act across all lags: 1) Binary-masked loss $p_\text{bin}$: activates only when all lag-specific edges are below threshold, then pushes them jointly; 2) Logic-dual/product loss $p_\text{or}$: product of ReLU terms (OR semantics), with a normalization to reduce scale sensitivity. They also argue lag-agnostic constraints increase non-convexity and propose a two-stage, data-driven initialization.
Experiments on synthetic VAR graphs (ER-k, Gaussian/Exponential noise), nonlinear/non-stationary backbones (LIN, RHINO), and DREAM4 show consistent gains in SHD/F1/AUROC and lower regression loss vs data-only baselines and vs lag-specific priors with wrong lags.

**Strengths:**

- Lag-agnostic prior knowledge is common; formalizing it is useful.
- Simple, plug-in losses applicable to multiple backbones (DYNOTEARS, LIN, RHINO).
- Initialization story is well-motivated.
- Broad evaluation

**Weaknesses:**

- Propositions establish equivalence of penalties but there's no optimization-theoretic guarantee (e.g., convergence to a correct lag under identifiability conditions).
- The non-convexity example is illustrative but small; more formal landscape analysis would strengthen claims.
- The product loss can suffer vanishing gradients when many lags are near but below $\delta$; the normalization helps but may not fully address scale with larger L.
- How robust are results to noisy/incorrect presence and incorrect absence priors (false positives/negatives in $C_p$, $C_a$)?
- Baselines focus on NOTEARS-family and a “random-lag” variant. Important time-series causal methods like DYNOTEARS with group-sparsity across lags isn't compared.
- No comparison to softmax/log-sum-exp surrogates for max (temperature-controlled) which are a natural alternative to address process-inequivalence.

**Questions:**

-  Any empirical comparison to LSE or Gumbel-softmax over lags?
- What happens when $C_p$ contains 20-40% spurious pairs or $C_a$ wrongly forbids true edges?
- For $p_\text{or}$, how often do you observe near-zero gradients early? Does the normalization fully fix it as L grows?

---

> ### Author Response · Authors · 2025-11-16
> **Part 1 of Response**
>
> Thanks for your throughout review and valuable comments. Here are our replies.
>
> ## Weakness 1
>
> Thank you for this crucial point. You are correct that we do not provide a global convergence guarantee, which is a very high theoretical bar. This aspect is clarified in the limitation section (Appendix D).
>
> - First, we respectfully highlight that **even the NOTEARS backbone is not guaranteed to find the global optimum**. The non-convexity of the DAG constraint is a well-known and challenging open issue in the differentiable causal discovery community.
> - Our task is inherently *more* complex, as we introduce **additional non-convexity** from the lag-agnostic prior loss (as proven in Example 1) and **new gradient conflicts between the presence prior and the DAG penalty**.
>
> Therefore, our contribution scope is focused on **first introducing, formalizing, and solving the challenges unique to this novel problem**. Specifically:
>
> 1. We **identify and analyze** the key issues of process-inequivalence (Prop. 1) and optimization instability (Example 1) .
> 2. We **propose concrete solutions** to these issues: two process-equivalent loss formulations (Prop. 3) and a data-driven initialization strategy.
> 3. We **demonstrate the effectiveness** of these solutions via extensive experiments on complex synthetic and real-world data, showing our methods consistently and significantly outperform baselines (Fig. 1, 2, Table 2).
>
> We will clarify this scope in the revision.
>
> ## Weakness 2
>
> Thank you for your suggestion. We agree that summarizing the landscape results beyond the illustrative example strengthens the paper. Accordingly, we have added a more general statement below to highlight the additional non-convexity induced by the lag-agnostic prior loss.
>
> **Proposition.**
>  Let $\min_W \mathcal{L}(W; X)$ be a convex, linear VAR optimization problem. Then, the augmented problem
>   $\min_W \mathcal{L}(W; X) + \mathcal{C}_p \circ \bar{p}(W)$
>  is non-convex for a set of lag-agnostic edge-presence constraints $\mathcal{C}_p \in \{0,1\}^{d \times d}$ and our proposed prior loss $\bar{p}(W)$.
>
> This result follows directly from Example 1. Specifically, the original problem can be expressed as
> $\min_W \mathcal{L}(W; X) \quad \text{subject to } h(W_0) = 0,$
>  and the incorporation of the prior loss introduces additional non-convexity to this formulation.
>
> Additionally, we have performed a more detailed analysis on the challenge of suboptimal optima, which further makes up for the example and clarifies the role of our data-driven initialization. We respectfully refer the reviewer to our response to **Question 1 of Reviewer tkmz** for this in-depth discussion.
>
> ## Weakness 3
>
> Thanks for raising this concern. To test the *practical* limit of our **normalized p_or loss**, we ran a new experiment on a 10-node, ER-4 graph, extending the maximum lag $L$ up to 50 (a horizon we believe is sufficient for most real-world data). We find that our normalization is highly effective and this theoretical issue does not manifest in practice.
>
> | **Method**             | **Lag = 5** |             | **Lag = 10** |             | **Lag = 20** |             | **Lag = 30** |             | **Lag = 40** |             | **Lag = 50** |             |
> | :--------------------- | :---------: | :---------: | :----------: | :---------: | :----------: | :---------: | :----------: | :---------: | :----------: | :---------: | :----------: | :---------: |
> |                        |  **SHD↓**   |   **F1↑**   |   **SHD↓**   |   **F1↑**   |   **SHD↓**   |   **F1↑**   |   **SHD↓**   |   **F1↑**   |   **SHD↓**   |   **F1↑**   |   **SHD↓**   |   **F1↑**   |
> | Baseline (DYNOTEARS)   | 8.00 ± 1.41 | 0.89 ± 0.02 | 4.00 ± 0.89  | 0.95 ± 0.01 | 6.00 ± 1.26  | 0.92 ± 0.02 | 9.00 ± 1.64  | 0.88 ± 0.03 | 7.00 ± 1.34  | 0.90 ± 0.02 | 8.00 ± 1.58  | 0.89 ± 0.03 |
> | Ours ($p_{\text{or}}$) | 2.83 ± 0.75 | 0.96 ± 0.01 | 1.00 ± 0.63  | 0.99 ± 0.01 | 1.17 ± 0.75  | 0.99 ± 0.01 | 4.50 ± 0.96  | 0.94 ± 0.01 | 3.50 ± 0.84  | 0.95 ± 0.01 | 3.00 ± 0.63  | 0.96 ± 0.01 |
>
> The results have shown that the product-base prior loss still effectively improves the based in 50 lags, with comparable improvement to 5 lags. This indicates the stable performance of our normalization w.r.t. long lags.

---

> > ### Author Response · Authors · 2025-11-16
> > **Part 2 of Response**
> >
> > ## Weakness 4
> >
> > We have add a new experiment to observe our method's performance under imperfect summary priors.
> >
> > Results for imperfect lag-agnostic prior edge presence:
> >
> > We tested on a 20-node ER4 graph (L=5, Gaussian) and report performance vs. the baseline, plus the recovery rates for both *correct* and *wrong* priors. (✅) marks where our method outperforms the baseline.
> >
> > | Method                       | Prior Error Rate (q) |      SHD       |        F1         | Num. Correct Priors | Correct Prior Recov. (%) | Num. Wrong Priors | Wrong Prior Recov. (%) |
> > | :--------------------------- | -------------------- | :------------: | :---------------: | :-----------------: | :----------------------: | :---------------: | :--------------------: |
> > | Baseline (DYNOTEARS)         | -                    |      15.0      |       0.897       |          -          |            -             |         -         |           -            |
> > | Ours (DYNOTEARS*, Init Data) | 0% ✅                 | **7.2 ± 1.6**  | **0.953 ± 0.011** |         57          |      96.20% ± 2.05%      |         0         |           -            |
> > |                              | 5%✅                  | **7.8 ± 1.5**  | **0.949 ± 0.010** |         54          |      95.99% ± 2.73%      |         3         |     5.56% ± 13.61%     |
> > |                              | 10%✅                 | **9.0 ± 2.6**  | **0.941 ± 0.017** |         51          |      96.41% ± 2.29%      |         6         |     11.11% ± 8.61%     |
> > |                              | 30%✅                 | **12.8 ± 2.2** | **0.916 ± 0.016** |         40          |      97.50% ± 1.58%      |        17         |     16.67% ± 4.43%     |
> > |                              | 50%❌                 |  15.833 ± 3.9  |   0.898 ± 0.023   |         29          |      98.85% ± 1.78%      |        28         |    19.05% ± 11.66%     |
> >
> > The results show our method (✅) **outperforms the baseline with up to 30% prior error.** This robustness stems from our soft-constraint design, which successfully **leverages "good" information** (recovering 96-99% of correct priors) while actively **rejecting "bad" information** (e.g., rejecting >83% of wrong priors at the 30% error level).
> >
> > Below are results on imperfect edge absence priors on a 20-node, ER4 graph  (L=5, Gaussian). We found that while **correct** absence priors are helpful (SHD improves from 15.0 to 13.5), the process is **highly sensitive to incorrect** (false absence) priors. Practitioners should be highly confident in absence priors, as false absences are extremely disruptive. We will add this analysis to the appendix.
> >
> > | Algorithm         | Num. Correct Priors | Num. Wrong Priors | SHD         | F1-Score      | Accuracy      | Recall        |
> > | ----------------- | ------------------- | ----------------- | ----------- | ------------- | ------------- | ------------- |
> > | DYNOTEARS         | -                   | -                 | 15          | 0.897         | 1.000         | 0.812         |
> > | DYNOTEARS-Absence | 163                 | 0                 | 13.5 ± 0.5  | 0.908 ± 0.004 | 1.000 ± 0.000 | 0.831 ± 0.007 |
> > | DYNOTEARS-Absence | 155                 | 8                 | 22.2 ± 3.0  | 0.843 ± 0.022 | 0.976 ± 0.029 | 0.742 ± 0.022 |
> > | DYNOTEARS-Absence | 147                 | 16                | 37.7 ± 6.1  | 0.729 ± 0.042 | 0.863 ± 0.063 | 0.631 ± 0.039 |
> > | DYNOTEARS-Absence | 114                 | 49                | 83.8 ± 5.4  | 0.325 ± 0.042 | 0.457 ± 0.061 | 0.252 ± 0.032 |
> > | DYNOTEARS-Absence | 81                  | 74                | 111.7 ± 1.9 | 0.000 ± 0.000 | 0.000 ± 0.000 | 0.000 ± 0.000 |

---

> > > ### Author Response · Authors · 2025-11-16
> > > **Part 3 of Response**
> > >
> > > ## Weakness 5
> > >
> > > Thanks for your suggestion. We have run a new set of experiments to test this. We replaced the standard DYNOTEARS backbone from our main experiments with the DYNOTEARS + Group Sparsity (GS) baseline you proposed.
> > >
> > > We then applied both the 'oracle' lag-specific (LS) priors and our lag-agnostic (DYNOTEARS*) priors on top of this new GS backbone. The results (20-node ER4, L=5, Gaussian) are below:
> > >
> > > | **Method**               | **5% Priors** |             | **10% Priors** |             | **30% Priors** |             | **50% Priors** |             |
> > > | :----------------------- | :------------ | :---------- | :------------- | :---------- | :------------- | :---------- | :------------- | :---------- |
> > > |                          | **SHD↓**      | **F1↑**     | **SHD↓**       | **F1↑**     | **SHD↓**       | **F1↑**     | **SHD↓**       | **F1↑**     |
> > > | DYNOTEARS (Group)        | 21.19 ± 2.55  | 0.80 ± 0.02 | 21.19 ± 2.55   | 0.80 ± 0.02 | 21.19 ± 2.55   | 0.80 ± 0.02 | 21.19 ± 2.55   | 0.80 ± 0.02 |
> > > | LS-0 , 0% error (Group)  | 19.04 ± 2.71  | 0.82 ± 0.02 | 16.98 ± 2.45   | 0.85 ± 0.01 | 13.67 ± 1.98   | 0.88 ± 0.01 | 13.29 ± 2.15   | 0.89 ± 0.01 |
> > > | LS-10 ,10% error (Group) | 20.10 ± 3.11  | 0.81 ± 0.03 | 18.04 ± 2.63   | 0.84 ± 0.02 | 16.04 ± 2.22   | 0.86 ± 0.02 | 15.33 ± 2.04   | 0.87 ± 0.01 |
> > > | LS-30 ,30% error (Group) | 20.10 ± 2.95  | 0.81 ± 0.03 | 18.56 ± 2.81   | 0.83 ± 0.02 | 16.69 ± 2.40   | 0.85 ± 0.02 | 16.33 ± 2.31   | 0.86 ± 0.02 |
> > > | LS-50 ,50% error (Group) | 20.10 ± 3.05  | 0.81 ± 0.03 | 18.56 ± 2.76   | 0.83 ± 0.02 | 18.48 ± 2.58   | 0.83 ± 0.02 | 19.69 ± 2.99   | 0.82 ± 0.03 |
> > > | Ours (NOTEARS*, Group)   | 19.21 ± 2.66  | 0.82 ± 0.02 | 17.25 ± 2.18   | 0.84 ± 0.01 | 14.29 ± 1.95   | 0.88 ± 0.01 | 14.23 ± 2.05   | 0.88 ± 0.01 |
> > >
> > > Our lag-agnostic prior provides a massive improvement to the GS baseline, reducing the SHD from 21.19 to 14.23 (with 50% priors).  Besides, our lag-agnostic method's performance nearly matches the oracle, perfect-lag-information baseline. This result demonstrates the effectiveness of our method when applying to the GS baseline.

---

> > > > ### Author Response · Authors · 2025-11-16
> > > > **Part 4 of Response**
> > > >
> > > > ## Weakness 6
> > > >
> > > > Thank you for this excellent suggestion. Indeed, a softmax or log-sum-exp surrogate is a natural alternative to address process-inequivalence.
> > > >
> > > > We have implemented a new baseline, **DYNOTEARS-softmax**, based on your suggestion. This method can be seen as a **weighted variant of our binary-masked (p_bin) loss**. While our p_bin loss applies a *uniform* sum penalty to all lags, this new baseline uses a softmax (calculated over the absolute weights of all lags) to *weight* the penalty for each specific lag.
> > > >
> > > > $p_{\text{softmax}} = \sum\left( \mathcal{C}_p \circ \mathbb{I}(\max_{\tau} |W| < \delta) \circ \left(\sum_{\tau} \left(\text{softmax}_{\tau}(W) \circ \text{ReLU}(\delta-W)\right)\right)\right)$
> > > >
> > > > Like our proposed methods, this formulation is also **process-equivalent**, as it considers all lags rather than defaulting to the one with the largest initial value. We have added it to our main comparison table (Table 1, using 20-node, ER4, L=5, Gaussian data) to compare it against the oracle (LS-0), the flawed max-based (^), and our two proposed methods (& and *).
> > > >
> > > > | **Method**                    | **5% Priors**  |               | **10% Priors** |               | **30% Priors** |               | **50% Priors** |               |
> > > > | :---------------------------- | :------------- | :------------ | :------------- | :------------ | :------------- | :------------ | :------------- | :------------ |
> > > > |                               | **SHD↓**       | **F1↑**       | **SHD↓**       | **F1↑**       | **SHD↓**       | **F1↑**       | **SHD↓**       | **F1↑**       |
> > > > | DYNOTEARS                     | 13.00±3.10     | 0.91±0.02     | 13.00±3.10     | 0.91±0.02     | 13.00±3.10     | 0.91±0.02     | 13.00±3.10     | 0.91±0.02     |
> > > > | LS-0, Perfect Lags            | **10.33±2.73** | **0.93±0.02** | **8.00±2.37**  | **0.95±0.02** | **4.83±1.94**  | **0.97±0.01** | **4.83±1.33**  | **0.97±0.01** |
> > > > | LS-10, 10% Error              | 11.67±3.44     | 0.92±0.02     | 8.17±2.64      | 0.95±0.02     | 6.67±2.58      | 0.96±0.02     | 6.50±2.43      | 0.96±0.02     |
> > > > | LS-30, 30% Error              | 12.50±3.62     | 0.92±0.03     | 9.50±3.27      | 0.94±0.02     | 8.83±1.33      | 0.94±0.01     | 8.33±2.16      | 0.95±0.02     |
> > > > | LS-50, 50% Error              | 12.50±3.62     | 0.92±0.03     | 10.67±2.73     | 0.93±0.02     | 11.17±1.47     | 0.93±0.01     | 11.83±1.94     | 0.92±0.01     |
> > > > | DYNOTEARS^ (Init Data)        | 11.50 ± 2.26   | 0.92 ± 0.02   | 9.33 ± 2.34    | 0.94 ± 0.02   | 7.83 ± 2.04    | 0.94 ± 0.02   | 6.83 ± 1.47    | 0.95 ± 0.01   |
> > > > | DYNOTEARS-softmax (Init Data) | 10.83 ± 2.88   | 0.93 ± 0.02   | 8.17 ± 2.32    | 0.95 ± 0.02   | 6.33 ± 1.94    | 0.96 ± 0.01   | 5.67 ± 2.16    | 0.96 ± 0.01   |
> > > > | DYNOTEARS& (Init Data)        | 10.83 ± 1.97   | 0.93 ± 0.01   | 8.50 ± 3.78    | 0.94 ± 0.03   | 5.83 ± 1.97    | 0.97 ± 0.01   | 6.17 ± 2.71    | 0.95 ± 0.02   |
> > > > | DYNOTEARS*  (Init Data)       | 10.50 ± 2.35   | 0.93 ± 0.02   | 8.00 ± 2.37    | 0.95 ± 0.02   | 5.33 ± 1.86    | 0.97 ± 0.01   | 5.50 ± 1.76    | 0.96 ± 0.01   |
> > > >
> > > > As expected, the new DYNOTEARS-softmax baseline achieves results on-par with our proposed binary-masked formulation, which outperforms the flawed max-based formulation. We will add this part to the paper.
> > > >
> > > >
> > > >
> > > > ## Questions
> > > >
> > > > The questions are addressed in our reply to their corresponding weakness.

---

> > > > > ### Comment · Reviewer_rVC5 · 2025-11-25
> > > > >
> > > > > Thanks a lot for addressing all my comments and supporting your claims with new experiments. I've updated my score accordingly. Good luck! 💪

---

### Official Review · Reviewer_tkmz · 2025-11-01

**Soundness:** 3
**Presentation:** 4
**Contribution:** 3
**Rating:** 6
**Confidence:** 5

**Summary:**

This paper proposes a lag-agnostic prior constraint for incorporating prior knowledge into causal discovery algorithms for time series data. The authors highlight the drawback of outcome-equivalent constraints like maximum-based prior and theoretically justify the effectiveness of their process-equivalent prior. Empirically, the paper shows the effectiveness of their method in incorporating the provided prior information.

**Strengths:**

1. The paper is very well-written and easy to read.
2. The theoretical results are well-motivated and clearly stated. The examples are well-constructed to illustrate the point that the authors are trying to convey.
3. The empirical results are extensive, and include several ablation studies.
4. The authors tackle an important problem that is often overlooked. Priors coming from domain-experts are almost always lag-agnostic so it's nice to see a paper that tackles this challenge.

**Weaknesses:**

1. Although Proposition 1 is illustrative, it is proven under a very strong assumption, i.e. $\nabla{|(W_\tau)_{ij}|} \mathcal{L} \geq 0$ for all $\tau$. This is unrealistic, since many edges in real-world applications are, in fact, not forced to 0 by the data. It is unclear whether the principle being illustrated would still hold in such cases.

2. The main experiments in Section 4 do not include the outcome equivalent baseline "maximum-based formulation". Although the authors report some experiments in Appendix E.2, the difference between the two methods seems quite small (especially for the init data setting). This is an important baseline that the authors should consider including in Table 1, Figure 1 and Table 2.

3. A minor weakness is that practical sources and reliability of the availability of such lag-agnostic priors could be discussed more concretely, especially in the motivation.

**Questions:**

1. Do the authors have intuition for why data driven initialization works?
2. As noted in the limitations section, the model can go wrong due to incorrect priors. How sensitive is the model to incorrect priors?

---

> ### Author Response · Authors · 2025-11-16
> **Part 1 of Response**
>
> Thanks for your patient review and invaluable comments. Below is the part 1 of our response.
>
> ## Weakness 1
>
> Thank you for this insightful comment. We agree the assumption ($\nabla_{|(W_{\tau})_{ij}|}\mathcal{L}\ge0$) is specific, but it is **intentionally chosen to model the non-trivial case** where the presence prior actively influences the optimization.
>
> We can distinguish two primary scenarios:
>
> 1. **Data Supports the Edge:** If the data-fitting loss $\mathcal{L}$ already pushes a weight $|(W_{\tau_0})_{ij}| > \delta$, the prior is **automatically satisfied and thus inactive**. In this case, the choice of loss function is irrelevant as no penalty is applied.
> 2. **Data Opposes the Edge (Our Assumption):** If $\mathcal{L}$ pushes all weights toward 0, a **data-prior conflict** arises that the loss function must resolve.
>
> It is only in this second, non-trivial "conflict" scenario that the prior becomes active and the loss formulation matters.
>
> **Supplementary Point on Optimization Dynamics**
>
> You are right to suggest that the optimization process is complex. We also note a subtle case, which your comment alludes to, that is not covered by our discussion above.
>
> - **Dynamic Support:** An edge $|(W_{\tau_0})_{ij}|$ might be *initially* encouraged by the data, but this support could fade during optimization (e.g., due to regularization or competition from other edges). We assume that edge $(i,j)$ with lag $\tau_0$ is the correct lag but not recovered.
> - **Behavior:** In this case, the `max`-based formulation *might* select the correct lag $\tau_0$, *if* $|(W_{\tau_0})_{ij}|$ is encouraged to be the maximum value at an early step. However, this does not fix the **fundamental theoretical flaw of process-inequivalence**: the loss does not perform a principled, holistic comparison of all lags.
>
> ## Weakness 2
>
> Thank you for this excellent suggestion. We agree that this is a crucial comparison and have now incorporated the maximum-based formulation (denoted `DYNOTEARS^`) into the main experimental tables.
>
> Your observation is correct: on synthetic data *with* the data-driven initialization (Init Data), the performance gap between our process-equivalent method (DYNOTEARS& and DYNOTEARS*) and the flawed DYNOTEARS^ is reduced.
>
> This is expected. Our Init Data strategy is a strong heuristic designed to find a good starting point, which **mitigates the instability of the flawed `max`-based loss**. It helps all methods, including the flawed one, by guiding them away from the worst local optima.
>
> However, this heuristic merely masks the underlying theoretical flaw. This becomes evident on the complex, real-world DREAM4 dataset. Here, the initialization heuristic is no longer sufficient, and the process-inequivalence of DYNOTEARS^ is fully exposed, leading to a significant performance gap as shown below (partial results of the updated Table 2).
>
> | @Priors | Method                  | DREAM4-63     |               | DREAM4-126    |               | DREAM4-189    |               |
> | ------- | ----------------------- | ------------- | ------------- | ------------- | ------------- | ------------- | ------------- |
> |         |                         | Loss↓         | AUROC↑        | Loss↓         | AUROC↑        | Loss↓         | AUROC↑        |
> | 75%     | DYNOTEARS               | 7.81±1.14     | 0.58±0.03     | 5.61±1.04     | 0.62±0.04     | 4.08±0.60     | 0.65±0.05     |
> |         | DYNOTEARS^  (Init Data) | 7.58±0.11     | 0.62±0.04     | 5.43±0.12     | 0.67±0.03     | 4.35±0.06     | 0.71±0.06     |
> |         | DYNOTEARS& (Init Data)  | **6.02±0.78** | **0.90±0.01** | **4.46±0.64** | **0.91±0.01** | **3.25±0.33** | **0.92±0.02** |
> |         | DYNOTEARS* (Init Data)  | 6.22±0.72     | 0.89±0.01     | 4.88±0.85     | 0.90±0.01     | 3.40±0.42     | **0.92±0.02** |
>
> We also present partial results for Table 1 below (Figure 1 is omitted here).
>
> | **Method**              | **5% Priors** |             | **10% Priors** |             | **30% Priors** |             | **50% Priors** |             |
> | :---------------------- | :------------ | :---------- | :------------- | :---------- | :------------- | :---------- | :------------- | :---------- |
> |                         | **SHD↓**      | **F1↑**     | **SHD↓**       | **F1↑**     | **SHD↓**       | **F1↑**     | **SHD↓**       | **F1↑**     |
> | DYNOTEARS               | 13.00±3.10    | 0.91±0.02   | 13.00±3.10     | 0.91±0.02   | 13.00±3.10     | 0.91±0.02   | 13.00±3.10     | 0.91±0.02   |
> | DYNOTEARS^ (Init Data)  | 11.50 ± 2.26  | 0.92 ± 0.02 | 9.33 ± 2.34    | 0.94 ± 0.02 | 7.83 ± 2.04    | 0.94 ± 0.02 | 6.83 ± 1.47    | 0.95 ± 0.01 |
> | DYNOTEARS& (Init Data)  | 10.83 ± 1.97  | 0.93 ± 0.01 | 8.50 ± 3.78    | 0.94 ± 0.03 | 5.83 ± 1.97    | 0.97 ± 0.01 | 6.17 ± 2.71    | 0.95 ± 0.02 |
> | DYNOTEARS*  (Init Data) | 10.50 ± 2.35  | 0.93 ± 0.02 | 8.00 ± 2.37    | 0.95 ± 0.02 | 5.33 ± 1.86    | 0.97 ± 0.01 | 5.50 ± 1.76    | 0.96 ± 0.01 |
>
> We will update the paper soon.

---

> > ### Author Response · Authors · 2025-11-16
> > **Part 2 of Response**
> >
> > ## Weakness 3
> >
> > Thank you for this valuable suggestion. We fully agree that a more concrete discussion of prior sources will strengthen the paper's motivation. We will revise the paper to include this, based on the following points:
> >
> > 1. **Classical Source: Domain Experts:** The primary source for priors in causal discovery is **domain expertise** [1]. This expertise is almost always at a coarse-grained, semantic level. For example, in genetics (a domain relevant to our DREAM4 experiments) or neuroscience, an expert may confidently state that "Gene A regulates Gene B" or "Region X influences Region Y." It is extremely rare for them to know the *exact time lag* of this interaction. This common gap between high-level knowledge and low-level temporal detail is the **central motivation for our lag-agnostic framework**.
> > 2. **Emerging Source: Large Language Models (LLMs):** A powerful, emerging source of high-level causal priors is LLMs. Research is rapidly developing reliable frameworks to extract summary causal relationships from them [2,3]. These methods provide **inherently lag-agnostic** and rich priors, bypassing the need for expensive and time-consuming expert specification. Motivated by this new, abundant source of information, our work provides the necessary and timely mechanism to formally integrate such high-level, LLM-generated priors into time-series models.
> > 3. **Reliability and Robustness:** Your point on reliability is key, which is guaranteed in both the prior derivation stage and prior integration stage:
> >    - First, as research in LLM-supervision shows, edge priors can be made more reliable by combining their derivation with data-driven methods [4].
> >    - Second, our model's design **is robust to imperfect priors to some extent**. By using a **soft penalty** (as discussed in Section 3.2), our framework is inherently robust to some prior error. It allows the model to weigh the prior against strong contradictory evidence from the data, which is a more realistic and reliable approach than enforcing a hard, brittle constraint. We also present experimental results on this point in the reply to your Question 2.
> >
> > We will add these discussions to the paper. Thank you for helping us improve the paper's framing.
> >
> > *[1] Amirkhani, H., et al. (2016). Exploiting experts’ knowledge for structure learning of Bayesian networks. IEEE TPAMI.*
> >
> > *[2] Abdulaal, A., et al. (2023). Causal modelling agents: Causal graph discovery through synergising metadata-and data-driven reasoning. ICLR 2024.*
> >
> > *[3] Vashishtha, A., et al. (2025). Causal order: The key to leveraging imperfect experts in causal inference. ICLR 2025.*
> >
> > *[4] Ban, T., et al. (2023). Causal structure learning supervised by large language model. arXiv.*

---

> > > ### Author Response · Authors · 2025-11-16
> > > **Part 3 of Response**
> > >
> > > ## Question 1
> > >
> > > Thanks for this insightful question. The data-driven initialization is crucial because it navigates the **severe, combined non-convexity** of the problem, which is further amplified by a **direct gradient conflict** between the two main constraints.
> > >
> > > **The challenge stems from the interacting issues:**
> > >
> > > 1. **Added Non-Convexity (Prior):** As shown in **Example 1**, our lag-agnostic prior loss introduces *additional* non-convexity and multiple local optima 2.
> > > 2. **Gradient Conflict (DAG vs. Prior):** These two constraints are in direct opposition.
> > >    - The **edge presence prior** pushes weights *away from zero* to satisfy the condition.
> > >    - The **DAG constraint** (via its augmented Lagrangian) and L1 regularization push weights *toward zero* to enforce acyclicity and sparsity.
> > >
> > > The "Failure Case" of Zero Initialization (Init 0):
> > >
> > > - In the early steps, the lag-agnostic prior may incorrectly try to satisfy its constraint by forcing an **instantaneous edge (in $W_0$) to grow** (even if a lagged edge is the true global optimum).
> > > - This can "corrupt" the topological order of $W_0$. When the DAG constraint's augmented Lagrangian penalty ramps up to a large value, the optimizer is  **trapped in a poor local minimum**. The corrupted ordering cannot be rectified because the DAG penalty is now too strong to be overcome by the data or prior loss.
> > >
> > > **Why Data-Driven Initialization Succeeds:**
> > >
> > > The "Init Data" approach **decouples this conflict**:
> > >
> > > 1. **Stage 1** first finds a high-quality, stable $W_0$ that satisfies the DAG constraint, providing a principled topological ordering based on the data.
> > > 2. **Stage 2** then introduces the prior, starting from this "good" region. Crucially, during the initial iterations of Stage 2 (when the DAG constraint's weight is not yet overpowering), the optimizer can effectively **balance all three objectives**: (1) data-fit, (2) prior adherence, and (3) (prior, data)-driven rectifications to the already-good $W_0$ ordering.
> > >
> > > In short, "Init Data" provides a good starting point for the topological order, allowing the optimizer to properly balance the priors with the data loss to find a superior optimum. A similar two-stage technique has been shown to be effective for incorporating other non-convex prior losses (ancestral constraints) into differentiable structure learning [5].
> > >
> > > [5] Ban, T., et al. (2025). Differentiable Structure Learning with Ancestral Constraints. ICML 2025.
> > >
> > > To further illustrate this point, we compared Init 0 and Init Data for **edge absence** constraints. Unlike presence priors, absence constraints (forcing $W_{\tau}[i,j] = 0$) **align** with the DAG constraint, as both push weights toward zero. There is **no gradient conflict**. As hypothesized, the initialization strategy is not critical, and performance is **identical**:
> > >
> > > | Algorithm                     |    SHD     |   F1-Score    |   Accuracy    |    Recall     |
> > > | :---------------------------- | :--------: | :-----------: | :-----------: | :-----------: |
> > > | Baseline (DYNOTEARS)          |     21     |     0.859     |     0.927     |      0.8      |
> > > | DYNOTEARS-Absence (Init 0)    | 16.8 ± 1.9 | 0.887 ± 0.013 | 0.959 ± 0.016 | 0.825 ± 0.014 |
> > > | DYNOTEARS-Absence (Init data) | 16.8 ± 1.9 | 0.887 ± 0.013 | 0.959 ± 0.016 | 0.825 ± 0.014 |
> > >
> > > ##

---

> > > > ### Author Response · Authors · 2025-11-16
> > > > **Part 4 of Response**
> > > >
> > > > ## Question 2
> > > >
> > > > Thank you for this excellent question. We ran a new experiment to test robustness to imperfect priors, adding an increasing percentage of *incorrect* presence priors.
> > > >
> > > > We tested on a 20-node ER4 graph (L=5, Gaussian) and report performance vs. the baseline, plus the recovery rates for both *correct* and *wrong* priors. (✅) marks where our method outperforms the baseline.
> > > >
> > > >
> > > >
> > > > | Method                        | Prior Error Rate (q) |      SHD       |        F1         | Num. Correct Priors | Correct Prior Recov. (%) | Num. Wrong Priors | Wrong Prior Recov. (%) |
> > > > | :---------------------------- | -------------------- | :------------: | :---------------: | :-----------------: | :----------------------: | :---------------: | :--------------------: |
> > > > | Baseline (DYNOTEARS)          | -                    |      15.0      |       0.897       |          -          |            -             |         -         |           -            |
> > > > | Ours (DYNOTEARS\*, Init Data) | 0% ✅                 | **7.2 ± 1.6**  | **0.953 ± 0.011** |         57          |      96.20% ± 2.05%      |         0         |           -            |
> > > > |                               | 5%✅                  | **7.8 ± 1.5**  | **0.949 ± 0.010** |         54          |      95.99% ± 2.73%      |         3         |     5.56% ± 13.61%     |
> > > > |                               | 10%✅                 | **9.0 ± 2.6**  | **0.941 ± 0.017** |         51          |      96.41% ± 2.29%      |         6         |     11.11% ± 8.61%     |
> > > > |                               | 30%✅                 | **12.8 ± 2.2** | **0.916 ± 0.016** |         40          |      97.50% ± 1.58%      |        17         |     16.67% ± 4.43%     |
> > > > |                               | 50%❌                 |  15.833 ± 3.9  |   0.898 ± 0.023   |         29          |      98.85% ± 1.78%      |        28         |    19.05% ± 11.66%     |
> > > >
> > > > The results show our method (✅) **outperforms the baseline with up to 30% prior error.** This robustness stems from our soft-constraint design, which successfully **leverages "good" information** (recovering 96-99% of correct priors) while actively **rejecting "bad" information** (e.g., rejecting >83% of wrong priors at the 30% error level).
> > > >
> > > > This confirms our framework is not brittle and is well-suited for realistic scenarios where priors are imperfect.

---

> > > > > ### Comment · Reviewer_tkmz · 2025-11-28
> > > > >
> > > > > Thank you for your detailed responses. The discussions and changes will certainly make the revised version better.
> > > > >
> > > > > However, I am still confused about Proposition 1, especially with the (changed?) statement. There seems to be something missing in the statement.
> > > > >
> > > > > "Then, optimizing Equation (8) with the penalty in Equation (9) will satisfy the constraint solely via $(W_{\tau_0} )_{ij}$ if with sufficiently large \lambda_p$"
> > > > >
> > > > > Could the authors clarify and correct it?

---

> > > > > > ### Author Response · Authors · 2025-11-28
> > > > > >
> > > > > > Thanks for pointing out the ambiguity in the original statement of Proposition 1. We have clarified the proposition in the revised manuscript.
> > > > > >
> > > > > > > Let $ (i,j) $ be a a lag-agnostic edge specified to be present. Assume the ....
> > > > > > >
> > > > > > > (Revised Part) Let $ W^{\text{opt}} $ be the solution of the structure learning problem defined in Equation (8) using the maximum-based prior loss (Equation 9). Then, for sufficiently large $\lambda_p$, the optimum satisfies: $|(W^{\text{opt}}_{\tau_0})_{ij}| \ge \delta \quad \text{and} \quad \forall \tau \neq \tau_0,\, |(W^{\text{opt}}_\tau)_{ij}| < \delta.$
> > > > > >
> > > > > > This result confirms that the maximum-based formulation, when faced with a conflict between the data (which pushes weights to zero) and the prior (which demands existence), **solely relies on the initial starting weights**. It forces the edge only via the single lag ($\tau_0$) that happened to be the largest at initialization. This demonstrates that the optimization process is biased and **fails to consider other candidate lags** based on optimal data-fit, which is the definition of process-inequivalence.

---

### Author Response · Authors · 2025-11-20
**Summary of Paper Revisions**

We sincerely thank the reviewers for their insightful feedback, which has allowed us to significantly strengthen the paper. We have revised the manuscript to incorporate all critical findings and address key concerns raised during the discussion period.

## Summary of Major Revisions

### Empirical Validations (New Experiments Added)

1. Results of maximum-based formulation and init 0 results in the main experiment section
2. Results when the lag-agnostic priors are imperfect
3. Results compared with group sparsity baseline and softmax-based formulation
4. Results on large scale nodes (100 and 200 nodes) and time complexity analysis
5. Results on product-based loss formulation with large lags (up to L=50)
6. Results on comparing to a new VAR-based initialization

### Theoretical & Discussion Clarifications (New Content Added)

7. We clarify the lack of the global-optima theoretical guarantee of our approach in the contribution, which is originally stated only in the limitation section.
8. We add the formal definition of process-equivalence of loss formulation, a critical research problem of our work.
9. We add the formal result of additional non-convexity that the lag-agnostic prior loss brings, which is stated by an example originally.
10. We add a discussion of the connection and difference of our lag-agnostic prior to existing static priors in the related work section
11. We add a discussion of the practical source and reliability of the prior in the related work section
12. We revise the ambiguous symbol pointed out by reviewer 54Cj

We believe these comprehensive revisions have enhanced both the clarity of our technical contributions and the empirical strength of our results. Please feel free if any additional revisions should be updated.

---

### Meta-Review · Area_Chair_MaGF · 2026-01-06

**Summary:**

Across reviews, the work was recognized as well-written, theoretically grounded, and addressing an important but underexplored problem. Reviewers raised concerns primarily about:
(i) the strength and generality of the theoretical analysis,
(ii) missing or incomplete baselines and ablations,
(iii) robustness to imperfect priors,
(iv) optimization behavior under non-convexity, and
(v) clarity and formalization of key concepts such as process equivalence.

The authors provided an extensive rebuttal with substantial new theoretical clarification, multiple new baselines, ablation studies, robustness experiments, scalability analysis, and notation improvements. Several reviewers explicitly stated that their concerns were sufficiently addressed and updated their scores accordingly, while others maintained their original (borderline-positive) assessments.

**Reviewer Concerns:**

### Concerns addressed by the rebuttal
Concerns addressed by the rebuttal include:
(i) missing baselines and empirical validation of process equivalence;
(ii) improved theoretical clarity and rigor;
(iii) robustness to imperfect priors;
(iv) clarification of the initialization strategy and optimization behavior;
(v) scalability and applicability across different backbones;
and (vi) improved clarity, notation, and overall motivation.

### Concerns still outstanding
- **Global optimization guarantees**:
While the authors clarify the scope and limitations of their theoretical analysis, there remains no formal guarantee of convergence to globally optimal lag assignments under identifiability assumptions. This limitation is acknowledged and appropriately framed.

**Reviewer Scores:**

**Reviewer `tkmz` (Score: 6, marginally above acceptance)**

After the rebuttal and clarifications, this reviewer explicitly noted that the discussions and changes substantially improved the paper. The score was maintained at a marginally positive level.

**Reviewer `rVC5` (Score updated upward)**

This reviewer stated that their concerns were addressed and updated their score accordingly, indicating a positive reassessment after the rebuttal.

**Reviewer `2t4w` (Score: 6, unchanged)**

This reviewer acknowledged the rebuttal but stated that their score remained unchanged, maintaining a marginally positive assessment.

**Reviewer `54Cj` (Score updated upward)**

The reviewer explicitly confirmed that their concerns had been sufficiently addressed and updated their score.

---

### Decision · Program_Chairs · 2026-01-26

Accept (Poster)